# Behind the Scenes of Gradient Descent: A Trajectory Analysis via Basis Function Decomposition

**Jianhao Ma, Lingjun Guo, Salar Fattahi**
Department of Industrial and Operational Engineering, University of Michigan, Ann Arbor
`{jianhao, glingjun, fattahi}@umich.edu`

## Abstract

This work analyzes the solution trajectory of gradient-based algorithms via a novel *basis function decomposition*. We show that, although solution trajectories of gradient-based algorithms may vary depending on the learning task, they behave almost monotonically when projected onto an appropriate orthonormal function basis. Such projection gives rise to a basis function decomposition of the solution trajectory. Theoretically, we use our proposed basis function decomposition to establish the convergence of gradient descent (GD) on several representative learning tasks. In particular, we improve the convergence of GD on symmetric matrix factorization and provide a completely new convergence result for the orthogonal symmetric tensor decomposition. Empirically, we illustrate the promise of our proposed framework on realistic deep neural networks (DNNs) across different architectures, gradient-based solvers, and datasets. Our key finding is that gradient-based algorithms monotonically learn the coefficients of a particular orthonormal function basis of DNNs defined as the eigenvectors of the conjugate kernel after training. Our code is available at `github.com/jianhaoma/function-basis-decomposition`.

## 1 Introduction

Learning highly nonlinear models amounts to solving a nonconvex optimization problem, which is typically done via different variants of gradient descent (GD). But how does GD learn nonlinear models? Classical optimization theory asserts that, in the face of nonconvexity, GD and its variants may lack any meaningful optimality guarantee; they produce solutions that—while being first- or second-order optimal (Nesterov, 1998; Jin et al., 2017)—may not be globally optimal. In the rare event where the GD can recover a globally optimal solution, the recovered solution may correspond to an overfitted model rather than one with desirable generalization.

Inspired by the large empirical success of gradient-based algorithms in learning complex models, recent work has postulated that typical training losses have *benign landscapes*: they are devoid of spurious local minima and their global solutions coincide with true solutions—i.e., solutions corresponding to the true model. For instance, different variants of low-rank matrix factorization (Ge et al., 2016; 2017) and deep linear NNs (Kawaguchi, 2016) have benign landscapes. However, when spurious solutions *do exist* (Safran & Shamir, 2018) or global and true solutions *do not coincide* (Ma & Fattahi, 2022b), such a holistic view of the optimization landscape cannot explain the success of gradient-based algorithms. To address this issue, another line of research has focused on analyzing the solution trajectory of different algorithms. Analyzing the solution trajectory has been shown extremely powerful in sparse recovery (Vaskevicius et al., 2019), low-rank matrix factorization (Li et al., 2018; Stöger & Soltanolkotabi, 2021), and linear DNNs (Arora et al., 2018; Ma & Fattahi, 2022a). However, these analyses are tailored to specific models and thereby cannot be directly generalized.

In this work, we propose a unifying framework for analyzing the optimization trajectory of GD based on a novel *basis function decomposition*. We show that, although the dynamics of GD may vary

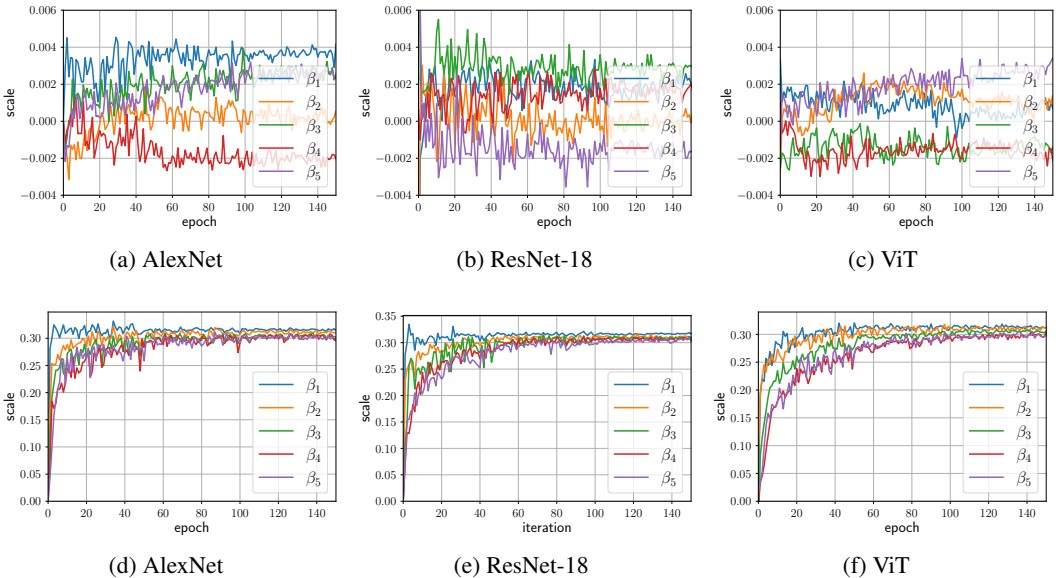

Figure 1: The solution trajectories of LARS on AlexNet and ResNet-18 and AdamW on ViT with $\ell_2$-loss after projecting onto two different orthonormal bases. The first row shows the trajectories of the top-5 coefficients after projecting onto a randomly generated orthonormal basis. The second row shows the trajectories of the top-5 coefficients after projecting onto the eigenvectors of the conjugate kernel evaluated at the last epoch. More detail on our implementation can be found in Appendix B.

drastically on different models, they behave almost monotonically when projected onto an appropriate choice of orthonormal function basis.

**Motivating example:**  Our first example illustrates this phenomenon on DNNs. We study the optimization trajectories of two adaptive gradient-based algorithms, namely AdamW and LARS, on three different DNN architectures, namely AlexNet (Krizhevsky et al., 2017), ResNet-18 (He et al., 2016), and Vision Transformer (ViT) (Dosovitskiy et al., 2020) with the CIFAR-10 dataset. The first row of the Figure 1 shows the top-5 coefficients of the solution trajectory when projected onto a randomly generated orthonormal basis. We see that the trajectories of the coefficients are highly non-monotonic and almost indistinguishable (they range between -0.04 to 0.06), implying that the energy of the obtained model is spread out on different orthogonal components. The second row of Figure 1 shows the same trajectory after projecting onto an orthogonal basis defined as the eigenvectors of the conjugate kernel after training (Long, 2021) (see Section 3.4 and Appendix B for more details). Unlike the previous case, the top-5 coefficients carry more energy and behave monotonically (modulo the small fluctuations induced by the stochasticity in the algorithm) in all three architectures, until they plateau around their steady state. In other words, *the algorithm behaves more monotonically after projecting onto a correct choice of orthonormal basis.*

## 1.1 MAIN CONTRIBUTIONS

The monotonicity of the projected solution trajectory motivates the use of an appropriate basis function decomposition to analyze the behavior of gradient-based algorithms. In this paper, we show how an appropriate basis function decomposition can be used to provide a much simpler convergence analysis for gradient-based algorithms on several representative learning problems, from simple kernel regression to complex DNNs. Our main contributions are summarized below:

- **Global convergence of GD via basis function decomposition:** We prove that GD learns the coefficients of an appropriate function basis that forms the true model. In particular, we show that GD learns the true model when applied to the expected $\ell_2$-loss under certain gradient independence and gradient dominance conditions. Moreover, we characterize the convergence rate of GD, identifying conditions under which it enjoys linear or sublinear

convergence rates. Our result does not require a benign landscape for the loss function and can be applied to both convex and nonconvex settings.

- **Application in learning problems:** We show that our general framework is well-suited for analyzing the solution trajectory of GD on different representative learning problems. Unlike the existing results, our proposed method leads to a much simpler trajectory analysis of GD for much broader classes of models. Using our technique, we improve the convergence of GD on the symmetric matrix factorization and provide an entirely new convergence result for GD on the orthogonal symmetric tensor decomposition. We also prove that GD enjoys an incremental learning phenomenon in both problems.

- **Empirical validation on DNNs:** We empirically show that our proposed framework applies to DNNs beyond GD. More specifically, we show that different gradient-based algorithms monotonically learn the coefficients of a particular function basis defined as the eigenvectors of the conjugate kernel after training (also known as "after kernel regime"). We show that this phenomenon happens across different architectures, datasets, solvers, and loss functions, strongly motivating the use of function basis decomposition to study deep learning.

## 2  General Framework: Function Basis Decomposition

We study the optimization trajectory of GD on the expected (population) $\ell_2$-loss

$$\min_{\boldsymbol{\theta} \in \Theta} \mathcal{L}(\boldsymbol{\theta}) := \frac{1}{2} \mathbb{E}_{\boldsymbol{x},y}\left[ (f_{\boldsymbol{\theta}}(\boldsymbol{x}) - y)^2 \right]. \quad \text{(expected $\ell_2$-loss)}$$

Here the input $\boldsymbol{x} \in \mathbb{R}^d$ is drawn from an unknown distribution $\mathcal{D}$, and the output label $y$ is generated as $y = f^\star(\boldsymbol{x}) + \varepsilon$, where $\varepsilon$ is an additive noise, independent of $\boldsymbol{x}$, with mean $\mathbb{E}[\varepsilon] = 0$ and variance $\mathbb{E}[\varepsilon^2] = \sigma_\varepsilon^2 < \infty$. The model $f_{\boldsymbol{\theta}}(\boldsymbol{x})$ is characterized by a parameter vector $\boldsymbol{\theta} \in \mathbb{R}^m$, which naturally induces a set of admissible models (model space for short) $\mathcal{F}_\Theta := \{f_{\boldsymbol{\theta}} : \boldsymbol{\theta} \in \mathbb{R}^m\}$. We do not require the true model $f^\star$ to lie within the model space; instead, we seek to obtain a model $f_{\boldsymbol{\theta}^\star} \in \mathcal{F}_\Theta$ that is closest to $f^\star$ in $L^2(\mathcal{D})$-distance. In other words, we consider $f^\star = f_{\boldsymbol{\theta}^\star}(\boldsymbol{x}) + f_\perp^\star(\boldsymbol{x})$, where $\boldsymbol{\theta}^\star = \arg\min_{\boldsymbol{\theta}} \|f_{\boldsymbol{\theta}} - f^\star\|_{L^2(\mathcal{D})}$. [1] To minimize the expected $\ell_2$-loss, we use vanilla GD with constant step-size $\eta > 0$:

$$\boldsymbol{\theta}_{t+1} = \boldsymbol{\theta}_t - \eta \nabla \mathcal{L}(\boldsymbol{\theta}_t). \quad \text{(GD)}$$

**Definition 1** (Orthonormal function basis). *A set of functions $\{\phi_i(\boldsymbol{x})\}_{i \in \mathcal{I}}$ forms an **orthonormal function basis** for the model space $\mathcal{F}_\Theta$ with respect to the $L^2(\mathcal{D})$-metric if*

- *for any $i \in \mathcal{I}$, we have $\mathbb{E}_{x \sim \mathcal{D}}[\phi_i^2(\boldsymbol{x})] = 1$;*

- *for any $i, j \in \mathcal{I}$ such that $i \neq j$, we have $\mathbb{E}_{x \sim \mathcal{D}}[\phi_i(\boldsymbol{x})\phi_j(\boldsymbol{x})] = 0$;*

- *for any $f_{\boldsymbol{\theta}} \in \mathcal{F}_\Theta$, there exists a unique sequence of basis coefficients $\{\beta_i(\boldsymbol{\theta})\}_{i \in \mathcal{I}}$ such that $f_{\boldsymbol{\theta}}(\boldsymbol{x}) = \sum_{i \in \mathcal{I}} \beta_i(\boldsymbol{\theta}) \phi_i(\boldsymbol{x})$.*

**Example 1** (Orthonormal basis for polynomials). *Suppose that $\mathcal{F}_\Theta$ is the class of all univariate real polynomials of degree at most $n$, that is, $\mathcal{F}_\Theta = \{\sum_{i=1}^{n+1} \theta_i x^{i-1} : \boldsymbol{\theta} \in \mathbb{R}^{n+1}\}$. If $\mathcal{D}$ is a uniform distribution on $[-1, 1]$, then the so-called Legendre polynomials form an orthonormal basis for $\mathcal{F}_\Theta$ with respect to the $L^2(\mathcal{D})$-metric (Olver et al., 2010, Chapter 14). Moreover, if $D$ is a normal distribution, then Hermite polynomials define an orthonormal basis for $\mathcal{F}_\Theta$ with respect to the $L^2(\mathcal{D})$-metric (Olver et al., 2010, Chapter 18).[2]*

**Example 2** (Orthonormal basis for symmetric matrix factorization). *Suppose that the true model is defined as $f_{\boldsymbol{U}^\star}(\mathbf{X}) = \langle \boldsymbol{U}^\star \boldsymbol{U}^{\star\top}, \mathbf{X} \rangle$ with some rank-$r$ matrix $\boldsymbol{U}^\star \in \mathbb{R}^{d \times r}$, and consider an "overparameterized" function class $\mathcal{F}_\Theta = \{f_{\boldsymbol{U}}(\mathbf{X}) : \boldsymbol{U} \in \mathbb{R}^{d \times r'}\}$ where $r' \geq r$ is an overestimation of the rank. Moreover, suppose that the elements of $\mathbf{X} \sim \mathcal{D}$ are iid with zero mean and unit variance. Consider the eigenvalues of $\boldsymbol{U}^\star \boldsymbol{U}^{\star\top}$ as $\sigma_1 \geq \cdots \geq \sigma_d$ with $\sigma_{r+1} = \cdots = \sigma_d = 0$, and their*

---

[1] Given a probability distribution $\mathcal{D}$, we define the $L_2(\mathcal{D})$-norm as $\|f\|_{L_2(\mathcal{D})}^2 = \mathbb{E}_{\mathbf{x} \sim \mathcal{D}}\left[ f^2(\mathbf{x}) \right]$.

[2] Both Legendre and Hermite polynomials can be derived sequentially using Gram-Schmidt procedure. For instance, the first three Legendre polynomials are defined as $P_1(x) = 1/\sqrt{2}$, $P_2(x) = \sqrt{3/2}x$, and $P_3(x) = \sqrt{5/8}(3x^2 - 1)$.

*corresponding eigenvectors $z_1, \ldots, z_d$. It is easy to verify that the functions $\phi_{ij}(\mathbf{X}) = \langle z_i z_j^\top, \mathbf{X} \rangle$ for $1 \le i, j \le d$ define a valid orthogonal basis for $\mathcal{F}_\Theta$ with respect to the $L^2(\mathcal{D})$-metric. Moreover, for any $f_U(\mathbf{X})$, the basis coefficients can be obtained as $\beta_{ij}(U) = \mathbb{E}\left[\langle UU^\top, \mathbf{X} \rangle \langle z_i z_j^\top, \mathbf{X} \rangle\right] = \langle z_i z_j^\top, UU^\top \rangle$. As will be shown in Section 3.2, this choice of orthonormal basis significantly simplifies the dynamics of GD for symmetric matrix factorization.*

Given the input distribution $\mathcal{D}$, we write $f_{\theta^\star}(x) = \sum_{i \in \mathcal{I}} \beta_i(\theta^\star) \phi_i(x)$, where $\{\phi(x)\}_{i \in \mathcal{I}}$ is an orthonormal basis for $\mathcal{F}_\Theta$ with respect to $L^2(\mathcal{D})$-metric, and $\{\beta_i(\theta^\star)\}_{i \in \mathcal{I}}$ are the *true basis coefficients*. For short, we denote $\beta_i^\star = \beta_i(\theta^\star)$. In light of this, the expected loss can be written as:

$$\mathcal{L}(\theta) = \underbrace{\frac{1}{2} \sum_{i \in \mathcal{I}} (\beta_i(\theta) - \beta_i^\star)^2}_{\text{optimization error}} + \underbrace{\frac{1}{2} \|f_\perp^\star\|_{L^2(\mathcal{D})}^2}_{\text{approximation error}} + \underbrace{\sigma_\varepsilon^2/2}_{\text{noise}}. \tag{1}$$

Accordingly, GD takes the form

$$\theta_{t+1} = \theta_t - \eta \sum_{i \in \mathcal{I}} (\beta_i(\theta_t) - \beta_i^\star) \nabla \beta_i(\theta_t). \tag{GD dynamic}$$

Two important observations are in order based on GD dynamic: first, due to the decomposed nature of the expected loss, the solution trajectory becomes independent of the approximation error and noise. Second, in order to prove the global convergence of GD, it suffices to show the convergence of $\beta_i(\theta_t)$ to $\beta_i^\star$. In fact, we will show that the coefficients $\beta_i(\theta_t)$ enjoy simpler dynamics for particular choices of orthonormal basis that satisfy appropriate conditions.

**Assumption 1** (Boundedness and smoothness). *There exist constants $L_f, L_g, L_H > 0$ such that*

$$\|f_\theta\|_{L^2(\mathcal{D})} \le L_f, \|\nabla f_\theta\|_{L^2(\mathcal{D})} \le L_g, \|\nabla^2 f_\theta\|_{L^2(\mathcal{D})} \le L_H. \tag{2}$$

The above assumptions are common in the optimization literature (Bubeck et al., 2015) and sometimes necessary to ensure the convergence of local-search algorithms (Patel & Berahas, 2022). Moreover, although these assumptions may not hold globally, all of our subsequent results hold when Assumption 1 is satisfied within any bounded region for $\theta$ that includes the solution trajectory. We will also relax these assumptions for several learning problems.

**Proposition 1** (Dynamic of $\beta_i(\theta_t)$). *Under Assumption 1 and based on GD dynamic, we have*

$$\beta_i(\theta_{t+1}) = \beta_i(\theta_t) - \eta \sum_{j \in \mathcal{I}} (\beta_j(\theta_t) - \beta_j^\star) \langle \nabla \beta_i(\theta_t), \nabla \beta_j(\theta_t) \rangle \pm \mathcal{O}(\eta^2 L_H L_f^2 L_g^2). \tag{3}$$

The above proposition holds for *any* valid choice of orthonormal basis $\{\phi_i(x)\}_{i \in \mathcal{I}}$. Indeed, there may exist multiple choices for the orthonormal basis, and not all of them would lead to equally simple dynamics for the coefficients. Examples of "good" and "bad" choices of orthonormal basis were presented for DNNs in our earlier motivating example. Indeed, an *ideal* choice of orthogonal basis should satisfy $\langle \nabla \beta_i(\theta_t), \nabla \beta_j(\theta_t) \rangle \approx 0$ for $i \ne j$, i.e., the gradients of the coefficients remain orthogonal along the solution trajectory. Under such assumption, the dynamics of $\beta_i(\theta_t)$ *almost* decompose over different indices:

$$\beta_i(\theta_{t+1}) \approx \beta_i(\theta_t) - \eta (\beta_i(\theta_t) - \beta_i^\star) \|\nabla \beta_i(\theta_t)\|^2 \pm \mathcal{O}(\eta^2), \tag{4}$$

where the last term accounts for the second-order interactions among the basis coefficients. If such an ideal orthonormal basis exists, then our next theorem shows that GD efficiently learns the true basis coefficients. To streamline the presentation, we assume that $\beta_1^\star \ge \cdots \ge \beta_k^\star > 0$ and $\beta_i^\star = 0, i > k$ for some $k < \infty$. We refer to the index set $\mathcal{S} = \{1, \ldots, k\}$ as *signal* and the index set $\mathcal{E} = \mathcal{I} \backslash \mathcal{S}$ as *residual*. When there is no ambiguity, we also refer to $\beta_i(\theta_t)$ as a signal if $i \in \mathcal{S}$.

**Theorem 1** (Convergence of GD with finite ideal basis). *Suppose that the initial point $\theta_0$ satisfies*

$$\beta_i(\theta_0) \ge C_1 \alpha, \qquad \text{for all } i \in \mathcal{S}, \qquad\qquad \text{(lower bound on signals at } \theta_0\text{)}$$

$$\|f_{\theta_0}\|_{L^2(\mathcal{D})} = \left( \sum_{i \in \mathcal{I}} \beta_i^2(\theta_0) \right)^{1/2} \le C_2 \alpha, \qquad \text{(upper bound on energy at } \theta_0\text{)}$$

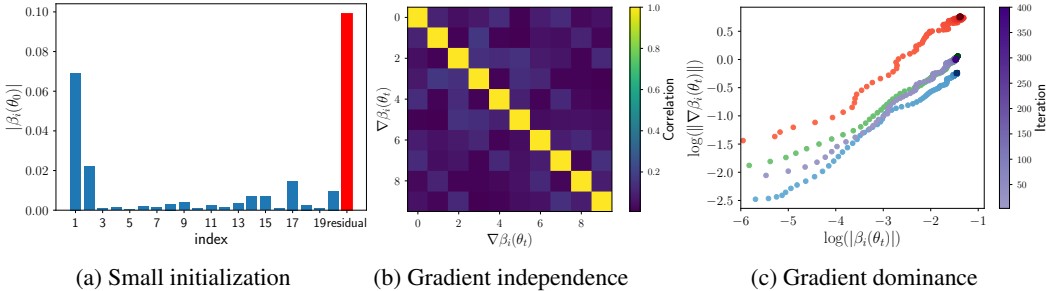

(a) Small initialization     (b) Gradient independence     (c) Gradient dominance

Figure 2: The conditions of Theorem 1 are approximately satisfied for LARS on a 2-layer CNN with MNIST dataset. Here, the coefficients are obtained by projecting the solution trajectory onto the eigenvectors of the conjugate kernel after training. (a) The top-20 basis coefficients at a small random initial point along with residual energy on the remaining coefficients. (b) The maximum value of $|\cos(\nabla \beta_i(\boldsymbol{\theta}), \nabla \beta_j(\boldsymbol{\theta}))|$ for $1 \leq i < j \leq 10$ along the solution trajectory. (c) The scaling of $\|\nabla \beta_i(\boldsymbol{\theta}_t)\|$ with respect to $|\beta_i(\boldsymbol{\theta}_t)|$ for the top-4 coefficients.

*for $C_1, C_2 > 0$ and $\alpha \lesssim \beta_k^\star$. Moreover, suppose that the orthogonal function basis is finite, i.e., $|\mathcal{I}| = d$ for some finite $d$, and the gradients of the coefficients satisfy the following conditions for every $0 \leq t \leq T$:*

$$\langle \nabla \beta_i(\boldsymbol{\theta}_t), \nabla \beta_j(\boldsymbol{\theta}_t) \rangle = 0 \qquad \text{for all } i \neq j, \qquad \text{(gradient independence)}$$

$$\|\nabla \beta_i(\boldsymbol{\theta}_t)\| \geq C \, |\beta_i(\boldsymbol{\theta}_t)|^\gamma \qquad \text{for all } i \in \mathcal{S}, \qquad \text{(gradient dominance)}$$

*for $C > 0$ and $1/2 \leq \gamma \leq 1$. Then, GD with step-size $\eta \lesssim \frac{\alpha^{2\gamma}}{\sqrt{d}C^2 L_H L_g^2 L_f^2} \beta_k^{\star 2\gamma} \log^{-1}\left(\frac{d\beta_k^\star}{C_1 \alpha}\right)$ satisfies:*

- *If $\gamma = \frac{1}{2}$, then within $T = \mathcal{O}\left(\frac{1}{C^2 \eta \beta_k^\star} \log\left(\frac{\beta_k^\star}{C_1 \alpha}\right)\right)$ iterations, we have $\|f_{\boldsymbol{\theta}_T} - f_{\boldsymbol{\theta}^\star}\|_{L^2(\mathcal{D})} \lesssim \alpha$.*

- *If $\frac{1}{2} < \gamma \leq 1$, then within $T = \mathcal{O}\left(\frac{1}{C^2 \eta \beta_k^\star \alpha^{2\gamma-1}}\right)$ iterations, we have $\|f_{\boldsymbol{\theta}_T} - f_{\boldsymbol{\theta}^\star}\|_{L^2(\mathcal{D})} \lesssim \alpha$.*

Theorem 1 shows that, under certain conditions on the basis coefficients and their gradients, GD with constant step-size converges to a model that is at most $\alpha$-away from the true model. In particular, to achieve an $\epsilon$-accurate solution for any $\epsilon > 0$, GD requires $\mathcal{O}((1/\epsilon) \log(1/\epsilon))$ iterations for $\gamma = 1/2$, and $\mathcal{O}(1/\epsilon^{2\gamma})$ iterations for $1/2 < \gamma \leq 1$ (ignoring the dependency on other problem-specific parameters). Due to its generality, our theorem inevitably relies on a small step-size and leads to a conservative convergence rate for GD. Later, we will show how our proposed approach can be tailored to specific learning problems to achieve better convergence rates in each setting.

**How realistic are the assumptions of Theorem 1?** A natural question arises as to whether the conditions for Theorem 1 are realistic. We start with the conditions on the initial point. Intuitively, these assumptions entail that a non-negligible fraction of the energy is carried by the signal at the initial point. We note that these assumptions are mild and expected to hold in practice. For instance, We will show in Section 3 that, depending on the learning task, they are guaranteed to hold with fixed, random, or spectral initialization.[3] We have also empirically verified that these conditions are satisfied for DNNs with random or default initialization. For instance, Figure 2a illustrates the top-20 basis coefficients at $\boldsymbol{\theta}_0$ for LARS with random initialization on a realistic CNN. It can be seen that a non-negligible fraction of the energy at the initial point is carried by the first few coefficients.

The conditions on the coefficient gradients are indeed harder to satisfy; as will be shown later, the existence of an ideal orthonormal basis may not be guaranteed even for linear NNs. Nonetheless, we have empirically verified that, with an appropriate choice of the orthonormal basis, the gradients of the coefficients remain *approximately independent* throughout the solution trajectory. Figure 2b shows that, when the orthonormal basis is chosen as the eigenvectors of the conjugate kernel after training, the maximum value of $|\cos(\nabla \beta_i(\boldsymbol{\theta}_t), \nabla \beta_j(\boldsymbol{\theta}_t))|$ remains small throughout the solution trajectory. Finally, we turn to the gradient dominance condition. Intuitively, this condition entails

---

[3]If $\boldsymbol{\theta}_0$ is selected from an isotropic Gaussian distribution, then $C_1$ and $C_2$ may scale with $k$ and $d$. However, to streamline the presentation, we keep this dependency implicit.

that the gradient of each signal scales with its norm. We prove that this condition is guaranteed to hold for kernel regression, symmetric matrix factorization, and symmetric tensor decomposition. Moreover, we have empirically verified that the gradient dominance holds across different DNN architectures. Figure 2c shows that this condition is indeed satisfied for the top-4 basis coefficients of the solution trajectory (other signal coefficients behave similarly). We also note that our theoretical result on GD may not naturally extend to LARS. Nonetheless, our extensive simulations suggest that our proposed analysis can be extended to other stochastic and adaptive variants of GD (see Section 3.4 and Appendix B); a rigorous verification of this conjecture is left as future work.

## 3 APPLICATIONS

In this section, we show how our proposed basis function decomposition can be used to study the performance of GD in different learning tasks, from simple kernel regression to complex DNNs. We start with the classical kernel regression, for which GD is known to converge linearly Karimi et al. (2016). Our purpose is to revisit GD through the lens of basis function decomposition, where there is a natural and simple choice for the basis functions. Next, we apply our approach to two important learning problems, namely symmetric matrix factorization and orthogonal symmetric tensor decomposition. In particular, we show how our proposed approach can be used to improve the convergence of GD for the symmetric matrix factorization and leads to a completely new convergence result for the orthogonal symmetric tensor decomposition. Finally, through extensive experiments, we showcase the promise of our proposed basis function decomposition on realistic DNNs.

### 3.1 KERNEL REGRESSION

In kernel regression (KR), the goal is to fit a regression model $f_{\boldsymbol{\theta}}(\boldsymbol{x}) = \sum_{i=1}^{d} \theta_i \phi_i(\boldsymbol{x})$ from the function class $\mathcal{F}_{\Theta} = \{f_{\boldsymbol{\theta}}(\boldsymbol{x}) : \boldsymbol{\theta} \in \mathbb{R}^d\}$ to observation $y$, where $\{\phi_i(\boldsymbol{x})\}_{i=1}^{d}$ are some known kernel functions. Examples of KR are linear regression, polynomial regression (including those described in Example 1), and neural tangent kernel (NTK) (Jacot et al., 2018). Without loss of generality, we may assume that the kernel functions $\{\phi_i(\boldsymbol{x})\}_{i=1}^{d}$ are orthonormal.[4] Under this assumption, the basis coefficients can be defined as $\beta_i(\boldsymbol{\theta}) = \theta_i$ and the expected loss can be written as

$$\mathcal{L}(\boldsymbol{\theta}_t) = \frac{1}{2}\mathbb{E}\left[(f_{\boldsymbol{\theta}_t}(\boldsymbol{x}) - f_{\boldsymbol{\theta}^\star}(\boldsymbol{x}))^2\right] = \frac{1}{2}\|\boldsymbol{\theta} - \boldsymbol{\theta}^\star\|^2 = \frac{1}{2}\sum_{i=1}^{d}(\beta_i(\boldsymbol{\theta}_t) - \theta_i^\star)^2. \tag{5}$$

Moreover, the coefficients satisfy the gradient independence condition. Therefore, an adaptation of Proposition 1 reveals that the dynamics of the basis coefficients are independent of each other.

**Proposition 2** (dynamics of $\beta_i(\boldsymbol{\theta}_t)$)**.** *Consider GD with a step-size that satisfies $0 < \eta < 1$. Then,*

- *for $i \in \mathcal{S}$, we have $\beta_i(\boldsymbol{\theta}_t) = \beta_i^\star - (1-\eta)^t(\beta_i^\star - \beta_i(\boldsymbol{\theta}_0))$,*

- *for $i \notin \mathcal{S}$, we have $\beta_i(\boldsymbol{\theta}_t) = (1-\eta)^t\beta_i(\boldsymbol{\theta}_0)$.*

Without loss of generality, we assume that $0 < \theta_k^\star \leq \cdots \leq \theta_1^\star \leq 1$ and $\|\boldsymbol{\theta}_0\|_\infty \leq \alpha$. Then, given Proposition 2, we have $|\beta_i(\boldsymbol{\theta}_t)| \leq 2 + \alpha$ for every $1 \leq i \leq k$. Therefore, the gradient dominance is satisfied with parameters $(C, \gamma) = (1/\sqrt{2+\alpha}, 1/2)$. Since both gradient independence and gradient dominance are satisfied, the convergence of GD can be established with an appropriate initial point.

**Theorem 2.** *Suppose that $\boldsymbol{\theta}_0 = \alpha\mathbf{1}$, where $\alpha \lesssim k|\theta_k^\star|/d$. Then, within $T \lesssim (1/\eta)\log(k|\theta_1^\star|/\alpha)$ iterations, GD with step-size $0 < \eta < 1$ satisfies $\|\boldsymbol{\theta}_T - \boldsymbol{\theta}^\star\| \lesssim \alpha$.*

Theorem 2 reveals that GD with large step-size and small initial point converges linearly to an $\epsilon$-accurate solution, provided that the initialization scale is chosen as $\alpha = \epsilon$. This is indeed better than our result on the convergence of GD for general models in Theorem 1.

---

[4]Suppose that $\{\phi_i(\boldsymbol{x})\}_{i=1}^{d}$ are not orthonormal. Let $\{\widetilde{\phi}_i(\boldsymbol{x})\}_{i\in\mathcal{I}}$ be any orthonormal basis for $\mathcal{F}_{\Theta}$. Then, there exists a matrix $A$ such that $\phi_i(\boldsymbol{x}) = \sum_j A_{ij}\widetilde{\phi}_j(\boldsymbol{x})$ for every $1 \leq i \leq d$. Therefore, upon defining $\widetilde{\boldsymbol{\theta}} = \boldsymbol{\theta}^\top A$, one can write $f_{\widetilde{\boldsymbol{\theta}}^\star}(\boldsymbol{x}) = \sum_{i\in\mathcal{I}}\widetilde{\theta}_i^\star\widetilde{\phi}_i(\boldsymbol{x})$ which has the same form as the regression model.

## 3.2 SYMMETRIC MATRIX FACTORIZATION

In symmetric matrix factorization (SMF), the goal is to learn a model $f_{\boldsymbol{U}^\star}(\mathbf{X}) = \langle \boldsymbol{U}^\star \boldsymbol{U}^{\star\top}, \mathbf{X} \rangle$ with a low-rank matrix $\boldsymbol{U}^\star \in \mathbb{R}^{d \times r}$, where we assume that each element of $X \sim \mathcal{D}$ is iid with $\mathbb{E}[X_{ij}] = 0$ and $\mathbb{E}[X_{ij}^2] = 1$. Examples of SMF are matrix sensing Li et al. (2018) and completion Ge et al. (2016). Given the eigenvectors $\{\boldsymbol{z}_1, \ldots, \boldsymbol{z}_d\}$ of $\boldsymbol{U}^\star \boldsymbol{U}^{\star\top}$ and a function class $\mathcal{F}_\Theta = \{f_{\boldsymbol{U}}(\mathbf{X}) : \boldsymbol{U} \in \mathbb{R}^{d \times r'}\}$ with $r' \geq r$, it was shown in Example 2 that the functions $\phi_{ij}(\mathbf{X}) = \langle \boldsymbol{z}_i \boldsymbol{z}_j^\top, \mathbf{X} \rangle$ define a valid orthogonal basis for $\mathcal{F}_\Theta$ with coefficients $\beta_{ij}(\boldsymbol{U}) = \langle \boldsymbol{z}_i \boldsymbol{z}_j^\top, \boldsymbol{U}\boldsymbol{U}^\top \rangle$. Therefore, we have

$$\mathcal{L}(\boldsymbol{U}_t) = \frac{1}{4}\mathbb{E}\left[(f_{\boldsymbol{U}_t}(\mathbf{X}) - f_{\boldsymbol{U}^\star}(\mathbf{X}))^2\right] = \frac{1}{4}\mathbb{E}\left[\left(\left\langle \boldsymbol{U}_t\boldsymbol{U}_t^\top - \boldsymbol{U}^\star\boldsymbol{U}^{\star\top}, \mathbf{X}\right\rangle\right)^2\right] = \frac{1}{4}\sum_{i,j=1}^{r'}(\beta_{ij}(\boldsymbol{U}_t) - \beta_{ij}^\star)^2.$$

Here, the true basis coefficients are defined as $\beta_{ii}^\star = \sigma_i$ for $i \leq r$, and $\beta_{ij}^\star = 0$ otherwise. Moreover, one can write $\|\nabla\beta_{ii}(\boldsymbol{U})\|_F = 2\|\boldsymbol{z}_i\boldsymbol{z}_i^\top\boldsymbol{U}\|_F = 2\sqrt{\beta_{ii}(\boldsymbol{U})}$. Therefore, gradient dominance holds with parameters $(C, \gamma) = (2, 1/2)$. However, gradient independence *does not* hold for this choice of function basis: given any pair $(i,j)$ and $(i,k)$ with $j \neq k$, we have $\langle \nabla\beta_{ij}(\boldsymbol{U}), \nabla\beta_{ik}(\boldsymbol{U})\rangle = \langle \boldsymbol{z}_j\boldsymbol{z}_k^\top, \boldsymbol{U}\boldsymbol{U}^\top\rangle$ which may not be zero. Despite the absence of gradient independence, our next proposition characterizes the dynamic of $\beta_{ij}(\boldsymbol{U}_t)$ via a finer control over the coefficient gradients.

**Proposition 3.** *Suppose that* $\gamma := \min_{1 \leq i \leq r}\{\sigma_i - \sigma_{i+1}\} > 0$. *Let* $\boldsymbol{U}_0 = \alpha\boldsymbol{B}$, *where* $\alpha \lesssim \min\left\{(\eta\sigma_r^2)^{\Omega(\sigma_1/\gamma)}, (\sigma_r/d)^{\Omega(\sigma_1/\gamma)}, (\kappa\log^2(d))^{-\Omega(1/(\eta\sigma_r))}\right\}$ *and the entries of* $\boldsymbol{B}$ *are independently drawn from a standard normal distribution. Suppose that the step-size for GD satisfies* $\eta \lesssim 1/\sigma_1$. *Then, with probability of at least* $1 - \exp(-\Omega(r'))$:

- *For* $1 \leq i \leq r$, *we have* $0.99\sigma_i \leq \beta_{ii}(\boldsymbol{U}_t) \leq \sigma_i$ *within* $\mathcal{O}\left((1/(\eta\sigma_i))\log(\sigma_i/\alpha)\right)$ *iterations.*

- *For* $t \geq 0$ *and* $i \neq j$ *or* $i, j > r$, *we have* $|\beta_{ij}(\boldsymbol{U}_t)| \lesssim \mathrm{poly}(\alpha)$.

Proposition 3 shows that GD with small random initialization learns larger eigenvalues before the smaller ones, which is commonly referred to as *incremental learning*. Incremental learning for SMF has been recently studied for gradient flow (Arora et al., 2019a; Li et al., 2020), as well as GD with identical initialization for the special case $r' = d$ Chou et al. (2020). To the best of our knowledge, Proposition 3 is the first result that provides a full characterization of the incremental learning phenomenon for GD with random initialization on SMF.

**Theorem 3.** *Suppose that the conditions of Proposition 3 are satisfied. Then, with probability of at least* $1 - \exp(-\Omega(r'))$ *and within* $T \lesssim (1/(\eta\sigma_r))\log(\sigma_r/\alpha)$ *iterations, GD satisfies*

$$\left\|\boldsymbol{U}_T\boldsymbol{U}_T^\top - \boldsymbol{M}^\star\right\|_F \lesssim r'\log(d)d\alpha^2. \tag{6}$$

It has been shown in (Stöger & Soltanolkotabi, 2021, Thereom 3.3) that GD with small random initialization satisfies $\left\|\boldsymbol{U}_T\boldsymbol{U}_T^\top - \boldsymbol{M}^\star\right\|_F \lesssim \left(d^2/r'^{15/16}\right)\alpha^{21/16}$ within the same number of iterations. Theorem 3 improves the dependency of the final error on the initialization scale $\alpha$.

## 3.3 ORTHOGONAL SYMMETRIC TENSOR DECOMPOSITION

We use our approach to provide a new convergence guarantee for GD on the orthogonal symmetric tensor decomposition (OSTD). In OSTD, the goal is to learn $f_{\boldsymbol{U}^\star}(\mathbf{X}) = \langle \mathbf{T}_{\boldsymbol{U}^\star}, \mathbf{X}\rangle$, where $\boldsymbol{U}^\star = [\boldsymbol{u}_1^\star, \ldots, \boldsymbol{u}_r^\star] \in \mathbb{R}^{d \times r}$ and $\mathbf{T}_{\boldsymbol{U}^\star} = \sum_{i=1}^r \boldsymbol{u}_i^{\star\otimes l} = \sum_{i=1}^d \sigma_i \boldsymbol{z}_i^{\otimes l}$ is a symmetric tensor with order $l$ and rank $r$. Here, $\sigma_1 \geq \cdots \geq \sigma_d$ are tensor eigenvalues with $\sigma_{r+1} = \cdots = \sigma_d = 0$, and $\boldsymbol{z}_1, \ldots, \boldsymbol{z}_d$ are the corresponding tensor eigenvectors. The notation $\boldsymbol{u}^{\otimes l}$ refers to the $l$-time outer product of $\boldsymbol{u}$. We assume that $\mathbf{X} \sim \mathcal{D}$ is an $l$-order tensor whose elements are iid with zero mean and unit variance. Examples of OSTD are tensor regression (Tong et al., 2022) and completion (Liu et al., 2012).

When the rank of $\mathbf{T}_{\boldsymbol{U}^\star}$ is unknown, it must be overestimated. Even when the rank is known, its overestimation can improve the convergence of gradient-based algorithms (Wang et al., 2020). This leads to an *overparameterized* model $f_{\boldsymbol{U}}(\mathbf{X}) = \langle \mathbf{T}_{\boldsymbol{U}}, \mathbf{X}\rangle$, where $\mathbf{T}_{\boldsymbol{U}} = \sum_{i=1}^{r'} \boldsymbol{u}_i^{\otimes l}$ with an overestimated rank $r' \geq r$. Accordingly, the function class is defined as $\mathcal{F}_\Theta = \left\{f_{\boldsymbol{U}}(\mathbf{X}) : \boldsymbol{U} = [\boldsymbol{u}_1, \cdots, \boldsymbol{u}_{r'}] \in \mathbb{R}^{d \times r'}\right\}$.

Upon defining a multi-index $\Lambda = (j_1, \cdots, j_l)$, the functions $\phi_\Lambda(\mathbf{X}) = \left\langle \otimes_{k=1}^l \boldsymbol{z}_{j_k}, \mathbf{X} \right\rangle$ for $1 \leq j_1, \ldots, j_l \leq d$ form an orthonormal basis for $\mathcal{F}_\Theta$ with basis coefficients defined as

$$\beta_\Lambda(\boldsymbol{U}) = \mathbb{E}\left[\langle \mathbf{T}_{\boldsymbol{U}}, \mathbf{X} \rangle \left\langle \otimes_{k=1}^l \boldsymbol{z}_{j_k}, \mathbf{X} \right\rangle\right] = \sum_{i=1}^{r'} \left\langle \boldsymbol{u}_i^{\otimes l}, \otimes_{k=1}^l \boldsymbol{z}_{j_k} \right\rangle = \sum_{i=1}^{r'} \prod_{k=1}^l \langle \boldsymbol{u}_i, \boldsymbol{z}_{j_k} \rangle,$$

and the expected loss can be written as

$$\mathcal{L}(\boldsymbol{U}) = \frac{1}{2}\mathbb{E}\left[(f_{\boldsymbol{U}}(\mathbf{X}) - f_{\boldsymbol{U}^\star}(\mathbf{X}))^2\right] = \frac{1}{2}\left\|\sum_{i=1}^{r'} \boldsymbol{u}_i^{\otimes l} - \sum_{i=1}^r \sigma_i \boldsymbol{z}_i^{\otimes l}\right\|_F^2 = \frac{1}{2}\sum_\Lambda \left(\beta_\Lambda(\boldsymbol{U}) - \beta_\Lambda^\star\right)^2,$$

where the true basis coefficients are $\beta_{\Lambda_i}^\star = \sigma_i$ for $\Lambda_i = (i, \ldots, i), 1 \leq i \leq r$, and $\beta_\Lambda^\star = 0$ otherwise. Unlike KR and SMF, neither gradient independence nor gradient dominance are satisfied for OSTD with a random or equal initialization. However, we show that these conditions are *approximately* satisfied throughout the solution trajectory, provided that the initial point is nearly aligned with the eigenvectors $\boldsymbol{z}_1, \ldots, \boldsymbol{z}_{r'}$; in other words, $\cos(\boldsymbol{u}_i(0), \boldsymbol{z}_i) \approx 1$ for every $1 \leq i \leq r'$.[5] Assuming that the initial point satisfies this alignment condition, we show that the *entire* solution trajectory remains aligned with these eigenvectors, i.e., $\cos(\boldsymbol{u}_i(t), \boldsymbol{z}_i) \approx 1$ for every $1 \leq i \leq r'$ and $1 \leq t \leq T$. Using this key result, we show that both gradient independence and gradient dominance are *approximately* satisfied throughout the solution trajectory. We briefly explain the intuition behind our approach for gradient dominance and defer our rigorous analysis for gradient independence to the appendix. Note that if $\cos(\boldsymbol{u}_i(t), \boldsymbol{z}_i) \approx 1$, then $\beta_{\Lambda_i}(\boldsymbol{U}_t) \approx \langle \boldsymbol{u}_i(t), \boldsymbol{z}_i \rangle^l$ and $\|\nabla\beta_{\Lambda_i}(\boldsymbol{U}_t)\|_F \approx \|\nabla_{\boldsymbol{u}_i}\beta_{\Lambda_i}(\boldsymbol{U}_t)\| \approx l \langle \boldsymbol{u}_i(t), \boldsymbol{z} \rangle^{l-1}$. Therefore, gradient dominance holds with parameters $(C, \gamma) = (l, (l-1)/l)$. We will make this intuition rigorous in Appendix G.

**Proposition 4.** *Suppose that the initial point $\boldsymbol{U}_0$ is chosen such that $\|\boldsymbol{u}_i(0)\| = \alpha^{1/l}$ and $\cos(\boldsymbol{u}_i(0), \boldsymbol{z}_i) \geq \sqrt{1-\gamma}$, for all $1 \leq i \leq r'$, where $\alpha \lesssim d^{-l^3}$ and $\gamma \lesssim (l\kappa)^{-l/(l-2)}$. Then, GD with step-size $\eta \lesssim 1/(l\sigma_1)$ satisfies:*

- *For $1 \leq i \leq r$, we have $0.99\sigma_i \leq \beta_{\Lambda_i}(\boldsymbol{U}_t) \leq 1.01\sigma_i$ within $\mathcal{O}\left((1/(\eta l\sigma_r))\alpha^{-\frac{l-2}{l}}\right)$ iterations.*

- *For $t \geq 0$ and $\Lambda \neq \Lambda_i$, we have $|\beta_\Lambda(\boldsymbol{U}_t)| = \mathrm{poly}(\alpha)$.*

Proposition 4 shows that, similar to SMF, GD learns the tensor eigenvalues incrementally. However, unlike SMF, we require a specific alignment for the initial point. We note that such initial point can be obtained in a pre-processing step via tensor power method within a number of iterations that is almost independent of $d$ (Anandkumar et al., 2017, Theorem 1). We believe that Proposition 4 can be extended to random initialization; we leave the rigorous verification of this conjecture to future work. Equipped with this proposition, we next establish the convergence of GD on OSTD.

**Theorem 4.** *Suppose that the conditions of Proposition 4 are satisfied. Then, within $T \lesssim (1/(\eta l\sigma_r))\alpha^{-(l-2)/l}$ iterations, GD satisfies*

$$\|\mathbf{T}_{\boldsymbol{U}_T} - \mathbf{T}_{\boldsymbol{U}^\star}\|_F^2 \lesssim r d^l \gamma \sigma_1^{\frac{l-1}{l}} \alpha^{\frac{1}{l}}. \tag{7}$$

Theorem 4 shows that, with appropriate choices of $\eta$ and $\alpha$, GD converges to a solution that satisfies $\|\mathbf{T}_{\boldsymbol{U}_T} - \mathbf{T}_{\boldsymbol{U}^\star}\|_F^2 \leq \epsilon$ within $\mathcal{O}(d^{l(l-2)}/\epsilon^{l-2})$ iterations. To the best of our knowledge, this is the first result establishing the convergence of GD with a large step-size on OSTD.

### 3.4 EMPIRICAL VERIFICATION ON NEURAL NETWORKS

In this section, we numerically show that the conjugate kernel after training (A-CK) can be used as a valid orthogonal basis for DNNs to capture the monotonicity of the solution trajectory of different optimizers on image classification tasks. To ensure consistency with our general framework, we use $\ell_2$-loss, which is shown to have a comparable performance with the commonly-used cross-entropy loss Hui & Belkin (2020). In Appendix B, we extend our simulations to cross-entropy loss. The conjugate kernel (CK) is a method for analyzing the generalization performance of DNNs that uses

---

[5]We use the notations $\boldsymbol{u}_t$ or $\boldsymbol{u}(t)$ interchangeably to denote the solution at iteration $t$.

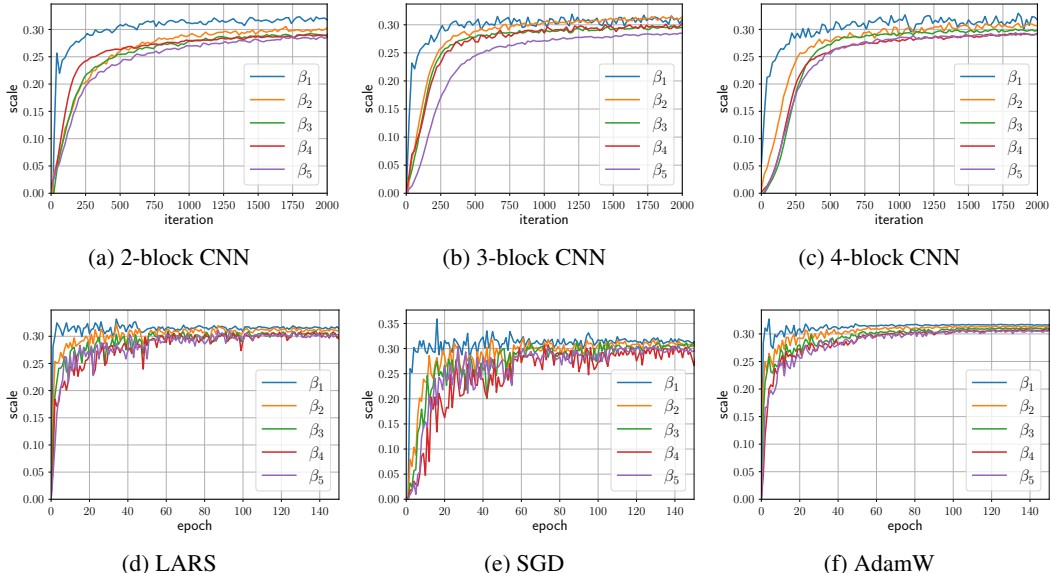

(a) 2-block CNN      (b) 3-block CNN      (c) 4-block CNN

(d) LARS      (e) SGD      (f) AdamW

Figure 3: (First row) the projected trajectory of LARS on CNNs with MNIST dataset. The test accuracies for 2-Block, 3-Block, and 4-block CNN are $96.10\%, 96.48\%, 96.58\%$. (Second row) the projected trajectories of different optimizers on AlexNet with the CIFAR-10 dataset. The test accuracies for LARS, SGD, and AdamW are $90.54\%$, $91.03\%$, and $90.26\%$, respectively. We use the following settings for each optimizer: (d) LARS: learning rate of 2, Nesterov momentum of 0.9, and weight decay of $1 \times 10^{-4}$. (e) SGD: learning rate of 2 with "linear warm-up", Nesterov momentum of 0.9, weight decay of $1 \times 10^{-4}$. (f) AdamW: learning rate of 0.01.

the second to last layer (the layer before the last linear layer) at the *initial point* as the feature map (Daniely et al., 2016; Fan & Wang, 2020; Hu & Huang, 2021). Recently, Long (2021) shows that A-CK, a variant of CK that is evaluated at the *last* epoch, better explains the generalization properties of realistic DNNs. Surprisingly, we find that A-CK can be used not only to characterize the generalization performance but also to capture the underlining solution trajectory of different gradient-based algorithms.

To formalize the idea, note that any neural network whose last layer is linear can be characterized as $f_{\boldsymbol{\theta}}(\boldsymbol{x}) = \boldsymbol{W}\psi(\boldsymbol{x})$, where $\boldsymbol{x} \in \mathbb{R}^d$ is the input drawn from the distribution $\mathcal{D}$, $\psi(\boldsymbol{x}) \in \mathbb{R}^m$ is the feature map with number of features $m$, and $\boldsymbol{W} \in \mathbb{R}^{k \times m}$ is the last linear layer with $k$ referring to the number of classes. We denote the trained model, i.e., the model in the last epoch, by $f_{\boldsymbol{\theta}_{\infty}}(\boldsymbol{x}) = \boldsymbol{W}_{\infty}\psi_{\infty}(\boldsymbol{x})$. To form an orthogonal basis, we use SVD to obtain a series of basis functions $\phi_i(\boldsymbol{x}) = \boldsymbol{W}_{\infty,i}\psi_{\infty}(\boldsymbol{x})$ that satisfy $\mathbb{E}_{\boldsymbol{x}\sim\mathcal{D}}[\|\phi_i(\boldsymbol{x})\|^2] = 1$ and $\mathbb{E}_{\boldsymbol{x}\sim\mathcal{D}}[\langle\phi_i(\boldsymbol{x}), \phi_j(\boldsymbol{x})\rangle] = \delta_{ij}$ where $\delta_{ij}$ is the delta function. Hence, the coefficient $\beta_i(\boldsymbol{\theta}_t)$ at each epoch $t$ can be derived as $\beta_i(\boldsymbol{\theta}_t) = \mathbb{E}_{\boldsymbol{x}\sim\mathcal{D}}[\langle f_{\boldsymbol{\theta}_t}(\boldsymbol{x}), \phi_j(\boldsymbol{x})\rangle]$, where the expectation is estimated by its sample mean on the test set. More details on our implementation can be found in Appendix B.

**Performance on convolutional neural networks:** We use LARS to train CNNs with varying depths on MNIST dataset. These networks are trained such that their test accuracies are above $96\%$. Figures 3a-3c illustrate the evolution of the top-5 basis coefficients after projecting LARS onto the orthonormal basis obtained from A-CK. It can be observed that the basis coefficients are consistently monotonic across different depths, elucidating the generality of our proposed basis function decomposition. In the appendix, we discuss the connection between the convergence of the basis functions and the test accuracy for different architectures and loss functions.

**Performance with different optimizers:** The monotonic behavior of the projected solution trajectory is also observed across different optimizers. Figures 3d-3f show the solution trajectories of three optimizers, namely LARS, SGD, and AdamW, on AlexNet with the CIFAR-10 dataset. It can be seen that all three optimizers have a monotonic trend after projecting onto the orthonormal basis obtained from A-CK. Although our theoretical results only hold for GD, our simulations highlight the strength of the proposed basis function decomposition in capturing the behavior of other gradient-based algorithms on DNN.

ACKNOWLEDGEMENTS

We thank Richard Y. Zhang and Tiffany Wu for helpful feedback. We would also like to thank Ruiqi Gao and Chenwei Wu for their insightful discussions. This research is supported, in part, by NSF Award DMS-2152776, ONR Award N00014-22-1-2127, MICDE Catalyst Grant, MIDAS PODS grant and Startup Funding from the University of Michigan.

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

CONTENTS

## A    DISCUSSION AND FUTURE DIRECTION

**Extension to empirical loss and implication on generalization.**    The most significant future direction is to extend our analysis of expected loss to the finite sample regime. It has been shown that incremental learning can drive generalization (Gissin et al., 2019). Hence, our proposed framework is likely to explain the puzzling generalization ability of overparameterized machine learning models. For such an extension, the main technical difficulty is the relaxation of gradient independence. In the finite-sample regime, gradient independence does not hold exactly due to the randomness of samples. Therefore, an alternative approach would be to establish gradient independence approximately and with high probability. We have successfully applied our framework to matrix factorization and tensor decomposition, where gradient independence only holds approximately. Therefore, we believe that more general guarantees on the finite sample regime are not out of reach.

**Extension to other optimization algorithms.**    In this paper, we mainly focused on GD as a representative of various optimization algorithms. Nonetheless, we believe that our analysis can be adapted to investigate other local-search optimization algorithms, such as GD with momentum (Nesterov, 1983), SGD (Robbins & Monro, 1951), and Adam (Kingma & Ba, 2014). Overall, our approach can offer a unified framework for examining the implicit bias and incremental learning phenomena for different local-search algorithms.

## B    ADDITIONAL EXPERIMENTS

In this section, we provide more details on our simulation and further explore the empirical strength of the proposed basis function decomposition on different datasets, optimizers, loss functions, and batch sizes; see Table 1 for a summary of our simulations in this section.

| Architectures | CNN | AlexNet | VGG11 | ResNet-18 | ResNet-34 | ResNet-50 | ViT |
|---|---|---|---|---|---|---|---|
| Datasets | MNIST | CIFAR-10 | CIFAR-100 | | | | |
| Optimizers | SGD | AdamW | LARS | | | | |
| Losses | $\ell_2$-loss | CE loss | | | | | |

Table 1: The summary of our experiments.

### B.1    NUMERICAL VERIFICATION OF OUR THEORETICAL RESULTS

In this section, we provide experimental evidence to support our theoretical results on kernel regression (KR), symmetric matrix factorization (SMF), and orthogonal symmetric tensor decomposition (OSTD). The results are presented in Figure 4.

**Kernel regression.**    We randomly generate 20 orthonormal kernel functions. The true model is comprised of 4 signal terms with basis coefficients $10, 5, 3, 1$. Figure 4a shows the trajectories of the top-4 basis coefficients of GD with initial point $\boldsymbol{\theta}_0 = 5 \times 10^{-7} \times \mathbf{1}$ and step-size $\eta = 0.4$. It can be seen that GD learns different coefficients at the same rate, which is in line with Proposition 2.

**Symmetric matrix factorization.**    In this simulation, we aim to recover a rank-4 matrix $\boldsymbol{M}^\star = \boldsymbol{V}\boldsymbol{\Sigma}\boldsymbol{V}^\top \in \mathbb{R}^{20 \times 20}$. In particular, we assume that $\boldsymbol{V} \in \mathbb{R}^{20 \times 4}$ is a randomly generated orthonormal matrix and $\boldsymbol{\Sigma} = \mathrm{Diag}\{10, 5, 3, 1\}$. We consider a fully over-parameterized model where $\boldsymbol{U} \in \mathbb{R}^{20 \times 20}$ (i.e., $r' = 20$). Figure 4b illustrates the incremental learning phenomenon that was proved in Proposition 3 for GD with small Gaussian initialization $\boldsymbol{U}_{ij} \overset{i.i.d.}{\sim} \mathcal{N}(0, \alpha^2), \alpha = 5 \times 10^{-7}$ and step-size $\eta = 0.04$.

**Orthogonal symmetric tensor decomposition.**    Finally, we present our simulations for OSTD. We aim to recover a rank-4 symmetric tensor of the form $\mathbf{T}^\star = \sum_{i=1}^{4} \sigma_i \boldsymbol{z}_i^{\otimes 4}$ where $\sigma_i$ are the nonzero eigenvalues with values $\{10, 5, 3, 1\}$ and $\boldsymbol{z}_i \in \mathbb{R}^{10}$ are the corresponding eigenvectors. We again consider a fully over-parameterized model with $r' = 10$. Figure 4c shows the incremental learning phenomenon for GD with an aligned initial point that satisfies $\cos(\boldsymbol{u}_i(0), \boldsymbol{z}_i) \geq 0.9983, 1 \leq i \leq r'$ and step-size $\eta = 0.001$.

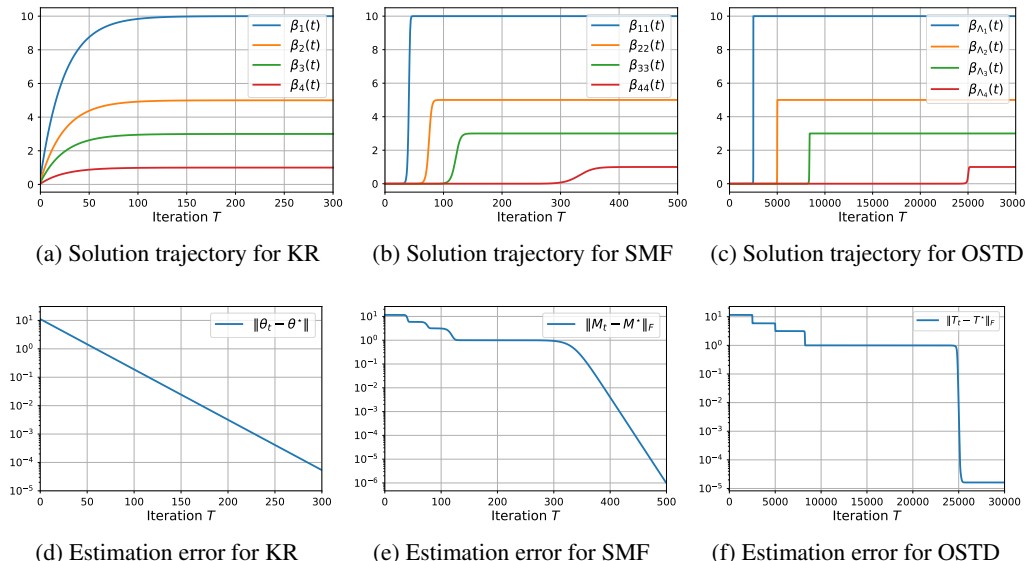

(a) Solution trajectory for KR  (b) Solution trajectory for SMF  (c) Solution trajectory for OSTD

(d) Estimation error for KR  (e) Estimation error for SMF  (f) Estimation error for OSTD

Figure 4: Experimental verification to support Theorems 2, 3, and 4 in Section 3. The first row shows the projected trajectories of GD onto the specific basis functions we defined for each problem. The second row shows the estimation error.

## B.2 Derivation of Basis Functions for DNNs

In this section, we provide more details on how we evaluate our proposed orthogonal basis induced by A-CK and calculate the corresponding coefficients $\beta_i(\boldsymbol{\theta}_t)$ for an arbitrary neural network. First, recall that any neural network whose last layer is linear can be characterized as $f_{\boldsymbol{\theta}}(\boldsymbol{x}) = \boldsymbol{W}\psi(\boldsymbol{x})$, where $\boldsymbol{x} \in \mathbb{R}^d$ is the input drawn from the distribution $\mathcal{D}$, $\psi(\boldsymbol{x}) \in \mathbb{R}^m$ is the feature map with number of features $m$, and $\boldsymbol{W} \in \mathbb{R}^{k \times m}$ is the last linear layer with $k$ referring to the number of classes. We denote the trained model, i.e., the model in the last epoch, by $f_{\boldsymbol{\theta}_\infty}(\boldsymbol{x}) = \boldsymbol{W}_\infty \psi_\infty(\boldsymbol{x})$. To form an orthogonal basis, we use SVD to obtain a series of basis functions $\phi_i(\boldsymbol{x}) = \boldsymbol{W}_{\infty,i}\psi_\infty(\boldsymbol{x})$ that satisfy $\mathbb{E}_{\boldsymbol{x}\sim\mathcal{D}}[\|\phi_i(\boldsymbol{x})\|^2] = 1$ and $\mathbb{E}_{\boldsymbol{x}\sim\mathcal{D}}[\langle\phi_i(\boldsymbol{x}), \phi_j(\boldsymbol{x})\rangle] = \delta_{ij}$ where $\delta_{ij}$ is the delta function. Hence, the coefficient $\beta_i(\boldsymbol{\theta}_t)$ at each epoch $t$ can be derived as $\beta_i(\boldsymbol{\theta}_t) = \mathbb{E}_{\boldsymbol{x}\sim\mathcal{D}}[\langle f_{\boldsymbol{\theta}_t}(\boldsymbol{x}), \phi_j(\boldsymbol{x})\rangle]$. In all of our implementation, we use the test dataset to approximate the population distribution.

**Step 1: Obtaining the orthogonal basis $\phi_i(\boldsymbol{x})$.**  We denote $\boldsymbol{\Psi} = [\psi(\boldsymbol{x}_1), \cdots, \psi(\boldsymbol{x}_N)] \in \mathbb{R}^{m \times N}$ as the feature matrix where $N$ is the number of the test data points. We write the SVD of $\boldsymbol{\Psi}$ as $\boldsymbol{\Psi} = \boldsymbol{U}\boldsymbol{\Sigma}\boldsymbol{V}^\top$. The right singular vectors collected in $\boldsymbol{V}$ can be used to define the desired orthogonal basis of $\{\psi_i(\boldsymbol{x})\}_{i=1}^m$. To this goal, we write the prediction matrix as $\boldsymbol{F} = \boldsymbol{W}\boldsymbol{\Psi} = \widetilde{\boldsymbol{W}}\widetilde{\boldsymbol{\Psi}}$ where $\widetilde{\boldsymbol{W}} = \boldsymbol{W}\boldsymbol{U}\boldsymbol{\Sigma}$ and $\widetilde{\boldsymbol{\Psi}} = \boldsymbol{V}^\top$. Our goal is to define a set of matrices $\boldsymbol{A}_i$ such that $\phi_i(\boldsymbol{x}) = \boldsymbol{A}_i\widetilde{\psi}_i(\boldsymbol{x})$ form a valid orthonormal basis for $\boldsymbol{F} = \boldsymbol{W}\boldsymbol{\Psi} = \widetilde{\boldsymbol{W}}\widetilde{\boldsymbol{\Psi}}$. Before designing such $\boldsymbol{A}_i$, first note that, due to the orthogonality of $\{\widetilde{\psi}_i(\boldsymbol{x})\}$, we have

$$\mathbb{E}\left[\|\phi_i(\boldsymbol{x})\|^2\right] = \|\boldsymbol{A}_i\|_F^2, \quad \mathbb{E}\left[\langle\phi_i(\boldsymbol{x}), \phi_j(\boldsymbol{x})\rangle\right] = \langle\boldsymbol{A}_i, \boldsymbol{A}_j\rangle. \qquad (8)$$

Therefore, it suffices to ensure that $\{\boldsymbol{A}_i\}$ are orthonormal. Consider the SVD of $\widetilde{\boldsymbol{W}}$ as $\widetilde{\boldsymbol{W}} = \sum_i \sigma_i\boldsymbol{u}_i\boldsymbol{v}_i^\top$. We define $\boldsymbol{A}_i = \boldsymbol{u}_i\boldsymbol{v}_i^\top$. Clearly, defined $\{\boldsymbol{A}_i\}$ are orthonormal. Moreover, it is easy to see that the basis coefficients (treated as the true basis coefficients) are exactly the singular values of $\widetilde{\boldsymbol{W}}$.

**Step 2: Obtaining the basis coefficients $\beta_i(\boldsymbol{\theta}_t)$.**  After obtaining the desired orthonormal basis $\phi_i(\boldsymbol{x})$, we can calculate the coefficient $\beta_i(\boldsymbol{\theta}_t)$ for each epoch. Given the linear layer $\boldsymbol{W}_t$ and the feature matrix $\boldsymbol{\Phi}_t$ at epoch $t$, we can obtain the coefficients for the signal terms by projecting the prediction matrix $\boldsymbol{F}_t = \boldsymbol{W}_t\boldsymbol{\Psi}_t$ onto $\widetilde{\boldsymbol{\Psi}} = \boldsymbol{V}^\top$. In particular, we write the prediction matrix as

$\mathcal{P}_{\widetilde{\psi}}\boldsymbol{F}_t = \mathcal{P}_{\widetilde{\psi}}\boldsymbol{W}_t\boldsymbol{\Psi}_t = \boldsymbol{W}_t\boldsymbol{\Psi}_t\boldsymbol{V}\boldsymbol{V}^\top = \widetilde{\boldsymbol{W}}_t\widetilde{\boldsymbol{\Psi}}$ where $\widetilde{\boldsymbol{W}}_t = \boldsymbol{W}_t\boldsymbol{\Psi}_t\boldsymbol{V}$. Hence, the basis coefficients can be easily calculated as $\beta_i(\boldsymbol{\theta}_t) = \left\langle \widetilde{\boldsymbol{W}}_t, \boldsymbol{u}_i\boldsymbol{v}_i^\top \right\rangle$.

## B.3 FURTHER DETAILS ON THE EXPERIMENTS

In this section, we provide more details on our experiments presented in the main body of the paper and compare them with other DNN architectures.

All of our experiments are implemented in `Python 3.9, Pytorch 1.12.1` environment and run through a local server SLURM using NVIDIA Tesla with V100-PCIE-16GB GPUs. We use an additional NNGeometry package for calculating batch gradient, and our implemention of ViT is adapted from `https://juliusruseckas.github.io/ml/cifar10-vit.html`. To ensure consistency with our theoretical results, we drop the last softmax operator and use the $\ell_2$-loss throughout this section. All of our training data are augmented by `RandomCrop` and `RandomHorizontalFlip`, and normalized by mean and standard deviation.

**Experimental details for Figure 1.** Here, we describe our implementation details for Figure 1, and present additional experiments on VGG-11, ResNet-34, and ResNet-50 with the CIFAR-10 dataset. The results can be seen in Figure 5. We use standard data augmentation for all architectures except for ViT. For ViT, we only use data normalization.

To obtain a stable A-CK, we trained the above models for 300 epochs. For ResNet-18, we used LARS with a learning rate of 0.5 and applied small initialization with $\alpha = 0.3$, i.e., we scale the default initial point by $\alpha = 0.3$. For ResNet-34 and ResNet-50, we choose the default learning rate and apply small initialization with $\alpha = 0.3$. For ViT, we use AdamW with a learning rate of 0.01. The remaining parameters are set to their default values.

**Experiments details for the first row of Figure 3.** Here we conduct experiments on MNIST dataset with different CNN architectures. The CNNs are composed of $l$ blocks of layers, followed by a single fully connected layer. A block of a CNN consists of a convolutional layer, an activation layer, and a pooling layer. In our experiments, we use ReLU activation and vary the depth of the network. For the first block, we used identity pooling. For the remaining blocks, we used max-pooling.

For 2-block CNN, we set the convolutional layer width to 256 and 64, respectively. For 3-block CNN, we set the convolutional layer width to 256, 128, and 64, respectively. And for 4-block CNN, we set the convolutional layer width to 256, 128, 128, and 64, respectively. To train these networks, we used LARS with the learning rate of 0.05. The remaining parameters are set to their default values. We run 20 epochs to calculate A-CK.

**Experiments details for second row of Figure 3.** We conduct experiments to compare the performance of different optimizers on the CIFAR-10 dataset. In particular, we use AlexNet to compare the performance of three optimizers, i.e., SGD, AdamW, LARS. For SGD, we set the base learning rate to be 2 with Nesterov momentum of 0.9 and weight decay of 0.0001, together with the "linear warm-up" technique.[6] For AdamW, we set the learning rate to 0.01 and keep the remaining parameters unchanged. For LARS, we set the learning rate to 2 with Nesterov momentum of 0.9 and weight decay of 0.0001. The remaining parameters are set to the default setting.

## B.4 EXPERIMENTS FOR CIFAR-100

In this section, we conduct experiments using the CIFAR-100 dataset which is larger than both CIFAR-10 and MNIST. Our simulations are run on AlexNet, VGG-11, ViT, ResNet-18, ResNet-34, and ResNet-50. In particular, we use the "loss scaling trick" (Hui & Belkin, 2020) defined as follows: consider the datapoint $(\boldsymbol{x}, \boldsymbol{y})$ where $\boldsymbol{x} \in \mathbb{R}^d$ is the input and $\boldsymbol{y} \in \mathbb{R}^k$ is a one-hot vector with 1 at

---

[6]In "linear warm-up", we linearly increase the learning rate in the first 5 epochs. More precisely, we set the initial learning rate to $1 \times 10^{-5}$ and linearly increase it to the selected learning rate in 5 epochs. After the first 5 epochs, the learning rate follows a regular decay scheme.

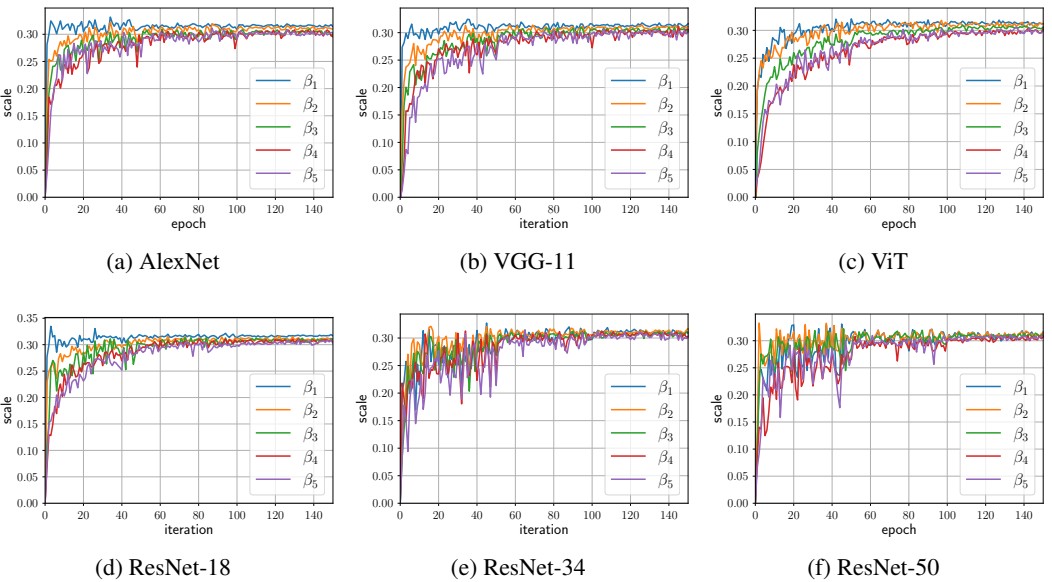

(a) AlexNet  (b) VGG-11  (c) ViT

(d) ResNet-18  (e) ResNet-34  (f) ResNet-50

Figure 5: Solution trajectories for different architectures trained on CIFAR-10. The test accuracies for Alexnet, VGG-11 and ViT are 90.54%, 91.11%, 77.99%, respectively. The test accuracies for ResNet-18, ResNet-34 and ResNet-50 are 88.31%, 93.93% and 94.06%, respectively.

position $i$. Then, the scaled $\ell_2$-loss is defined as

$$\ell_{2,\text{scaling}}(\boldsymbol{x}) = k \cdot (f_{\boldsymbol{\theta}}(\boldsymbol{x})[i] - M)^2 + \sum_{i' \neq i} (f_{\boldsymbol{\theta}}(\boldsymbol{x})[i'])^2, \tag{9}$$

for some constants $k, M > 0$. We set these parameters to $k = 1, M = 4$. For AlexNet and VGG-11, we use LARS with a base learning rate of $\eta = 1$. For ViT, we use AdamW with a base learning rate of $0.01$ and batch size of $256$. For ResNet architectures, we use SGD with a base learning rate of $0.3$ and batch size of $64$. We also add $5$ warm-up epochs for ResNets. All the remaining parameters for the above architectures are set to their default values. The results can be seen in Figure 6. Our experiments highlight a trade-off between the monotonicity of the projected solution trajectories and the test accuracy: in order to obtain a higher test accuracy, one typically needs to pick a larger learning rate, which in turn results in more sporadic behavior of the solution trajectories. Nonetheless, even with large learning, the basis coefficients remain relatively monotonic after the first few epochs and converge to meaningful values.

## B.5 Experiments for Different Losses

Next, we compare the projected solution trajectories on two loss functions, namely $\ell_2$-loss and cross-entropy (CE) loss. We use LARS to train AlexNet on the CIFAR-10 dataset with both $\ell_2$-loss and CE loss. In particular, we add the softmax operator before training the CE loss. For CE loss, we set the base learning rate of LARS to $1$. For $\ell_2$-loss, we use the base learning rate of $2$. The remaining parameters are set to their default values. The results can be seen in Figure 7. We observe that, similar to the $\ell_2$-loss, the solution trajectory of the CE loss behaves monotonically after projecting onto the orthogonal basis induced by A-CK. Inspired by these observations, another venue for future research would be to extend our framework to general loss functions. Interestingly, the convergence of the basis coefficients is much slower than those of the $\ell_2$-loss. This is despite the fact that CE loss can learn slightly faster than $\ell_2$-loss in terms of test accuracy as shown by Hui & Belkin (2020).

## B.6 Experiments for Different Batch Size

Next, we study the effect of different batch sizes on the solution trajectory. We train AlexNet on the CIFAR-10 dataset. When testing for different batch sizes, we follow the "linear scaling" rule, i.e.,

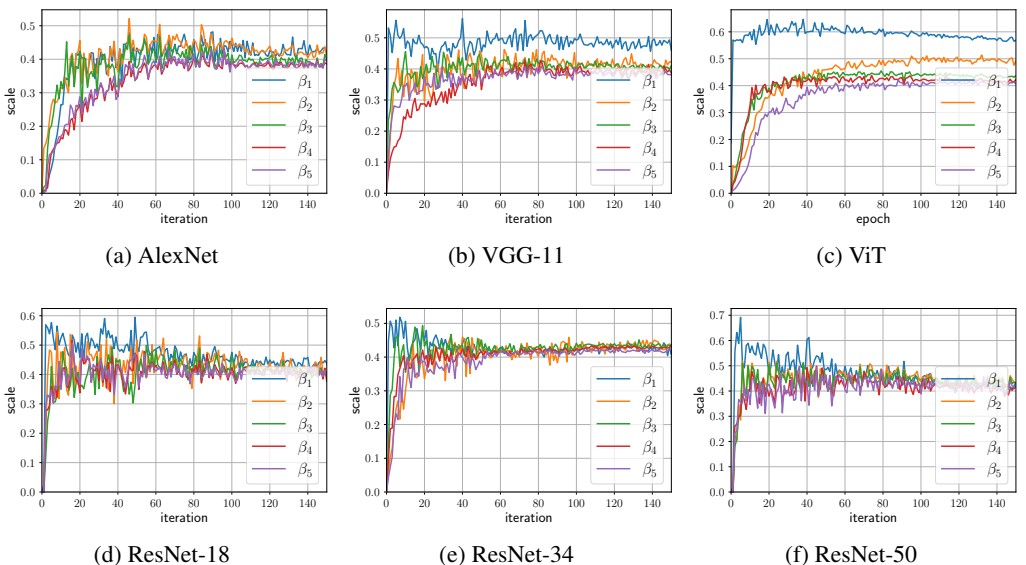

(a) AlexNet       (b) VGG-11       (c) ViT

(d) ResNet-18       (e) ResNet-34       (f) ResNet-50

Figure 6: Solution trajectories for different architectures trained on CIFAR-100. The test accuracies for AlexNet, VGG-11, and ViT are 61.20%, 50.64%, and 50.41%, respectively. The test accuracies for ResNet-18, ResNet-34 and ResNet-50 are 78.11%, 74.26% and 80.05%, respectively.

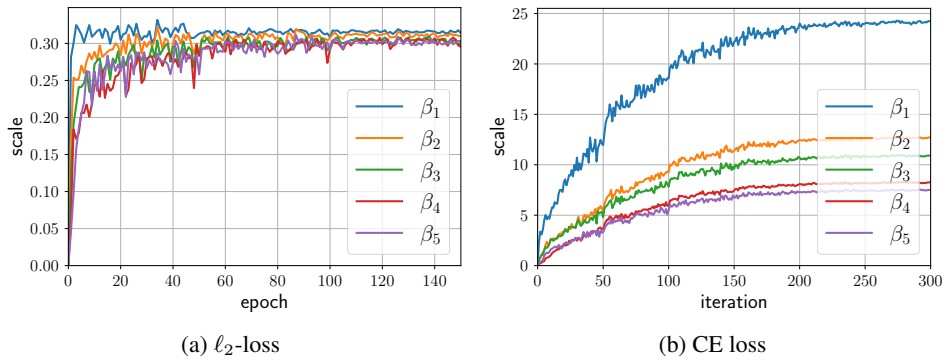

(a) $\ell_2$-loss            (b) CE loss

Figure 7: Solution trajectories of LARS on the CIFAR-10 dataset with $\ell_2$-loss and CE loss. The test accuracies are 90.54% for $\ell_2$-loss and 91.09% for CE loss.

the learning rate scales linearly with the batch size. For batch size of 32, we used SGD with a base learning rate $\eta = 0.1$ and 5 warm-up epochs. For batch size of 64, we used SGD with a base learning rate of $\eta = 0.2$ and 5 warm-up epochs. For batch size of 256, we used LARS with a base learning rate of 1. The remaining hyperparameters are set to their default values. The results are reported in Figure 8. We see that the projected solution trajectories share a similar monotonic behavior for different batch sizes.

## B.7 EXPERIMENTS FOR RESNET-18 WITH SGD ON CIFAR-10

Lastly, we train ResNet-18 with SGD on CIFAR-10. The results can be seen in Figure 9. To achieve good generalization, we use a large learning rate $\eta = 0.3$ (with 5 warm-up epochs starting at $1 \times 10^{-5}$), and decrease the learning rate by 0.33 for every 50 epochs. Moreover, We use a large batch size 512 to better imitate the trajectory of GD. Based on the simulation result, we can find an approximately monotonic (modulo the fluctuations caused by the randomness of the gradients) behavior of the dynamics of the top-5 components, which again validifies our theoretical results.

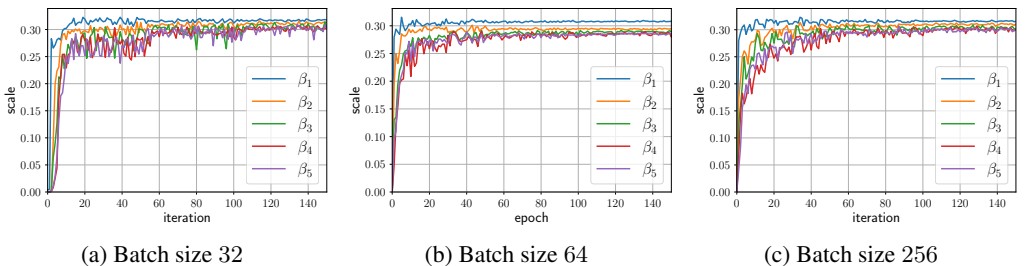

(a) Batch size 32       (b) Batch size 64      (c) Batch size 256

Figure 8: Solution trajectories for AlexNet on the CIFAR-10 dataset with different batch sizes. The test accuracies for batch size 32, 64 and 256 are 91.50%, 91.79%, and 90.12%, respectively.

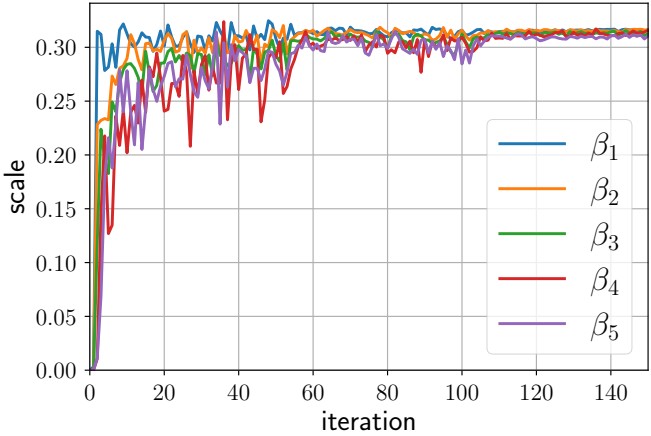

Figure 9: Solution trajectories for ResNet-18 on the CIFAR-10 dataset with SGD solver. The best test accuracy is 94.45%.

## C    RELATED WORK

**GD for general nonconvex optimization.**    Gradient descent and its stochastic or adaptive variants are considered as the "go-to" algorithms in large-scale (unconstrained) nonconvex optimization. Because of their first-order nature, they are known to converge to first-order stationary points (Nesterov, 1998). Only recently it has been shown that GD (Lee et al., 2019; Panageas et al., 2019) and its variants, such as perturbed GD (Jin et al., 2017) and SGD (Fang et al., 2019; Daneshmand et al., 2018), can avoid saddle points and converge to a second-order stationary point. However, these guarantees do not quantify the distance between the obtained solution and the globally optimal and/or true solutions. To the best of our knowledge, the largest subclass of nonconvex optimization problems for which GD or its variants converge to meaningful solutions are those with benign landscapes. These problems include different variants of low-rank matrix optimization with exactly parameterized rank, namely matrix completion (Ge et al., 2016), matrix sensing (Ge et al., 2017; Zhang et al., 2021), dictionary learning (Sun et al., 2016), and robust PCA (Fattahi & Sojoudi, 2020), as well as deep linear neural networks (Kawaguchi, 2016). However, benign landscape is too restrictive to hold in practice; for instance,Zhang (2021) shows that spurious local minima are ubiquitous in the low-rank matrix optimization, even under fairly mild conditions. Therefore, the notion of benign landscape cannot be used to explain the success of local search algorithms in more complex learning tasks.

**GD for specific learning problems.**    Although there does not exist a unifying framework to study the global convergence of GD for general learning tasks, its convergence has been established in specific learning problems, such as kernel regression (which includes neural tangent kernel (Jacot et al., 2018)), sparse recovery (Vaskevicius et al., 2019), matrix factorization (Li et al., 2018), tensor decomposition (Wang et al., 2020; Ge et al., 2021), and linear neural network (Arora et al., 2018). In what follows, we review specific learning tasks that are most related to our work.

The convergence of GD on kernel regression was studied far before the emergence of deep learning. Bauer et al. (2007); Raskutti et al. (2014) establish the convergence of gradient descent on a special class of nonparametric kernel regression called reproducing kernel Hilbert space (RKHS). Recently, Jacot et al. (2018) discovered that under some conditions, neural networks can be approximated by a specific type of kernel models called neural tangent kernel (NTK). Later on, a series of papers studied the optimization (Allen-Zhu et al., 2019b) and generalization (Allen-Zhu et al., 2019a; Arora et al., 2019b) properties of NTK. As for the matrix factorization, Li et al. (2018); Stöger & Soltanolkotabi (2021) studied the global convergence of GD on the symmetric matrix sensing with noiseless measurements and overestimated rank. Later, these results were extended to noisy (Zhuo et al., 2021), asymmetric (Ye & Du, 2021), and robust (Ma & Fattahi, 2022b) variants. Wang et al. (2020); Ge et al. (2021) studied the dynamic of a modified GD for overcomplete nonconvex tensor decomposition. Moreover, Razin et al. (2021; 2022) analyzed the implicit regularization and the incremental learning of gradient flow in hierarchical tensor decomposition and showed its connection to neural networks.

**Conjugate kernel.**    Conjugate kernel (CK) at the initial point has been considered as one of the promising methods for studying the generalization properties of DNNs (Daniely et al., 2016; Hu & Huang, 2021; Fan & Wang, 2020). However, similar to NTK, a major shortcoming of CK is that it cannot fully characterize the behavior of the practical neural networks (Vyas et al., 2022). Recent results have suggested that the conjugate kernel evaluated *after training* (for both NTK and CK) can better describe the generalization properties of DNNs (Fort et al., 2020; Long, 2021). In our work, we show that such "after kernel regime" can also be adapted to study the optimization trajectory of practical DNNs.

## D    PROOFS FOR GENERAL FRAMEWORK

### D.1    PROOF OF PROPOSITION 1

To prove this proposition, we first combine (1) and (GD):

$$\boldsymbol{\theta}_{t+1} = \boldsymbol{\theta}_t - \eta \nabla \mathcal{L}(\boldsymbol{\theta}_t) = \boldsymbol{\theta}_t - \eta \sum_{i \in \mathcal{I}} \left(\beta_i(\boldsymbol{\theta}_t) - \beta_i^\star\right) \nabla \beta_i(\boldsymbol{\theta}_t). \tag{10}$$

For notational simplicity, we denote $E(\boldsymbol{\theta}_t) = \frac{1}{2} \sum_{i \in \mathcal{E}} \beta_i^2(\boldsymbol{\theta}_t)$. Then, one can write

$$\boldsymbol{\theta}_{t+1} = \boldsymbol{\theta}_t - \eta \nabla \mathcal{L}(\boldsymbol{\theta}_t) = \boldsymbol{\theta}_t - \eta \sum_{i \in \mathcal{S}} \left( \beta_i(\boldsymbol{\theta}_t) - \beta_i^\star \right) \nabla \beta_i(\boldsymbol{\theta}_t) - \eta \nabla E(\boldsymbol{\theta}_t). \tag{11}$$

Due to the Mean-Value Theorem, there exists a $\xi \in \mathbb{R}^m$ such that

$$\beta_i(\boldsymbol{\theta}_{t+1}) = \beta_i \left( \boldsymbol{\theta}_t - \eta \sum_{j \in \mathcal{I}} \left( \beta_j(\boldsymbol{\theta}_t) - \beta_j^\star \right) \nabla \beta_j(\boldsymbol{\theta}_t) \right)$$

$$= \beta_i(\boldsymbol{\theta}_t) - \eta \sum_{j \in \mathcal{I}} \left( \beta_j(\boldsymbol{\theta}_t) - \beta_j^\star \right) \langle \nabla \beta_i(\boldsymbol{\theta}_t), \nabla \beta_j(\boldsymbol{\theta}_t) \rangle + \frac{\eta^2}{2} \left\langle \nabla \mathcal{L}(\boldsymbol{\theta}_t), \nabla^2 \beta_i(\xi) \nabla \mathcal{L}(\boldsymbol{\theta}_t) \right\rangle. \tag{12}$$

On the other hand, one can write

$$\left| \left\langle \nabla \mathcal{L}(\boldsymbol{\theta}_t), \nabla^2 \beta_i(\xi) \nabla \mathcal{L}(\boldsymbol{\theta}_t) \right\rangle \right| \leq \sup_{\boldsymbol{\theta}} \left\| \nabla^2 \beta_i(\boldsymbol{\theta}) \right\| \left\| \nabla \mathcal{L}(\boldsymbol{\theta}_t) \right\|^2. \tag{13}$$

For $\sup_{\boldsymbol{\theta}} \left\| \nabla^2 \beta_i(\boldsymbol{\theta}) \right\|$, we further have

$$\begin{aligned}
\sup_{\boldsymbol{\theta}} \left\| \nabla^2 \beta_i(\boldsymbol{\theta}) \right\| &= \sup_{\boldsymbol{\theta}} \left\| \nabla^2 \mathbb{E}[f_{\boldsymbol{\theta}}(\boldsymbol{x}) \phi(\boldsymbol{x})] \right\| \\
&= \sup_{\boldsymbol{\theta}} \left\| \mathbb{E}[\nabla^2 f_{\boldsymbol{\theta}}(\boldsymbol{x}) \phi(\boldsymbol{x})] \right\| \\
&\leq \sup_{\boldsymbol{\theta}} \mathbb{E} \left[ \left\| \nabla^2 f_{\boldsymbol{\theta}}(\boldsymbol{x}) \right\| |\phi(\boldsymbol{x})| \right] \\
&\overset{(a)}{\leq} \sup_{\boldsymbol{\theta}} \left( \mathbb{E} \left[ \left\| \nabla^2 f_{\boldsymbol{\theta}}(\boldsymbol{x}) \right\|^2 \right] \right)^{1/2} \left( \mathbb{E} \left[ \phi^2(\boldsymbol{x}) \right] \right)^{1/2} \\
&\overset{(b)}{\leq} L_H.
\end{aligned} \tag{14}$$

Here, we used Cauchy-Schwartz inequality for (a). Moreover, for (b), we used Assumption 1 and the definition of the orthonormal basis. Hence, we have

$$\beta_i(\boldsymbol{\theta}_{t+1}) = \beta_i(\boldsymbol{\theta}_t) - \eta \sum_{j \in \mathcal{I}} \left( \beta_j(\boldsymbol{\theta}_t) - \beta_j^\star \right) \langle \nabla \beta_i(\boldsymbol{\theta}_t), \nabla \beta_j(\boldsymbol{\theta}_t) \rangle \pm (1/2) \eta^2 L_H \left\| \nabla \mathcal{L}(\boldsymbol{\theta}_t) \right\|^2. \tag{15}$$

Now, it suffices to bound $\left\| \nabla \mathcal{L}(\boldsymbol{\theta}_t) \right\|^2$. Using Cauchy-Schwarz inequality, we have

$$\begin{aligned}
\left\| \nabla \mathcal{L}(\boldsymbol{\theta}_t) \right\|^2 &= \left\| \nabla \mathbb{E} \left[ \frac{1}{2} \left( f_{\boldsymbol{\theta}_t} - f_{\boldsymbol{\theta}^\star} \right)^2 \right] \right\|^2 \\
&= \left\| \mathbb{E} \left[ \left( f_{\boldsymbol{\theta}_t} - f_{\boldsymbol{\theta}^\star} \right) \nabla f_{\boldsymbol{\theta}_t} \right] \right\|^2 \\
&\leq \left\| f_{\boldsymbol{\theta}_t} - f_{\boldsymbol{\theta}^\star} \right\|_{L^2(\mathcal{D})}^2 \left\| \nabla f_{\boldsymbol{\theta}_t} \right\|_{L^2(\mathcal{D})}^2 \\
&\leq 4 L_g^2 L_f^2.
\end{aligned} \tag{16}$$

Therefore, we conclude that

$$\beta_i(\boldsymbol{\theta}_{t+1}) = \beta_i(\boldsymbol{\theta}_t) - \eta \sum_{j \in \mathcal{I}} \left( \beta_j(\boldsymbol{\theta}_t) - \beta_j^\star \right) \langle \nabla \beta_i(\boldsymbol{\theta}_t), \nabla \beta_j(\boldsymbol{\theta}_t) \rangle \pm 2 \eta^2 L_H L_g^2 L_f^2. \tag{17}$$

which completes the proof. $\qquad\square$

### D.2 PROOF OF THEOREM 1

*Proof.* Invoking the gradient independence condition, Proposition 1 can be simplified as

$$\begin{aligned}
\beta_i(\boldsymbol{\theta}_{t+1}) &= \beta_i(\boldsymbol{\theta}_t) - \eta \sum_{j \in \mathcal{I}} \left( \beta_j(\boldsymbol{\theta}_t) - \beta_j^\star \right) \langle \nabla \beta_i(\boldsymbol{\theta}_t), \nabla \beta_j(\boldsymbol{\theta}_t) \rangle \pm 2 \eta^2 L_H L_g^2 L_f^2 \\
&= \beta_i(\boldsymbol{\theta}_t) - \eta \left( \beta_i(\boldsymbol{\theta}_t) - \beta_i^\star \right) \left\| \nabla \beta_i(\boldsymbol{\theta}_t) \right\|^2 \pm 2 \eta^2 L_H L_g^2 L_f^2.
\end{aligned} \tag{18}$$

We next provide upper and lower bounds for the residual and signal terms. Recall that $\mathcal{S} = \{i \in \mathcal{I} : \beta_i^\star \neq 0\}$, and $\mathcal{E} = \mathcal{I}\backslash\mathcal{S}$. We first consider the dynamic of the signal term $\beta_i(\boldsymbol{\theta}_t), i \in \mathcal{S}$. Without loss of generality, we assume $\beta_i^\star > 0$. Then, due to the gradient dominance condition, we have the following lower bound

$$
\begin{aligned}
\beta_i(\boldsymbol{\theta}_{t+1}) &\geq \beta_i(\boldsymbol{\theta}_t) - \eta\left(\beta_i(\boldsymbol{\theta}_t) - \beta_i^\star\right)\|\nabla\beta_i(\boldsymbol{\theta}_t)\|^2 - 2\eta^2 L_H L_g^2 L_f^2 \\
&\geq \left(1 + C^2\eta\left(\beta_i^\star - \beta_i(\boldsymbol{\theta}_t)\right)\beta_i^{2\gamma-1}(\boldsymbol{\theta}_t)\right)\beta_i(\boldsymbol{\theta}_t) - 2\eta^2 L_H L_g^2 L_f^2, \quad i \in \mathcal{S}
\end{aligned}
\tag{19}
$$

Next, for the dynamic of the residual term $\beta_i(\boldsymbol{\theta}_t), i \in \mathcal{E}$, we have

$$
\beta_i(\boldsymbol{\theta}_{t+1}) = \left(1 - \eta\|\nabla\beta_i(\boldsymbol{\theta}_t)\|^2\right)\beta_i(\boldsymbol{\theta}_t) \pm 2\eta^2 L_H L_g^2 L_f^2.
\tag{20}
$$

Next, we show that $\|\nabla\beta_i(\boldsymbol{\theta}_t)\| \leq L_g$. One can write,

$$
\begin{aligned}
\|\nabla\beta_i(\boldsymbol{\theta}_t)\| &= \|\nabla\mathbb{E}\left[f_{\boldsymbol{\theta}_t}(\boldsymbol{x})\phi_i(\boldsymbol{x})\right]\| \\
&\leq \mathbb{E}\left[\|\nabla f_{\boldsymbol{\theta}_t}(\boldsymbol{x})\|\,|\phi_i(\boldsymbol{x})|\right] \\
&\leq \left(\mathbb{E}\left[\|\nabla f_{\boldsymbol{\theta}_t}(\boldsymbol{x})\|^2\right]\right)^{1/2}\left(\mathbb{E}\left[\phi^2(\boldsymbol{x})\right]\right)^{1/2} \\
&\leq L_g.
\end{aligned}
\tag{21}
$$

Due to our choice of the step-size, we have $\eta \lesssim \frac{1}{L_g^2}$, which in turn implies $\left|1 - \eta\|\nabla\beta_i(\boldsymbol{\theta}_t)\|^2\right| \leq 1$. Therefore, we have

$$
|\beta_i(\boldsymbol{\theta}_{t+1})| \leq |\beta_i(\boldsymbol{\theta}_t)| + 2\eta^2 L_H L_g^2 L_f^2.
\tag{22}
$$

Now, we are ready to prove the theorem. We divide it into two cases.

**Case 1:** $\gamma = \frac{1}{2}$. In this case, since we set the step-size $\eta \lesssim \frac{\alpha}{\sqrt{d}C^2 L_H L_g^2 L_f^2}\beta_k^\star \log^{-1}\left(\frac{d\beta_k^\star}{C_1\alpha}\right)$, we can simplify the dynamics of both signal and residual terms in Equation 19 and Equation 20 as follows

$$
\begin{aligned}
\beta_i(\boldsymbol{\theta}_{t+1}) &\geq \left(1 + 0.5C^2\eta\left(\beta_i^\star - \beta_i(\boldsymbol{\theta}_t)\right)\right)\beta_i(\boldsymbol{\theta}_t) \quad &\forall i \in \mathcal{S}, \\
|\beta_i(\boldsymbol{\theta}_{t+1})| &\leq |\beta_i(\boldsymbol{\theta}_t)| + 2\eta^2 L_H L_g^2 L_f^2 \quad &\forall i \in \mathcal{E}.
\end{aligned}
\tag{23}
$$

We first analyze the dynamic of signal $\beta_i(\boldsymbol{\theta}_t)$ for $i \in \mathcal{S}$. To this goal, we further divide this case into two phases. In the first phase, we assume $C_1\alpha \leq \beta_i(\boldsymbol{\theta}_t) \leq \frac{1}{2}\beta_i^\star$. Under this assumption, we can simplify the dynamic of $\beta_i(\boldsymbol{\theta}_t)$ as

$$
\beta_i(\boldsymbol{\theta}_{t+1}) \geq \left(1 + 0.25C^2\eta\beta_i^\star\right)\beta_i(\boldsymbol{\theta}_t).
\tag{24}
$$

Therefore, within $T_1 = \mathcal{O}\left(\frac{1}{C^2\eta\beta_i^\star}\log\left(\frac{\beta_i^\star}{C_1\alpha}\right)\right)$ iterations, $\beta_i(\boldsymbol{\theta}_t)$ becomes larger than $\frac{1}{2}\beta_i^\star$. In the second phase, we assume that $\beta_i(\boldsymbol{\theta}_t) \geq \beta_i^\star/2$ and define $y_t = \beta_i^\star - \beta_i(\boldsymbol{\theta}_t)$. One can write

$$
\begin{aligned}
y_{t+1} &\leq \left(1 - 0.5C^2\eta\beta_i(\boldsymbol{\theta}_t)\right)y_t \\
&\leq (1 - 0.25C^2\eta\beta_i^\star)y_t.
\end{aligned}
\tag{25}
$$

Hence, with additional $T_2 = \mathcal{O}\left(\frac{1}{C^2\eta\beta_i^\star}\log\left(\frac{d\beta_i^\star}{\alpha}\right)\right)$, we have $y_t \leq \frac{\alpha}{\sqrt{d}}$ which implies $\beta_i(\boldsymbol{\theta}_t) \geq \beta_i^\star - \frac{\alpha}{\sqrt{d}}$. Next, we show that there exists a time $t^\star$ such that $\beta_i^\star - \frac{1}{\sqrt{d}}\alpha \leq \beta_i(\boldsymbol{\theta}_{t^\star}) \leq \beta_i^\star + \frac{1}{\sqrt{d}}\alpha$. Without loss of generality, we assume that $t^\star$ is the first time that $\beta_i(\boldsymbol{\theta}_t) \geq \beta_i^\star - \frac{1}{\sqrt{d}}\alpha$. Due to the dynamic of $\beta_i(\boldsymbol{\theta}_t)$, the distance between two adjacent iterations can be upper bounded as

$$
\begin{aligned}
|\beta_i(\boldsymbol{\theta}_{t+1}) - \beta_i(\boldsymbol{\theta}_t)| &\leq \eta|\beta_i^\star - \beta_i(\boldsymbol{\theta}_t)|\,\|\nabla\beta_i(\boldsymbol{\theta}_t)\|^2 + 2\eta^2 L_H L_g^2 L_f^2 \\
&\leq \eta L_g^2|\beta_i^\star - \beta_i(\boldsymbol{\theta}_t)| + 2\eta^2 L_H L_g^2 L_f^2.
\end{aligned}
\tag{26}
$$

In particular, for $t = t^\star - 1$, we have $\beta_i(\boldsymbol{\theta}_{t^\star-1}) \leq \beta_i^\star - \frac{1}{\sqrt{d}}\alpha$, which in turn implies

$$
\begin{aligned}
\beta_i(\boldsymbol{\theta}_{t^\star}) &\leq \beta_i(\boldsymbol{\theta}_{t^\star-1}) + \eta L_g^2|\beta_i^\star - \beta_i(\boldsymbol{\theta}_{t^\star-1})| + 2\eta^2 L_H L_g^2 L_f^2 \\
&\leq \beta_i^\star + 2\eta^2 L_H L_g^2 L_f^2 \\
&\leq \beta_i^\star + \frac{1}{\sqrt{d}}\alpha.
\end{aligned}
\tag{27}
$$

Therefore, for each $i \in \mathcal{S}$, we have $|\beta_i(\boldsymbol{\theta}_t) - \beta_i^\star| \leq \frac{1}{\sqrt{d}}\alpha$ within $T_i = \mathcal{O}\left(\frac{1}{C^2\eta\beta_i^\star}\log\left(\frac{d\beta_i^\star}{C_1\alpha}\right)\right)$ iterations. Meanwhile, we can show that the residual term $|\beta_i(\boldsymbol{\theta}_t)|, \forall i \in \mathcal{E}$ remains small for $\max_{i \in \mathcal{S}} T_i = \mathcal{O}\left(\frac{1}{C^2\eta\beta_k^\star}\log\left(\frac{d\beta_k^\star}{C_1\alpha}\right)\right)$ iterations:

$$
\begin{aligned}
|\beta_i(\boldsymbol{\theta}_t)| &\leq |\beta_i(\boldsymbol{\theta}_0)| + \max_{i \in \mathcal{S}} T_i \cdot 2\eta^2 L_H L_g^2 L_f^2 \\
&= |\beta_i(\boldsymbol{\theta}_0)| + \mathcal{O}\left(\frac{1}{\beta_k^\star}\eta\log\left(\frac{\beta_k^\star}{C_1\alpha}\right)L_H L_g^2 L_f^2\right) \\
&= |\beta_i(\boldsymbol{\theta}_0)| + \mathcal{O}\left(\frac{1}{\sqrt{d}}\alpha\right).
\end{aligned}
\tag{28}
$$

Therefore, we have that within $T = \mathcal{O}\left(\frac{1}{\eta\beta_k^\star}\log\left(\frac{d\beta_k^\star}{C_1\alpha}\right)\right)$ iterations:

$$
\|f_{\boldsymbol{\theta}_T} - f_{\boldsymbol{\theta}^\star}\|_{L^2(\mathcal{D})}^2 = \sum_{i \in \mathcal{S}}(\beta_i^\star - \beta_i(\boldsymbol{\theta}_T))^2 + \sum_{i \in \mathcal{E}}\beta_i^2(\boldsymbol{\theta}_T) \lesssim \alpha^2.
\tag{29}
$$

**Case 2: $\frac{1}{2} < \gamma \leq 1$.** In this case, we have the following bounds for the signal and residual terms

$$
\begin{aligned}
\beta_i(\boldsymbol{\theta}_{t+1}) &\geq \left(1 + C^2\eta\left(\beta_i^\star - \beta_i(\boldsymbol{\theta}_t)\right)\beta_i^{2\gamma-1}(\boldsymbol{\theta}_t)\right)\beta_i(\boldsymbol{\theta}_t) - 2\eta^2 L_H L_g^2 L_f^2 \quad \forall i \in \mathcal{S}, \\
|\beta_i(\boldsymbol{\theta}_{t+1})| &\leq |\beta_i(\boldsymbol{\theta}_t)| + 2\eta^2 L_H L_g^2 L_f^2 \quad\quad\quad\quad\quad\quad\quad\quad\quad\quad \forall i \in \mathcal{E}.
\end{aligned}
\tag{30}
$$

We first analyze the dynamic of the signal term $\beta_i(\boldsymbol{\theta}_t)$ for $i \in \mathcal{S}$. We will show that $|\beta_i(\boldsymbol{\theta}_t) - \beta_i^\star| \leq \frac{\alpha}{\sqrt{k}}$ within $T = \mathcal{O}\left(\frac{1}{C^2\eta\beta_i^\star\alpha^{2\gamma-1}}\right)$ iterations. Due to $\eta \lesssim \frac{\alpha^{2\gamma}}{\sqrt{d}C^2 L_H L_g^2 L_f^2}\beta_k^{\star 2\gamma}$, we can further simplify the dynamic of $\beta_i(\boldsymbol{\theta}_t)$ as

$$
\beta_i(\boldsymbol{\theta}_{t+1}) \geq \left(1 + 0.5C^2\eta\left(\beta_i^\star - \beta_i(\boldsymbol{\theta}_t)\right)\beta_i^{2\gamma-1}(\boldsymbol{\theta}_t)\right)\beta_i(\boldsymbol{\theta}_t).
\tag{31}
$$

Next, we divide our analysis into two phases. In the first phase, we have $\beta_i(\boldsymbol{\theta}_t) \leq \frac{1}{2}\beta_i^\star$. We denote the number of iterations for this phase as $T_{i,1}$. We further divide this period into $\lceil\log(\beta_i^\star/2\alpha)\rceil$ substages. In each Substage $k$, we have $C_1 2^{k-1}\alpha \leq \beta_i(\boldsymbol{\theta}_t) \leq C_1 2^k\alpha$. Let $t_k$ be the number of iterations in Substage $k$. We first provide an upper bound for $t_k$. To this goal, note that at this substage

$$
\begin{aligned}
\beta_i(\boldsymbol{\theta}_{t+1}) &\geq \left(1 + 0.5C^2\eta\left(\beta_i^\star - \beta_i(\boldsymbol{\theta}_t)\right)\beta_i^{2\gamma-1}(\boldsymbol{\theta}_t)\right)\beta_i(\boldsymbol{\theta}_t) \\
&\geq \left(1 + 0.25C^2\eta\beta_i^\star\beta_i^{2\gamma-1}(\boldsymbol{\theta}_t)\right)\beta_i(\boldsymbol{\theta}_t).
\end{aligned}
\tag{32}
$$

Hence, we have

$$
t_k \leq \frac{\log(2)}{\log\left(1 + 0.25C^2\eta\beta_i^\star(C_1 2^{k-1}\alpha)^{2\gamma-1}\right)}.
\tag{33}
$$

Summing over $t_k$, we obtain an upper bound for $T_{i,1}$

$$
T_{i,1} = \sum_{i=1}^{\lceil\log(\beta_1^\star/2\alpha)\rceil} t_k \lesssim \sum_{k=1}^{\infty}\frac{1}{C^2\eta\beta_i^\star(C_1 2^{k-1}\alpha)^{2\gamma-1}} \lesssim \frac{1}{C^2\eta\beta_i^\star(C_1\alpha)^{2\gamma-1}}.
\tag{34}
$$

Via a similar argument, we can show that in the second phase, we have $|\beta_i(\boldsymbol{\theta}_t) - \beta_i^\star| \lesssim \frac{1}{\sqrt{d}}\alpha$ within additional $T_{i,2} = \mathcal{O}\left(\frac{1}{C^2\eta\beta_i^\star(C_1\alpha)^{2\gamma-1}}\right)$ iterations. Therefore, for each $i \in \mathcal{S}$, we conclude that $|\beta_i(\boldsymbol{\theta}_t) - \beta_i^\star| \leq \frac{\alpha}{\sqrt{d}}$ within $T_i = T_{i,1} + T_{i,2} = \mathcal{O}\left(\frac{1}{C^2\eta\beta_i^\star(C_1\alpha)^{2\gamma-1}}\right)$ iterations. Meanwhile, for the residual term $\beta_i(\boldsymbol{\theta}_t), i \in \mathcal{E}$, we have

$$
\begin{aligned}
|\beta_i(\boldsymbol{\theta}_t)| &\leq |\beta_i(\boldsymbol{\theta}_0)| + \max_{i \in \mathcal{S}} T_i \cdot 2\eta^2 L_H L_g^2 L_f^2 \\
&= |\beta_i(\boldsymbol{\theta}_0)| + \mathcal{O}\left(\frac{1}{\sqrt{d}}\alpha\right).
\end{aligned}
\tag{35}
$$

Therefore,

$$
\|f_{\boldsymbol{\theta}_T} - f_{\boldsymbol{\theta}^\star}\|_{L^2(\mathcal{D})}^2 = \sum_{i \in \mathcal{S}}(\beta_i^\star - \beta_i(\boldsymbol{\theta}_T))^2 + \sum_{i \in \mathcal{E}}\beta_i^2(\boldsymbol{\theta}_T) \lesssim \alpha^2,
\tag{36}
$$

within $T = \mathcal{O}\left(\frac{1}{C^2\eta\beta_k^\star(C_1\alpha)^{2\gamma-1}}\right)$ iterations. This completes the proof. $\quad\square$

# E  PROOFS FOR KERNEL REGRESSION

## E.1  PROOF OF PROPOSITION 2

Note that $\boldsymbol{\theta}_{t+1} = \boldsymbol{\theta}_t - \eta(\boldsymbol{\theta}_t - \boldsymbol{\theta}^\star)$, and $\beta_i(\boldsymbol{\theta}) = \theta_i$. Hence, for every $1 \leq i \leq k$, we have

$$\beta_i(\boldsymbol{\theta}_{t+1}) = \beta_i(\boldsymbol{\theta}_t) + \eta(\theta_i^\star - \beta_i(\boldsymbol{\theta}_t)). \tag{37}$$

This in turn implies

$$\beta_i^\star - \beta_i(\boldsymbol{\theta}_{t+1}) = (1-\eta)\left(\beta_i^\star - \beta_i(\boldsymbol{\theta}_t)\right) \implies \beta_i^\star - \beta_i(\boldsymbol{\theta}_t) = (1-\eta)^t \left(\beta_i^\star - \beta_i(\boldsymbol{\theta}_0)\right). \tag{38}$$

For $i > k$, we have

$$\beta_i(\boldsymbol{\theta}_{t+1}) = (1-\eta)\beta_i(\boldsymbol{\theta}_t) \implies \beta_i(\boldsymbol{\theta}_t) = (1-\eta)^t \beta_i(\boldsymbol{\theta}_0), \tag{39}$$

which completes the proof. $\qquad\square$

## E.2  PROOF OF THEOREM 2

Due to our choice of initial point $\boldsymbol{\theta}_0 = \alpha\mathbf{1}, \alpha \lesssim |\beta_k^\star|$, we have $|\beta_i^\star - \beta_i(\boldsymbol{\theta}_0)| \leq 2|\beta_i^\star|$. Hence, by Proposition 2, we have

$$|\beta_i^\star - \beta_i(\boldsymbol{\theta}_t)| = (1-\eta)^t |\beta_i^\star - \beta_i(\boldsymbol{\theta}_0)| \leq 2(1-\eta)^t \beta_i^\star, \quad i \in \mathcal{S} \tag{40}$$

$$|\beta_i(\boldsymbol{\theta}_t)| \leq (1-\eta)^t \alpha, \quad i \in \mathcal{E}. \tag{41}$$

Therefore, to prove $|\beta_i^\star - \beta_i(\boldsymbol{\theta}_t)| \leq \frac{\alpha}{\sqrt{k}}, i \in \mathcal{S}$, it suffices to have

$$2(1-\eta)^t \beta_i^\star \leq \frac{\alpha}{\sqrt{k}} \implies t \gtrsim \frac{1}{\eta} \log\left(\frac{k\theta_i}{\alpha}\right). \tag{42}$$

On the other hand, to ensure $|\beta_i(\boldsymbol{\theta}_t)| \leq \frac{\alpha}{\sqrt{d}}, i \in \mathcal{E}$, it suffices to have

$$(1-\eta)^t \alpha \leq \frac{\alpha}{\sqrt{d}} \implies t \gtrsim \frac{1}{\eta} \log(d). \tag{43}$$

Recall that $\alpha \lesssim \frac{k|\theta_k|}{d}$ and $|\theta_1| \geq |\theta_2| \geq \cdots \geq |\theta_k| > 0$. Therefore, within $T = \mathcal{O}\left(\frac{1}{\eta} \log\left(\frac{k|\theta_1|}{\alpha}\right)\right)$ iterations, we have

$$\|\boldsymbol{\theta}_T - \boldsymbol{\theta}^\star\| \leq \sqrt{\sum_{i=1}^{k} \frac{\alpha^2}{k} + \sum_{i=k+1}^{d} \frac{\alpha^2}{d}} \leq \sqrt{2}\alpha, \tag{44}$$

which completes the proof. $\qquad\square$

# F  PROOFS FOR SYMMETRIC MATRIX FACTORIZATION

## F.1  INITIALIZATION

We start by proving that both lower bound on signals at $\boldsymbol{\theta}_0$ and upper bound on energy at $\boldsymbol{\theta}_0$ are satisfied with high probability. Recall that each element of $\boldsymbol{U}_0$ is drawn from $\mathcal{N}(0, \alpha^2)$. The following proposition characterizes the upper and lower bounds for different coefficients $\beta_{ij}(\boldsymbol{U}_0)$.

**Proposition 5** (Initialization). *With probability at least $1 - e^{-\Omega(r')}$, we have*

$$\beta_{ii}(\boldsymbol{U}_0) = \langle \boldsymbol{z}_i \boldsymbol{z}_i^\top, \boldsymbol{U}_0 \boldsymbol{U}_0^\top \rangle \geq \frac{1}{4} r'\alpha^2, \quad \textit{for all } 1 \leq i \leq r, \tag{45}$$

*and*

$$|\beta_{ij}(\boldsymbol{U}_0)| = \left|\langle \boldsymbol{z}_i \boldsymbol{z}_j^\top, \boldsymbol{U}_0 \boldsymbol{U}_0^\top \rangle\right| \leq 4\log(d)r'\alpha^2, \quad \textit{for all } i \neq j \textit{ or } i, j > r. \tag{46}$$

*Proof.* First note that $\boldsymbol{U}_0^\top \boldsymbol{z}_i \sim \mathcal{N}(0, \alpha^2 I_{r' \times r'})$. Hence, a standard concentration bound on Gaussian random vectors implies

$$\mathbb{P}\left(\left|\left\|\boldsymbol{U}_0^\top \boldsymbol{z}_i\right\| - \alpha\sqrt{r'}\right| \geq \alpha\delta\right) \leq 2\exp\{-c\delta^2\}. \tag{47}$$

Via a union bound, we have that with probability of at least $1 - e^{-Cr'}$:

$$\left\|\boldsymbol{U}_0^\top \boldsymbol{z}_i\right\| \leq 2\alpha\sqrt{r'\log(d)}, \quad 1 \leq i \leq d, \qquad \left\|\boldsymbol{U}_0^\top \boldsymbol{z}_i\right\| \geq 0.5\alpha\sqrt{r'}, \quad \forall 1 \leq i \leq r. \tag{48}$$

Given these bounds, one can write

$$\beta_{ii}(\boldsymbol{U}_0) = \langle \boldsymbol{z}_i \boldsymbol{z}_i^\top, \boldsymbol{U}_0 \boldsymbol{U}_0^\top \rangle = \left\|\boldsymbol{U}_0^\top \boldsymbol{z}_i\right\|^2 \geq \frac{1}{4}r'\alpha^2 \quad \forall 1 \leq i \leq r, \tag{49}$$

and

$$|\beta_{ij}(\boldsymbol{U}_0)| = \left|\langle \boldsymbol{z}_i \boldsymbol{z}_j^\top, \boldsymbol{U}_0 \boldsymbol{U}_0^\top \rangle\right| \leq \left\|\boldsymbol{U}_0^\top \boldsymbol{z}_i\right\| \left\|\boldsymbol{U}_0^\top \boldsymbol{z}_j\right\| \leq 4\log(d)r'\alpha^2 \quad \forall 1 \leq i, j \leq d, \tag{50}$$

which completes the proof. $\qquad\square$

### F.2 ONE-STEP DYNAMICS

In this section, we characterize the one-step dynamics of the basis coefficients. To this goal, we first provide a more precise statement of Proposition 3 along with its proof.

**Proposition 6.** *For the diagonal element* $\beta_{ii}(\boldsymbol{U}_t)$*, we have*

$$\beta_{ii}(\boldsymbol{U}_{t+1}) = (1 + 2\eta(\sigma_i - \beta_{ii}(\boldsymbol{U}_t)))\beta_{ii}(\boldsymbol{U}_t) - 2\eta\sum_{j\neq i}\beta_{ij}^2(\boldsymbol{U}_t)$$

$$+ \eta^2\left(\sum_{j,k}\beta_{ij}(\boldsymbol{U}_t)\beta_{ik}(\boldsymbol{U}_t)\beta_{jk}(\boldsymbol{U}_t) - 2\sigma_i\sum_j\beta_{ij}^2(\boldsymbol{U}_t) + \sigma_i^2\beta_{ii}(\boldsymbol{U}_t)\right). \tag{51}$$

*where* $\sigma_i = 0$ *for* $r < i \leq d$*. Moreover, for every* $i \neq j$*, we have*

$$\beta_{ij}(\boldsymbol{U}_{t+1}) = (1 + \eta(\sigma_i + \sigma_j - 2\beta_{ii}(\boldsymbol{U}_t) - 2\beta_{jj}(\boldsymbol{U}_t)))\beta_{ij}(\boldsymbol{U}_t) + 2\eta\sum_{k\neq i,j}\beta_{ik}(\boldsymbol{U}_t)\beta_{jk}(\boldsymbol{U}_t)$$

$$+ \eta^2\left(\sum_{k,l}\beta_{ik}(\boldsymbol{U}_t)\beta_{kl}(\boldsymbol{U}_t)\beta_{lj}(\boldsymbol{U}_t) - (\sigma_i + \sigma_j)\sum_k\beta_{ik}(\boldsymbol{U}_t)\beta_{kj}(\boldsymbol{U}_t) + \sigma_i\sigma_j\beta_{ij}(\boldsymbol{U}_t)\right). \tag{52}$$

*Proof.* The iterations of GD on SMF take the form

$$\boldsymbol{U}_{t+1} = \boldsymbol{U}_t - \eta(\boldsymbol{U}_t \boldsymbol{U}_t^\top - \boldsymbol{M}^\star)\boldsymbol{U}_t. \tag{53}$$

This leads to

$$\boldsymbol{U}_{t+1}\boldsymbol{U}_{t+1}^\top = \boldsymbol{U}_t\boldsymbol{U}_t^\top - \eta(\boldsymbol{U}_t\boldsymbol{U}_t^\top - \boldsymbol{M}^\star)\boldsymbol{U}_t\boldsymbol{U}_t^\top - \eta\boldsymbol{U}_t\boldsymbol{U}_t^\top(\boldsymbol{U}_t\boldsymbol{U}_t^\top - \boldsymbol{M}^\star)$$
$$+ \eta^2(\boldsymbol{U}_t\boldsymbol{U}_t^\top - \boldsymbol{M}^\star)\boldsymbol{U}_t\boldsymbol{U}_t^\top(\boldsymbol{U}_t\boldsymbol{U}_t^\top - \boldsymbol{M}^\star). \tag{54}$$

Recall that $\beta_{ij}(\boldsymbol{U}_t) = \langle \boldsymbol{U}_t\boldsymbol{U}_t^\top, \boldsymbol{z}_i\boldsymbol{z}_j^\top\rangle$, $\boldsymbol{U}_t\boldsymbol{U}_t^\top = \sum_{i,j}\beta_{ij}(\boldsymbol{U}_t)\boldsymbol{z}_i\boldsymbol{z}_j^\top$, and $\boldsymbol{M}^\star = \sum_{i=1}^r \sigma_i\boldsymbol{z}_i\boldsymbol{z}_i^\top$. Based on these definitions, one can write

$$\beta_{ij}(\boldsymbol{U}_{t+1}) = \langle \boldsymbol{U}_{t+1}\boldsymbol{U}_{t+1}^\top, \boldsymbol{z}_i\boldsymbol{z}_j^\top\rangle$$
$$= \langle \boldsymbol{U}_t\boldsymbol{U}_t^\top, \boldsymbol{z}_i\boldsymbol{z}_j^\top\rangle - \eta\langle(\boldsymbol{U}_t\boldsymbol{U}_t^\top - \boldsymbol{M}^\star)\boldsymbol{U}_t\boldsymbol{U}_t^\top, \boldsymbol{z}_i\boldsymbol{z}_j^\top\rangle - \eta\langle\boldsymbol{U}_t\boldsymbol{U}_t^\top(\boldsymbol{U}_t\boldsymbol{U}_t^\top - \boldsymbol{M}^\star), \boldsymbol{z}_i\boldsymbol{z}_j^\top\rangle$$
$$+ \eta^2\langle(\boldsymbol{U}_t\boldsymbol{U}_t^\top - \boldsymbol{M}^\star)\boldsymbol{U}_t\boldsymbol{U}_t^\top(\boldsymbol{U}_t\boldsymbol{U}_t^\top - \boldsymbol{M}^\star), \boldsymbol{z}_i\boldsymbol{z}_j^\top\rangle. \tag{55}$$

In light of the above equality and the orthogonality of $\{z_i\}_{i \in [d]}$, the RHS of Equation 55 can be written in terms of $\beta_{kl}(U_t), 1 \leq k, l \leq d$. In particular

$$
\begin{aligned}
\langle (U_t U_t^\top - M^\star) U_t U_t^\top, z_i z_j^\top \rangle &= \left\langle \left( \sum_{i,j} (\beta_{ij}(U_t) - \beta_{ij}^\star) z_i z_j^\top \right) \sum_{i,j} \beta_{ij}(U_t) z_i z_j^\top, z_i z_j^\top \right\rangle \\
&= \left\langle \sum_{i,j} \sum_k (\beta_{ik}(U_t) - \beta_{ik}^\star) \beta_{kj}(U_t) z_i z_j^\top, z_i z_j^\top \right\rangle \\
&= \sum_k (\beta_{ik}(U_t) - \beta_{ik}^\star) \beta_{kj}(U_t) \\
&= (\beta_{ii}(U_t) - \sigma_i) \beta_{ij}(U_t) + \sum_{k \neq i} (\beta_{ik}(U_t) - \beta_{ik}^\star) \beta_{kj}(U_t).
\end{aligned}
\tag{56}
$$

Other terms in Equation 55 can be written in terms of $\beta_{ij}(U_t)$ in an identical fashion. Substituting these derivations back in Equation 55, we obtain

$$
\beta_{ij}(U_{t+1}) = (1 + \eta (\sigma_i + \sigma_j)) \beta_{ij}(U_t) - 2\eta \sum_k \beta_{ik}(U_t) \beta_{kj}(U_t)
$$
$$
+ \eta^2 \left( \sum_{k,l} \beta_{ik}(U_t) \beta_{kl}(U_t) \beta_{lj}(U_t) - (\sigma_i + \sigma_j) \sum_k \beta_{ik}(U_t) \beta_{kj}(U_t) + \sigma_i \sigma_j \beta_{ij}(U_t) \right).
\tag{57}
$$

Note that the above equality holds for any $1 \leq i, j \leq r'$. In particular, for $i = j$, we further have

$$
\beta_{ii}(U_{t+1}) = (1 + 2\eta (\sigma_i - \beta_{ii}(U_t))) \beta_{ii}(U_t) - 2\eta \sum_{j \neq i} \beta_{ij}^2(U_t)
$$
$$
+ \eta^2 \left( \sum_{j,k} \beta_{ij}(U_t) \beta_{ik}(U_t) \beta_{jk}(U_t) - 2\sigma_i \sum_j \beta_{ij}^2(U_t) + \sigma_i^2 \beta_{ii}(U_t) \right),
\tag{58}
$$

which completes the proof. $\qquad \square$

### F.3 PROOFS OF PROPOSITION 3 AND THEOREM 3

To streamline the presentation, we prove Proposition 3 and Theorem 3 simultaneously. The main idea behind our proof technique is to divide the solution trajectory into $r$ substages: in Substage $i$, the basis coefficient $\beta_{ii}(U_t)$ converges linearly to $\sigma_i$ while all the remaining coefficients remain almost unchanged. More precisely, suppose that Substage $i$ lasts from iteration $t_{i,s}$ to $t_{i,e}$. We will show that $\beta_{ii}(U_{t_{i,e}}) \approx \sigma_i$ and $\beta_{ij}(U_{t_{i,e}}) \approx \beta_{ij}(U_{t_{i,s}})$. Recall that $\gamma = \min_{1 \leq i \leq r} \sigma_i - \sigma_{i+1}$ is the eigengap of the true model, which we assume is strictly positive.

**Substage 1.** In the first stage, we show that $\beta_{11}(U_t)$ approaches $\sigma_1$ and $|\beta_{ij}(U_t)|, i, j \geq 2$ remains in the order of $\text{poly}(\alpha)$ within $T_1 = \mathcal{O}\left( \frac{1}{\eta \sigma_1} \log \left( \frac{\sigma_1}{\alpha} \right) \right)$ iterations. To formalize this idea, we further divide this substage into two phases. In the first phase (which we refer to as the warm-up phase), we show that $\beta_{11}(U_t)$ will quickly dominate the remaining terms $\beta_{ij}(U_t), \forall (i, j) \neq (1, 1)$ within $\mathcal{O}\left( \frac{1}{\eta \gamma} \log \log(d) \right)$ iterations. This is shown in the following lemma.

**Lemma 1** (Warm-up phase). *Suppose that the initial point satisfies Equation 45 and Equation 46. Then, within $\mathcal{O}\left( \frac{1}{\eta \gamma} \log \log(d) \right)$ iterations, we have*

$$
|\beta_{ij}(U_t)| \leq \beta_{11}(U_t) \lesssim r' \alpha^2 \log(d)^{1 + \sigma_1/\gamma}.
\tag{59}
$$

*Proof.* To show this, we use an inductive argument. Due to our choice of the initial point, we have $|\beta_{ij}(U_0)| \lesssim r' \alpha^2 \log(d)^{1 + \sigma_1/\gamma}, 1 \leq i, j \leq d$. Now, suppose that at time $t$, we have $|\beta_{ij}(U_t)| \lesssim$

$r'\alpha^2 \log(d)^{1+\sigma_1/\gamma}, 1 \le i, j \le d$. Then, by Proposition 6, we have

$$\beta_{11}(\boldsymbol{U}_{t+1}) = (1 + 2\eta(\sigma_1 - \beta_{11}(\boldsymbol{U}_t))) \beta_{11}(\boldsymbol{U}_t) - 2\eta \sum_{j \neq 1} \beta_{1j}^2(\boldsymbol{U}_t)$$

$$+ \eta^2 \left( \sum_{j,k} \beta_{1j}(\boldsymbol{U}_t)\beta_{1k}(\boldsymbol{U}_t)\beta_{jk}(\boldsymbol{U}_t) - 2\sigma_1 \sum_j \beta_{1j}^2(\boldsymbol{U}_t) + \sigma_1^2 \beta_{11}(\boldsymbol{U}_t) \right) \quad (60)$$

$$\ge \left( 1 + 2\eta\sigma_1 - \mathcal{O}\left(\eta dr'\alpha^2 \log(d)^{1+\sigma_1/\gamma}\right) \right) \beta_{11}(\boldsymbol{U}_t).$$

Similarly, for the remaining coefficients $\beta_{ij}(\boldsymbol{U}_t), \forall(i,j) \neq (1,1)$, we have

$$|\beta_{ij}(\boldsymbol{U}_{t+1})| \le \left( 1 + \eta\left(\sigma_1 + \sigma_2 + \mathcal{O}\left(dr'\alpha^2 \log(d)^{1+\sigma_1/\gamma}\right)\right) \right) |\beta_{ij}(\boldsymbol{U}_t)|. \quad (61)$$

Note that the stepsize satisfies $\eta \lesssim \frac{1}{\sigma_1}$, and $\sigma_1 - \sigma_2 \ge \gamma \gtrsim dr'\alpha^2 \log(d)^{1+\sigma_1/\gamma}$. Hence, we have

$$\begin{aligned}
\frac{\beta_{11}(\boldsymbol{U}_{t+1})}{|\beta_{ij}(\boldsymbol{U}_{t+1})|} &\ge \frac{1 + 2\eta\sigma_1 - \mathcal{O}\left(\eta dr'\alpha^2 \log(d)^{1+\sigma_1/\gamma}\right)}{1 + \eta\left(\sigma_1 + \sigma_2 + \mathcal{O}\left(dr'\alpha^2 \log(d)^{1+\sigma_1/\gamma}\right)\right)} \frac{\beta_{11}(\boldsymbol{U}_t)}{|\beta_{ij}(\boldsymbol{U}_t)|} \\
&= \left( 1 + \frac{\eta(\sigma_1 - \sigma_2 - \mathcal{O}(dr'\alpha^2 \log(d)^{1+\sigma_1/\gamma}))}{1 + \eta(\sigma_1 + \sigma_2 + \mathcal{O}(dr'\alpha^2 \log(d)^{1+\sigma_1/\gamma}))} \right) \frac{\beta_{11}(\boldsymbol{U}_t)}{|\beta_{ij}(\boldsymbol{U}_t)|} \\
&\ge \left( 1 + \frac{\eta\gamma}{1 + 0.5\eta(\sigma_1 + \sigma_2)} \right) \frac{\beta_{11}(\boldsymbol{U}_t)}{|\beta_{ij}(\boldsymbol{U}_t)|} \\
&\ge (1 + 0.5\eta\gamma) \frac{\beta_{11}(\boldsymbol{U}_t)}{|\beta_{ij}(\boldsymbol{U}_t)|}.
\end{aligned} \quad (62)$$

This further implies

$$\frac{\beta_{11}(\boldsymbol{U}_t)}{|\beta_{ij}(\boldsymbol{U}_t)|} \ge (1 + 0.5\eta\gamma)^t \frac{\beta_{11}(\boldsymbol{U}_0)}{|\beta_{ij}(\boldsymbol{U}_0)|}. \quad (63)$$

On the other hand, Equation 49 and Equation 50 imply that

$$\frac{\beta_{11}(\boldsymbol{U}_0)}{|\beta_{ij}(\boldsymbol{U}_0)|} \ge \frac{1}{16\log(d)}. \quad (64)$$

Hence, within $\mathcal{O}\left(\frac{1}{\eta\gamma}\log\log(d)\right)$ iterations, we have $\beta_{ij}(\boldsymbol{U}_t) \ge |\beta_{11}(\boldsymbol{U}_t)|, \forall(i,j) \neq (1,1)$. Moreover, we have that during this phase,

$$|\beta_{ij}(\boldsymbol{U}_t)| \le \beta_{11}(\boldsymbol{U}_t) \le r'\alpha^2 \log(d)(1 + 2\eta\sigma_1)^{\mathcal{O}\left(\frac{1}{\eta\gamma}\log\log(d)\right)} \lesssim r'\alpha^2 \log(d)^{1+\sigma_1/\gamma}, \quad (65)$$

which completes the proof. $\qquad \square$

After the warm-up phase, we show that $\beta_{11}(\boldsymbol{U}_t)$ quickly approaches $\sigma_1$ while the remaining coefficients remain small.

**Lemma 2** (Fast growth). *After the warm-up phase followed by* $\mathcal{O}\left(\frac{1}{\eta\sigma_1}\log\left(\frac{\sigma_1}{\alpha}\right)\right)$ *iterations, we have*

$$0.99\sigma_1 \le \beta_{11}(\boldsymbol{U}_t) \le \sigma_1. \quad (66)$$

*Moreover, for* $|\beta_{ij}(\boldsymbol{U}_t)|, \forall(i,j) \neq (1,1)$, *we have*

$$|\beta_{ij}(\boldsymbol{U}_t)| \lesssim \sigma_1 r' \log(d)\alpha^{\frac{2\sigma_1 - \sigma_i - \sigma_j}{2\sigma_1}}. \quad (67)$$

Before providing the proof of Lemma 2 we analyze an intermediate logistic map which, as will be shown later, closely resembles the dynamic of $\beta_{ij}(\boldsymbol{U}_t)$:

$$x_{t+1} = (1 + \eta\sigma - \eta x_t)x_t, \quad x_0 = \alpha. \quad \text{(logistic map)}$$

The following two lemmas characterize the dynamic of a single logistic map, as well as the dynamic of the ratio between two different logistic maps.

**Lemma 3** (Iteration complexity of logistic map). *Suppose that $\alpha \leq \varepsilon \leq 0.1\sigma$. Then, for the logistic map, we have $x_T \geq \sigma - \varepsilon$ within $T = \frac{1}{\log(1+\eta\sigma)} \left( \log(4\sigma/\alpha) + \log(4\sigma/\varepsilon) \right)$ iterations.*

**Lemma 4** (Separation between two logistic maps). *Let $\sigma_1, \sigma_2$ be such that $\sigma_1 - \sigma_2 > 0$, and*

$$x_{t+1} = (1 + \eta\sigma_1 - \eta x_t)x_t, \quad x_0 = \alpha$$
$$y_{t+1} = (1 + \eta\sigma_2 - \eta y_t)y_t, \quad y_0 = \alpha$$

*Then, within $T = \frac{1}{\log(1+\eta\sigma_1)} \log\left( \frac{16\sigma_1^2}{\varepsilon\alpha} \right)$ iterations, we have*

$$\sigma_1 - \varepsilon \leq x_T \leq \sigma_1, \quad y_T \leq \frac{16\sigma_1^2}{\varepsilon} \alpha^{\frac{\sigma_1 - \sigma_2}{\sigma_1 + \sigma_2}}.$$

The proofs of Lemmas 3 and 4 are deferred to Appendix F.4. We are now ready to provide the proof of Lemma 2.

*Proof of Lemma 2.* Similar to the proof of Lemma 1, we use an inductive argument. Suppose that $t_0$ is when the second phase starts. According to Lemma 1, we have $|\beta_{ij}(\boldsymbol{U}_{t_0})| \leq \beta_{11}(\boldsymbol{U}_{t_0}) \lesssim r'\alpha^2 \log(d)^{1+\sigma_1/\gamma} \lesssim \sigma_1 r' \log(d)\alpha^{\frac{2\sigma_1 - \sigma_i - \sigma_j}{2\sigma_1}}$. Therefore, the base case of our induction holds. Next, suppose that at some time $t$ within the second phase, we have $|\beta_{ij}(\boldsymbol{U}_t)| \lesssim \sigma_1\alpha^{\frac{2\sigma_1 - \sigma_i - \sigma_j}{2\sigma_1}}, \forall(i,j) \neq (1,1)$. Our goal is to show that $|\beta_{ij}(\boldsymbol{U}_{t+1})| \lesssim \sigma_1\alpha^{\frac{2\sigma_1 - \sigma_i - \sigma_j}{2\sigma_1}}$. To this goal, we consider two cases.

Case I: $i \leq r$ or $j \leq r$ and $(i,j) \neq (1,1)$. We have

$$|\beta_{ij}(\boldsymbol{U}_{t+1})| \leq (1 + \eta(\sigma_i + \sigma_j - 2\beta_{ii}(\boldsymbol{U}_t) - 2\beta_{jj}(\boldsymbol{U}_t)))|\beta_{ij}(\boldsymbol{U}_t)| + 2\eta \sum_{k \neq i,j} |\beta_{ik}(\boldsymbol{U}_t)\beta_{jk}(\boldsymbol{U}_t)|$$

$$+ \eta^2 \left| \left( \sum_{k,l} \beta_{ik}(\boldsymbol{U}_t)\beta_{kl}(\boldsymbol{U}_t)\beta_{lj}(\boldsymbol{U}_t) - (\sigma_i + \sigma_j)\sum_k \beta_{ik}(\boldsymbol{U}_t)\beta_{kj}(\boldsymbol{U}_t) + \sigma_i\sigma_j\beta_{ij}(\boldsymbol{U}_t) \right) \right|$$

$$\overset{(a)}{\leq} \left( 1 + \eta\left( \sigma_i + \sigma_j + \eta\sigma_i\sigma_j + \mathcal{O}\left( d\alpha^{\frac{\gamma}{\sigma_1}} \right) \right) \right)|\beta_{ij}(\boldsymbol{U}_t)|.$$

Here in (a) we used the assumption $|\beta_{ij}(\boldsymbol{U}_t)| \lesssim \sigma_1\alpha^{\frac{2\sigma_1 - \sigma_i - \sigma_j}{2\sigma_1}} \lesssim \sigma_1\alpha^{\frac{\gamma}{2\sigma_1}}, \forall(i,j) \neq (1,1)$. Hence, by Lemma 4, we have

$$|\beta_{ij}(\boldsymbol{U}_{t+1})| \lesssim \sigma_1\alpha^{\frac{2\sigma_1 - \sigma_i - \sigma_j}{2\sigma_1}}, \quad \text{where } 1 \leq i \leq r \text{ or } 1 \leq j \leq r. \tag{68}$$

Case II: $i, j \geq r + 1$. For $\beta_{ij}(\boldsymbol{U}_t)$ such that $i, j \geq r + 1$, its dynamic is characterized by

$$|\beta_{ij}(\boldsymbol{U}_{t+1})| \leq (1 - \eta(2\beta_{ii}(\boldsymbol{U}_t) + 2\beta_{jj}(\boldsymbol{U}_t)))|\beta_{ij}(\boldsymbol{U}_t)| + 2\eta \sum_{k \neq i,j} |\beta_{ik}(\boldsymbol{U}_t)\beta_{jk}(\boldsymbol{U}_t)|$$

$$+ \eta^2 \sum_{k,l} |\beta_{ik}(\boldsymbol{U}_t)\beta_{kl}(\boldsymbol{U}_t)\beta_{lj}(\boldsymbol{U}_t)| \tag{69}$$

$$\leq \left( 1 + \eta\mathcal{O}\left( d\alpha^{\frac{\gamma}{\sigma_1}} \right) \right)|\beta_{ij}(\boldsymbol{U}_t)|.$$

Hence, for $t \lesssim \frac{1}{\eta\sigma_1} \log\left( \frac{\sigma_1}{\alpha} \right)$, we have $|\beta_{ij}(\boldsymbol{U}_t)| \leq \left( 1 + \eta\mathcal{O}\left( d\alpha^{\frac{\gamma}{\sigma_1}} \right) \right)^{\mathcal{O}\left( \frac{1}{\eta\sigma_1} \log\left( \frac{\sigma_1}{\alpha} \right) \right)} |\beta_{ij}(\boldsymbol{U}_0)| \lesssim |\beta_{ij}(\boldsymbol{U}_0)|$ since we assume $\alpha \lesssim \left( \frac{\sigma_1}{d} \right)^{\sigma_1/\gamma}$. This completes our inductive proof for $|\beta_{ij}(\boldsymbol{U}_t)| \lesssim$

$\sigma_1 \alpha^{\frac{2\sigma_1 - \sigma_i - \sigma_j}{2\sigma_1}}, \forall (i,j) \neq (1,1)$ in the second phase. Finally, we turn to $\beta_{11}(\boldsymbol{U}_t)$. One can write

$$
\begin{aligned}
\beta_{11}(\boldsymbol{U}_{t+1}) &= (1 + 2\eta (\sigma_1 - \beta_{11}(\boldsymbol{U}_t))) \beta_{11}(\boldsymbol{U}_t) - 2\eta \sum_{j \neq 1} \beta_{1j}^2(\boldsymbol{U}_t) \\
&\quad + \eta^2 \left( \sum_{j,k} \beta_{1j}(\boldsymbol{U}_t) \beta_{1k}(\boldsymbol{U}_t) \beta_{jk}(\boldsymbol{U}_t) - 2\sigma_1 \sum_j \beta_{1j}^2(\boldsymbol{U}_t) + \sigma_1^2 \beta_{11}(\boldsymbol{U}_t) \right) \\
&\stackrel{(a)}{\geq} \left( 1 + 2\eta \left( \sigma_1 + 0.5\eta\sigma_1^2 - \beta_{11}(\boldsymbol{U}_t) - \mathcal{O}\left( d\alpha^{\frac{\gamma}{2\sigma_1}} \right) \right) \right) \beta_{11}(\boldsymbol{U}_t) \\
&\stackrel{(b)}{\geq} \left( 1 + 2\eta \left( 0.9995\sigma_1 + 0.5\eta\sigma_1^2 - \beta_{11}(\boldsymbol{U}_t) \right) \right) \beta_{11}(\boldsymbol{U}_t).
\end{aligned}
\tag{70}
$$

Here in (a) we used the fact that $|\beta_{ij}(\boldsymbol{U}_t)| \lesssim \sigma_1 r' \log(d) \alpha^{\frac{2\sigma_1 - \sigma_i - \sigma_j}{2\sigma_1}} \lesssim \sigma_1 \alpha^{\frac{\gamma}{2\sigma_1}}, \quad \forall (i,j) \neq (1,1)$. In (b), we used the assumption that $\alpha \lesssim \frac{\sigma_r}{d}^{2\sigma_1/\gamma}$. The above inequality together with Lemma 3 entails that within $\frac{1}{\log(1+\eta\sigma_1)} \log\left( \frac{1600\sigma_1}{\alpha} \right) = \mathcal{O}\left( \frac{1}{\eta\sigma_1} \log\left( \frac{\sigma_1}{\alpha} \right) \right)$ iterations, we have $\beta_{11}(\boldsymbol{U}_t) \geq 0.99\sigma_1$. This completes the proof of Lemma 2 and marks the end of Substage 1. $\qquad \square$

Next, we move on to Substage 2.

**Substage** 2. In Substage 2, we show that the second component $\beta_{22}(\boldsymbol{U}_t)$ converges to $\sigma_2$ within $\mathcal{O}\left( \frac{1}{\eta\sigma_2} \log\left( \frac{\sigma_2}{\alpha} \right) \right)$ iterations while the other coefficients remain small. To this goal, we first study the one-step dynamic of $\beta_{22}(\boldsymbol{U}_t)$:

$$
\begin{aligned}
\beta_{22}(\boldsymbol{U}_{t+1}) &= (1 + 2\eta (\sigma_2 - \beta_{22}(\boldsymbol{U}_t))) \beta_{22}(\boldsymbol{U}_t) - 2\eta \sum_{j \neq 2} \beta_{2j}^2(\boldsymbol{U}_t) \\
&\quad + \eta^2 \left( \sum_{j,k} \beta_{2j}(\boldsymbol{U}_t) \beta_{2k}(\boldsymbol{U}_t) \beta_{jk}(\boldsymbol{U}_t) - 2\sigma_2 \sum_j \beta_{2j}^2(\boldsymbol{U}_t) + \sigma_2^2 \beta_{22}(\boldsymbol{U}_t) \right).
\end{aligned}
\tag{71}
$$

Different from the dynamic of $\beta_{11}(\boldsymbol{U}_t)$, not all the coefficients $\beta_{ij}(\boldsymbol{U}_t)$ with $i = 2$ or $j = 2$ are smaller than $\beta_{22}(\boldsymbol{U}_t)$ at the beginning of Substage 2. In particular, the basis coefficient $|\beta_{12}(\boldsymbol{U}_t)|$ may be much larger than $\beta_{22}(\boldsymbol{U}_t)$ at the beginning of Substage 2. To see this, note that, according to Equation 68, we have $\beta_{12}(\boldsymbol{U}_t) \lesssim \alpha^{\frac{\sigma_1 - \sigma_2}{2\sigma_1}}$ and $\beta_{22}(\boldsymbol{U}_t) \lesssim \alpha^{\frac{2(\sigma_1 - \sigma_2)}{2\sigma_1}}$. Hence, it may be possible to have $|\beta_{12}(\boldsymbol{U}_t)| \asymp \sqrt{\beta_{22}(\boldsymbol{U}_t)} \gg \beta_{22}(\boldsymbol{U}_t)$. Therefore, the term $2\eta\beta_{12}^2(\boldsymbol{U}_t)$ in Equation 76 must be handled with extra care. Note that if we can show $\sigma_2\beta_{22}(\boldsymbol{U}_t) \gg \beta_{12}^2(\boldsymbol{U}_t)$, then $2\eta\beta_{12}^2(\boldsymbol{U}_t)$ can be combined with the first term in the RHS of Equation 76 and the argument made in Substage 1 can be repeated to complete the proof of Substage 2. However, our provided bound in Equation 68 can only imply $\beta_{12}(\boldsymbol{U}_t)^2 \asymp \beta_{22}(\boldsymbol{U}_t)$. Therefore, we need to provide a tighter analysis to show that $\beta_{12}^2(\boldsymbol{U}_t) \ll \sigma_2\beta_{22}(\boldsymbol{U}_t)$ along the trajectory. Upon controlling $\beta_{12}^2(\boldsymbol{U}_t)$, we can then show the convergence of $\beta_{22}(\boldsymbol{U}_t)$ similar to our analysis for $\beta_{11}(\boldsymbol{U}_t)$ in Substage 1.

To control the behavior of $\beta_{12}^2(\boldsymbol{U}_t)$, we study the ratio $\omega(t) := \frac{\beta_{12}^2(\boldsymbol{U}_t)}{\beta_{22}(\boldsymbol{U}_t)}$. We will show that $\omega(t) \ll \sigma_2$ along the trajectory. To this goal, we will show that $\omega(t)$ can only increase for $\mathcal{O}(\frac{1}{\eta\sigma_1} \log(\frac{\sigma_1}{\alpha}))$ iterations. Therefore, its maximum along the solution trajectory happens at $T = \mathcal{O}(\frac{1}{\eta\sigma_1} \log(\frac{\sigma_1}{\alpha}))$. Therefore, by bounding the maximum, we can show that $\omega(t)$ remains small throughout the solution trajectory.

First, at the initial point, we have $\omega(0) \leq 64 \log^2(d) r' \alpha^2$, which satisfies our claim. We next provide an upper bound for $\omega(t+1)$ based on $\omega(t)$. Note that

$$
\begin{aligned}
\omega(t+1) &\leq \frac{(1 + \eta(\sigma_1 + \sigma_2 - 2\beta_{11}(\boldsymbol{U}_t) + \mathrm{poly}(\alpha)))^2}{(1 + \eta(\sigma_2 - \mathrm{poly}(\alpha)))^2} \cdot \frac{\beta_{12}^2(\boldsymbol{U}_t)}{\beta_{22}(\boldsymbol{U}_t)} \\
&= \left(1 + \frac{\eta(\sigma_1 - 2\beta_{11}(\boldsymbol{U}_t) + \mathrm{poly}(\alpha))}{1 + \eta(\sigma_2 - \mathrm{poly}(\alpha))}\right)^2 \omega(t) \\
&\leq \left(1 + \eta(\sigma_1 - 2\beta_{11}(\boldsymbol{U}_t)) - 0.5\eta^2 \sigma_1 \sigma_2\right)^2 \omega(t) \\
&\leq \left((1 + \eta\sigma_1)^2 - 2\eta\beta_{11}(\boldsymbol{U}_t) - 0.4\eta^2 \sigma_1 \sigma_2\right) \omega(t).
\end{aligned}
\tag{72}
$$

Due to the first inequality, $\omega(t)$ can be increasing only until $\beta_{11}(\boldsymbol{U}_t) \geq \frac{\sigma_1}{2} \pm \mathrm{poly}(\alpha)$. On the other hand, due to the dynamic of $\beta_{11}$ in Substage 1, we can show that $\beta_{11}(\boldsymbol{U}_t) \geq \frac{\sigma_1}{2} \pm \mathrm{poly}(\alpha)$ in at most $T = \mathcal{O}\left(\frac{1}{\eta\sigma_1} \log\left(\frac{\sigma_1}{\alpha}\right)\right)$ iterations. Therefore, $\omega(t)$ takes its maximum at $T = \mathcal{O}\left(\frac{1}{\eta\sigma_1} \log\left(\frac{\sigma_1}{\alpha}\right)\right)$. On the other hand, we know that $\beta_{11}(\boldsymbol{U}_t)$ satisfies

$$
\beta_{11}(\boldsymbol{U}_{t+1}) = \left((1 + \eta\sigma_1)^2 - 2\eta\beta_{11}(\boldsymbol{U}_t) \pm \eta\, \mathrm{poly}(\alpha)\right) \beta_{11}(\boldsymbol{U}_t).
\tag{73}
$$

Hence, we can bound $\omega(t)$ as

$$
\begin{aligned}
\omega(T) &\leq \prod_{t=0}^{T-1} \left((1 + \eta\sigma_1)^2 - 2\eta\beta_{11}(\boldsymbol{U}_t) - 0.4\eta^2 \sigma_1 \sigma_2\right) \omega(0) \\
&\leq \prod_{t=0}^{T-1} \frac{(1 + \eta\sigma_1)^2 - 2\eta\beta_{11}(\boldsymbol{U}_t) - 0.4\eta^2 \sigma_1 \sigma_2}{(1 + \eta\sigma_1)^2 - 2\eta\beta_{11}(\boldsymbol{U}_t) \pm \eta\, \mathrm{poly}(\alpha)} \cdot \frac{\beta_{11}(\boldsymbol{U}_{t+1})}{\beta_{11}(\boldsymbol{U}_t)} \omega(0) \\
&\overset{(a)}{=} \prod_{t=0}^{T-1} (1 - \Omega(\eta^2 \sigma_1 \sigma_2)) \frac{\beta_{11}(\boldsymbol{U}_{t+1})}{\beta_{11}(\boldsymbol{U}_t)} \omega(0) \\
&\leq 256(1 - \Omega(\eta^2 \sigma_1 \sigma_2))^T \beta_{11}(\boldsymbol{U}_T) \log^2(d).
\end{aligned}
\tag{74}
$$

Here in (a) we used the fact that $\alpha \lesssim \left(\eta\sigma_r^2\right)^{\sigma_1/\gamma}$. Hence, we have

$$
\omega(T) \lesssim (1 - \Omega(\eta^2 \sigma_1 \sigma_2))^T \sigma_1 \log^2(d) \lesssim \left(\frac{\alpha}{\sigma_1}\right)^{\Omega(\eta\sigma_2)} \sigma_1 \log^2(d).
\tag{75}
$$

Due to our assumption $\alpha \lesssim \left(\kappa \log^2(d)\right)^{-\Omega\left(\frac{1}{\eta\sigma_r}\right)}$, we conclude that $\omega(t) \leq 0.01\sigma_2$. Therefore, equation 76 can be lower bounded as

$$
\begin{aligned}
\beta_{22}(\boldsymbol{U}_{t+1}) &\geq (1 + 1.99\eta(\sigma_2 - \beta_{22}(\boldsymbol{U}_t))) \beta_{22}(\boldsymbol{U}_t) - 2\eta \sum_{j>2} \beta_{2j}^2(\boldsymbol{U}_t) \\
&\quad + \eta^2 \left( \sum_{j,k} \beta_{2j}(\boldsymbol{U}_t) \beta_{2k}(\boldsymbol{U}_t) \beta_{jk}(\boldsymbol{U}_t) - 2\sigma_2 \sum_j \beta_{2j}^2(\boldsymbol{U}_t) + \sigma_2^2 \beta_{22}(\boldsymbol{U}_t) \right).
\end{aligned}
\tag{76}
$$

The rest of the proof is a line by line reconstruction of Substage 1 and hence omitted for brevity.

**Substage** $3 \leq k \leq r$. Via an identical argument to Substage 2, we can show that for each Substage $3 \leq k \leq r$, we have

$$
0.99\sigma_k \leq \beta_{kk}(\boldsymbol{U}_t) \leq \sigma_k.
\tag{77}
$$

within $T_k = \mathcal{O}\left(\frac{1}{\eta\sigma_k} \log\left(\frac{\sigma_1}{\alpha}\right)\right)$ iterations. This completes the proof of the first statement of Proposition 3.

To prove the second statement of Proposition 3 as well as Theorem 3, we next control the residual terms. First, we consider the residual term $\beta_{ij}(\boldsymbol{U}_t)$ where either $i \leq r$ or $j \leq r$. Note that $\beta_{ij}(\boldsymbol{U}_t) = \beta_{ji}(\boldsymbol{U}_t)$ and hence we can assume $i \leq r$ without loss of generality. We will show that $\beta_{ij}(\boldsymbol{U}_t)$ decreases linearly once the corresponding signal $\beta_{ii}(\boldsymbol{U}_t)$ converges to the vicinity of

$\sigma_i$. To this goal, it suffices to control the largest component $\beta_{\max}(U_t) := \max_{i \neq j, i \leq r} |\beta_{ij}(U_t)|$. Without loss of generality, we assume that the index $(i, j)$ attains the maximum at time $t$, i.e., $\beta_{\max}(U_t) = |\beta_{ij}(U_t)|$. One can write

$$
\begin{aligned}
&|\beta_{ij}(\boldsymbol{U}_{t+1})| \\
&\leq (1 + \eta (\sigma_i + \sigma_j - 2\beta_{ii}(\boldsymbol{U}_t) - 2\beta_{jj}(\boldsymbol{U}_t))) |\beta_{ij}(\boldsymbol{U}_t)| + 2\eta \sum_{k \neq i, j} |\beta_{ik}(\boldsymbol{U}_t)\beta_{jk}(\boldsymbol{U}_t)| \\
&\quad + \eta^2 \left| \left( \sum_{k,l} \beta_{ik}(\boldsymbol{U}_t)\beta_{kl}(\boldsymbol{U}_t)\beta_{lj}(\boldsymbol{U}_t) - (\sigma_i + \sigma_j) \sum_k \beta_{ik}(\boldsymbol{U}_t)\beta_{kj}(\boldsymbol{U}_t) + \sigma_i\sigma_j\beta_{ij}(\boldsymbol{U}_t) \right) \right| \quad (78) \\
&\overset{(a)}{\leq} (1 - \eta 0.9(\sigma_i + \sigma_j))|\beta_{ij}(\boldsymbol{U}_t)| \\
&\leq (1 - \eta 0.9\sigma_r)|\beta_{ij}(\boldsymbol{U}_t)|.
\end{aligned}
$$

Here in (a) we used the fact that $0.99\sigma_i \leq \beta_{ii}(U_t) \leq \sigma_i, 1 \leq i \leq r$ and the fact that $\beta_{\max}(U_t) = |\beta_{ij}(U_t)|$. Hence, we conclude that $\beta_{\max}(U_{t+1}) \leq (1 - 0.9\eta\sigma_r)\beta_{\max}(U_t)$. Therefore, within additional $\mathcal{O}\left(\frac{1}{\eta\sigma_r} \log\left(\frac{1}{\alpha}\right)\right)$ iterations, we have $|\beta_{ij}(U_t)| \leq r' \log(d)\alpha^2$ for all $(i, j)$ such that $i \leq r, i \neq j$.

The remaining residual terms, i.e., those coefficients $\beta_{ij}(U_t)$ for which $i, j > r$, can be bounded via the same approach in Case II of substage 1. In particular, we can show that $|\beta_{ij}(U_t)| \lesssim |\beta_{ij}(U_0)| \lesssim r' \log(d)\alpha^2, \forall i, j > r$. For brevity, we omit this step. This completes the proof of the second statement of Proposition 3.

Finally, to prove Theorem 3, we show that once $|\beta_{ij}(U_t)| \leq r' \log(d)\alpha^2, \forall (i, j) \neq (k, k), 1 \leq k \leq r$, the signals $\beta_{kk}(U_t)$ will further converge to $\sigma_k \pm \mathcal{O}\left(\alpha^2\right)$ within $\mathcal{O}\left(\frac{1}{\eta\sigma_k} \log\left(\frac{\sigma_k}{\alpha}\right)\right)$ iterations. To see this, we simplify the dynamic of $\beta_{kk}(U_t)$ as

$$
\begin{aligned}
\beta_{kk}(\boldsymbol{U}_{t+1}) &= (1 + 2\eta (\sigma_k - \beta_{kk}(\boldsymbol{U}_t))) \beta_{kk}(\boldsymbol{U}_t) - 2\eta \sum_{j \neq k} \beta_{kj}^2(\boldsymbol{U}_t) \\
&\quad + \eta^2 \left( \sum_{j,l} \beta_{kj}(\boldsymbol{U}_t)\beta_{kl}(\boldsymbol{U}_t)\beta_{jl}(\boldsymbol{U}_t) - 2\sigma_k \sum_j \beta_{kj}^2(\boldsymbol{U}_t) + \sigma_k^2\beta_{kk}(\boldsymbol{U}_t) \right) \quad (79) \\
&= (1 + \eta (\beta_{kk}(\boldsymbol{U}_t) - \sigma_k))^2 \beta_{kk}(\boldsymbol{U}_t) - \mathcal{O}\left(\eta dr'^2 \log^2(d)\alpha^4\right),
\end{aligned}
$$

which leads to

$$
\begin{aligned}
\sigma_k - \beta_{kk}(\boldsymbol{U}_{t+1}) &= (1 - \eta\beta_{kk}(\boldsymbol{U}_t)(2 + \eta(\sigma_k - \beta_{kk}(\boldsymbol{U}_t))))(\sigma_k - \beta_{kk}(\boldsymbol{U}_t)) + \mathcal{O}\left(\eta dr'^2 \log^2(d)\alpha^4\right) \\
&\leq (1 - 1.98\eta\sigma_k)(\sigma_k - \beta_{kk}(\boldsymbol{U}_t)) + \mathcal{O}\left(\eta dr'^2 \log^2(d)\alpha^4\right).
\end{aligned}
$$
(80)

Hence, within additional $\mathcal{O}\left(\frac{1}{\eta\sigma_k} \log\left(\frac{\sigma_k}{\alpha}\right)\right)$ iterations, we have $|\sigma_k - \beta_{kk}(U_{t+1})| = \mathcal{O}\left(\frac{1}{\sigma_k} dr'^2 \log^2(d)\alpha^4\right)$ for every $1 \leq k \leq r$.

In conclusion, we have

$$
\left\| \boldsymbol{U}_T\boldsymbol{U}_T^\top - \boldsymbol{M}^\star \right\|_F^2 = \sum_{i,j=1}^d \left( \beta_{ij}(\boldsymbol{U}_T) - \beta_{ij}^\star \right)^2 \leq r'^2 \log^2(d)d^2\alpha^4. \quad (81)
$$

within $\mathcal{O}\left(\frac{1}{\eta\sigma_r} \log\left(\frac{\sigma_r}{\alpha}\right)\right)$ iterations. This completes the proof of Theorem 3. $\qquad\square$

## F.4 ANALYSIS OF THE LOGISTIC MAP

In this section, we provide the proofs of Lemmas 3 and 4.

UPPER BOUND OF ITERATION COMPLEXITY

Recall the logistic map

$$x_{t+1} = (1 + \eta\sigma - \eta x_t)x_t, \quad x_0 = \alpha. \tag{82}$$

Here the initial value satisfies $0 < \alpha \ll \sigma$. Vaskevicius et al. (2019) provide both upper and lower bounds for $x_t$ that follows the above logistic map. However, their bounds are not directly applicable to our setting. Hence, we need to develop a new proof for Lemma 3.

*Proof of Lemma 3.* We divide the dynamic into two stages: (a) $x_t \leq \frac{1}{2}\sigma$, and (b) $x_t \geq \frac{1}{2}\sigma$.

**Stage 1:** $x_t \leq \frac{1}{2}\sigma$. We consider $K = \lceil \log\left(\frac{1}{2}\sigma/\alpha\right)\rceil$ substages, where in each Substage $k$, we have $\alpha e^k \leq x_t \leq \alpha e^{k+1}$. Suppose that $t_k$ is the number of iterations in Substage $k$. One can write

$$\begin{aligned} x_{t+1} &= (1 + \eta\sigma - \eta x_t)x_t \\ &\geq (1 + \eta\sigma - \eta\alpha e^{k+1})x_t. \end{aligned} \tag{83}$$

Hence, it suffices to find the smallest $t = t_{\min}$ such that

$$\alpha e^k(1 + \eta\sigma - \eta\alpha e^{k+1})^t \geq \alpha e^{k+1}. \tag{84}$$

Solving this inequality leads to

$$t_{\min} = \frac{1}{\log(1 + \eta\sigma - \eta\alpha e^{k+1})}. \tag{85}$$

Based on the above equality, we provide an upper bound for $t_{\min}$:

$$\begin{aligned} t_{\min} &= \frac{1}{\log(1 + \eta\sigma) + \log(1 - \eta\alpha e^{k+1}/(1 + \eta\sigma))} \\ &\overset{(a)}{\leq} \frac{1}{\log(1 + \eta\sigma) - \frac{\eta\alpha e^{k+1}}{1 + \eta\sigma - \eta\alpha e^{k+1}}} \\ &\leq \frac{1 + \frac{\eta\alpha e^{k+1}}{1 + \eta\sigma - \eta\alpha e^{k+1}}}{\log(1 + \eta\sigma)} \\ &\overset{(b)}{\leq} \frac{1 + \eta\alpha e^{k+1}}{\log(1 + \eta\sigma)}. \end{aligned} \tag{86}$$

Here in (a) we used the fact that $\log(1 + x) \leq \frac{x}{1+x}, \forall x > -1$ and in (b) we used the fact that $x_t \leq \frac{1}{2}\sigma$. Hence, we have $t_k \leq \frac{1 + \eta\alpha e^{k+1}}{\log(1+\eta\sigma)}$. Therefore, the total iteration complexity of Stage 1 is upper bounded by

$$T_1 = \sum_{k=0}^{K-1} t_k \leq \sum_{k=0}^{K-1} \frac{1 + \eta\alpha e^{k+1}}{\log(1 + \eta\sigma)} \leq \frac{\lceil\log\left(\frac{1}{2}\sigma/\alpha\right)\rceil + \eta\sigma}{\log(1 + \eta\sigma)} \leq \frac{\log(4\sigma/\alpha)}{\log(1 + \eta\sigma)}. \tag{87}$$

**Stage 2:** $x_t \geq \frac{1}{2}\sigma$. In this stage, we rewrite the equation 82 as

$$\sigma - x_{t+1} = (1 - \eta x_t)(\sigma - x_t). \tag{88}$$

Via a similar trick, we can show that within additional $T_2 = \frac{\log(4\sigma/\varepsilon)}{\log(1+\eta\sigma)}$ iterations, we have $x_t \geq \sigma - \varepsilon$, which can be achieved in the total number of iterations $T = T_1 + T_2 = \frac{1}{\log(1+\eta\sigma)}\left(\log(4\sigma/\alpha) + \log(4\sigma/\varepsilon)\right)$ iterations. This completes the proof of Lemma 3 $\qquad\square$

SEPARATION BETWEEN TWO INDEPENDENT SIGNALS

In this section, we show that there is a sharp separation between two logistic maps with signals $\sigma_1, \sigma_2$ provided that $\sigma_1 \neq \sigma_2$. In particular, suppose that $\sigma_1 - \sigma_2 \geq \gamma > 0$ and

$$\begin{aligned} x_{t+1} &= (1 + \eta\sigma_1 - \eta x_t)x_t, \quad x_0 = \alpha, \\ y_{t+1} &= (1 + \eta\sigma_2 - \eta y_t)y_t, \quad y_0 = \alpha. \end{aligned} \tag{89}$$

*Proof of Lemma 4.* By Lemma 3, we have $x_T \geq \sigma_1 - \varepsilon$ within $T = \frac{1}{\log(1+\eta\sigma_1)} \log\left(\frac{16\sigma_1^2}{\varepsilon\alpha}\right)$ iterations. Therefore, it suffices to show that $y_t$ remains small for $t \leq T$. To this goal, note that

$$y_{t+1} = (1 + \eta\sigma_2 - \eta y_t)y_t \leq (1 + \eta\sigma_2)y_t \leq (1 + \eta\sigma_2)^{t+1}\alpha. \tag{90}$$

Hence, we need to bound $\Gamma = (1 + \eta\sigma_2)^T$. Taking logarithm of both sides, we have

$$\begin{aligned}
\log(\Gamma) &= T\log(1 + \eta\sigma_2) \\
&= \log\left(\frac{16\sigma_1^2}{\varepsilon\alpha}\right)\frac{\log(1 + \eta\sigma_2)}{\log(1 + \eta\sigma_1)}.
\end{aligned} \tag{91}$$

Now, we provide a lower bound for the ratio $\log(1 + \eta\sigma_1)/\log(1 + \eta\sigma_2)$:

$$\begin{aligned}
\frac{\log(1 + \eta\sigma_1)}{\log(1 + \eta\sigma_2)} &= 1 + \frac{\log(1 + \frac{\eta(\sigma_1 - \sigma_2)}{1+\eta\sigma_2})}{\log(1 + \eta\sigma_2)} \\
&\geq 1 + \frac{\frac{\eta(\sigma_1 - \sigma_2)}{1+\eta\sigma_2} / \left(1 + \frac{\eta(\sigma_1 - \sigma_2)}{1+\eta\sigma_2}\right)}{\eta\sigma_2} \\
&\overset{(a)}{\geq} 1 + \frac{\sigma_1 - \sigma_2}{2\sigma_2} \\
&= \frac{\sigma_1 + \sigma_2}{2\sigma_2},
\end{aligned} \tag{92}$$

where (a) follows from the assumption $\eta \leq \frac{1}{4\sigma_1}$. Therefore, we have

$$\Gamma = \exp\left\{\log\left(\frac{16\sigma_1^2}{\varepsilon\alpha}\right)\frac{2\sigma_2}{\sigma_1 + \sigma_2}\right\} = \left(\frac{16\sigma_1^2}{\varepsilon\alpha}\right)^{2\sigma_2/(\sigma_1+\sigma_2)}. \tag{93}$$

which implies that

$$y_T \leq \alpha\left(\frac{16\sigma_1^2}{\varepsilon\alpha}\right)^{2\sigma_2/(\sigma_1+\sigma_2)} \leq \frac{16\sigma_1^2}{\varepsilon}\alpha^{\frac{\sigma_1-\sigma_2}{\sigma_1+\sigma_2}}. \tag{94}$$

This completes the proof of Lemma 4. $\qquad\square$

## G   PROOF FOR TENSOR DECOMPOSITION

In this section, we prove our results for the orthonormal symmetric tensor decomposition (OSTD). Different from matrix factorization, we use a special initialization that aligns with the ground truth. In particular, for all $1 \leq i \leq r'$, we assume that $\sin(\boldsymbol{u}_i(0), \boldsymbol{z}_i) \leq \gamma$ for some small $\gamma$. We will show that $\boldsymbol{u}_i(t)$ aligns with $\boldsymbol{z}_i$ along the whole optimization trajectory. To this goal, we define $v_{ij}(t) = \langle \boldsymbol{u}_i(t), \boldsymbol{z}_j \rangle$ for every $1 \leq i \leq r'$ and $1 \leq j \leq d$. Recall that $\Lambda$ is a multi-index with length $l$. We define $|\Lambda|_k$ as the number of times index $k$ appears as one of the elements of $\Lambda$.[7] Evidently, we have $0 \leq |\Lambda|_k \leq l$. Based on these definitions, one can write

$$\beta_\Lambda(\boldsymbol{U}) = \sum_{i=1}^{r'}\prod_{k=1}^{d}\langle\boldsymbol{u}_i, \boldsymbol{z}_k\rangle^{|\Lambda|_k} = \sum_{i=1}^{r'}\prod_{k=1}^{d}v_{ik}^{|\Lambda|_k}. \tag{95}$$

Now, it suffices to study the dynamic of $v_{ij}(t)$. In particular, we will show that $v_{ij}(t)$ remains small except for the top-$r$ diagonal elements $v_{jj}(t), 1 \leq j \leq r$, which will approach $\sigma_j^{1/l}$. To make this intuition more concrete, we divide the terms $\{v_{ij}(t)\}$ into three parts:

- **signal terms** defined as $v_{jj}(t), 1 \leq j \leq r$,
- **diagonal residual terms** defined as $v_{jj}(t), r + 1 \leq j \leq d$, and
- **off-diagonal residual terms** defined as $v_{ij}(t), \forall i \neq j$.

---

[7]For instance, assume that $\Lambda = (1, 1, 2)$. Then, $|\Lambda|_1 = 2$ and $|\Lambda|_3 = 0$.

Moreover, we define $V(t) = \max_{i \neq j} |v_{ij}(t)|$ as the maximum element of the off-diagonal residual terms at every iteration $t$. When there is no ambiguity, we will omit the dependence on iteration $t$. For example, we write $v_{ij} = v_{ij}(t)$ and $V = V(t)$. Similarly, when there is no ambiguity, we write $\beta_\Lambda(t)$ or $\beta_\Lambda$ in lieu of $\beta_\Lambda(\boldsymbol{U}(t))$.

Our next lemma characterizes the relationship between $\beta_\Lambda$ and $v_{ij}$.

**Lemma 5.** *Suppose that* $\max_{j \geq r+1} \left| v_{jj}^l \right| \lesssim \sigma_1^{\frac{l-1}{l}} V$, *and* $V \leq \sigma_1^{1/l} d^{-\frac{1}{l-1}}$. *Then,*

- *For* $\beta_{\Lambda_j}$ *with* $\Lambda_j = (j, \ldots, j)$, *we have*

$$\left| \beta_{\Lambda_j} - v_{jj}^l \right| \leq r' V^l. \tag{96}$$

- *For* $\beta_\Lambda$ *with at least two different indices in* $\Lambda$, *we have*

$$|\beta_\Lambda| \leq 2r \sigma_1^{\frac{l-1}{l}} V. \tag{97}$$

The proof of this lemma is deferred to Appendix G.1. Lemma 5 reveals that the magnitude of $\beta_\Lambda$ can be upper bounded by $\max_{i \neq j} |v_{ij}|$. Next, we control $v_{ij}$ by providing both lower and upper bounds on its dynamics.

**Proposition 7** (One-step dynamics for $v_{ij}(t)$). *Suppose that we have* $V(t) \lesssim \sigma_1^{1/l} d^{-\frac{1}{l-1}}$ *and* $v_{ii}(t) \leq \sigma_1^{1/l}, \forall 1 \leq i \leq d$. *Moreover, suppose that the step-size satisfies* $\eta \lesssim \frac{1}{l\sigma_1}$. *Then,*

- *For the signal term* $v_{ii}(t), 1 \leq i \leq r$, *we have*

$$v_{ii}(t+1) \geq v_{ii}(t) + \eta l \left( \sigma_i - v_{ii}^l(t) - 2d^{l-1} v_{ii}^{l-2}(t) V^2(t) \right) v_{ii}^{l-1}(t) - ld^l \eta \sigma_1^{\frac{l-1}{l}} V^l(t). \tag{98}$$

- *For the diagonal residual term* $v_{ii}(t), r+1 \leq i \leq d$, *we have*

$$v_{ii}(t+1) \leq v_{ii}(t) - \eta l v_{ii}^{2l-1}(t) + 2\eta l d^l \sigma_1^{\frac{l-1}{l}} V^l(t). \tag{99}$$

- *For the off-diagonal term* $V(t)$, *we have*

$$V(t+1) \leq V(t) + 3\eta l \sigma_1 V(t)^{l-1}. \tag{100}$$

The proof of this proposition is deferred to Appendix G.2. Equipped with the above one-step dynamics, we next provide a bound on the growth rate of $v_{ij}$.

**Proposition 8.** *Suppose that the initial point satisfies* $\sin(\boldsymbol{u}_i(0), \boldsymbol{z}_i) \leq \gamma$ *and* $\|\boldsymbol{u}_i(0)\| = \alpha^{1/l}$ *with* $\alpha \lesssim \frac{1}{d^{l^3}}, \gamma \lesssim \frac{1}{l\kappa}^{\frac{l}{l-2}}$. *Moreover, suppose that the step-size satisfies* $\eta \lesssim \frac{1}{l\sigma_1}$. *Then, within* $t^\star = \frac{8}{\eta l \sigma_r} \alpha^{-\frac{l-2}{l}}$ *iterations,*

- *For the signal term* $v_{ii}(t), 1 \leq i \leq r$, *we have*

$$\left| v_{ii}^l(t^\star) - \sigma_i \right| \leq 8d^{l-1} \sigma_i^{\frac{l-2}{l}} \alpha^{2/l} \gamma^2. \tag{101}$$

- *For the diagonal residual term* $v_{ii}(t), r+1 \leq i \leq d$, *we have*

$$|v_{ii}(t^*)| \leq 2\alpha^{1/l}. \tag{102}$$

- *For the off-diagonal term* $V(t)$, *we have*

$$V(t^*) \leq 2^{1/l} V(0) \leq (2\alpha)^{1/l} \gamma. \tag{103}$$

The proof of this proposition is deferred to Appendix G.3. With the above proposition, we are ready to prove Theorem 4.

*Proof of Thereom 4.* We have the following decomposition

$$\|\mathbf{T} - \mathbf{T}^\star\|_F^2 = \sum_\Lambda (\beta_\Lambda - \beta_\Lambda^\star)^2 . \tag{104}$$

Hence, it suffices to bound each $|\beta_\Lambda - \beta_\Lambda^\star|$. Combining Lemma 5 and Proposition 8, we have for every $\left|\beta_{\Lambda_j} - \beta_{\Lambda_j}^\star\right|, 1 \le j \le r$

$$
\begin{aligned}
\left|\beta_{\Lambda_j}(t^\star) - \beta_{\Lambda_j}^\star\right| &\overset{\text{Lemma 5}}{\le} \left|v_{jj}^l(t^\star) - \sigma_j\right| + r' V^l(t^\star) \\
&\overset{\text{Proposition 8}}{\le} 8 d^{l-1} \sigma_j^{\frac{l-2}{l}} \alpha^{2/l} \gamma^2 + 2 d \alpha \gamma^l \\
&\le 16 d^{l-1} \sigma_j^{\frac{l-2}{l}} \alpha^{2/l} \gamma^2,
\end{aligned}
\tag{105}
$$

where in the last inequality, we used $\sigma_j \ge 1$, $\alpha \le 1$, and $\gamma \le 1$. For the remaining diagonal elements $\beta_{\Lambda_j}, r+1 \le j \le d$, we have

$$
\begin{aligned}
\left|\beta_{\Lambda_j}(t^\star)\right| &\overset{\text{Lemma 5}}{\le} \left|v_{jj}^l(t^\star)\right| + r' V^l(t^\star) \\
&\overset{\text{Proposition 8}}{\le} 2^l \alpha + 2 d \alpha \gamma^l.
\end{aligned}
\tag{106}
$$

For the general $\beta_\Lambda$ with at least two different indices in the multi-index $\Lambda$, we have

$$
\begin{aligned}
|\beta_\Lambda(t^\star)| &\overset{\text{Lemma 5}}{\le} 2 r \sigma_1^{\frac{l-1}{l}} V(t^\star) \\
&\overset{\text{Proposition 8}}{\le} 4 r \sigma_1^{\frac{l-1}{l}} \alpha^{1/l} \gamma.
\end{aligned}
\tag{107}
$$

Hence, we conclude

$$
\begin{aligned}
\|\mathbf{T}(t^\star) - \mathbf{T}^\star\|_F^2 &\le \sum_{i=1}^r 16 d^{l-1} \sigma_i^{\frac{l-2}{l}} \alpha^{2/l} \gamma^2 + \sum_{i=r+1}^d \left(2^l \alpha + 2 d \alpha \gamma^l\right) + d^l \cdot 4 r \sigma_1^{\frac{l-1}{l}} \alpha^{1/l} \gamma \\
&\le 8 r d^l \gamma \sigma_1^{\frac{l-1}{l}} \alpha^{1/l},
\end{aligned}
\tag{108}
$$

which completes the proof of the theorem. $\qquad\square$

## G.1 PROOF OF LEMMA 5

*Proof.* We first analyze $\beta_{\Lambda_j}$. Note that

$$\beta_{\Lambda_j} = \sum_{i=1}^{r'} \langle \boldsymbol{u}_i, \boldsymbol{z}_j \rangle^l = \sum_{i=1}^{r'} v_{ij}^l = v_{jj}^l + \sum_{i \ne j} v_{ij}^l . \tag{109}$$

Hence,

$$\left|\beta_{\Lambda_j} - v_{jj}^l\right| = \left|\sum_{i \ne j} v_{ij}^l\right| \le r' V^l , \tag{110}$$

where we used the definition of $V$. For general $\beta_\Lambda$ where there are at least two different elements in the multi-index $\Lambda$, we have

$$
\begin{aligned}
|\beta_\Lambda| &= \left|\sum_{j=1}^{r'} \prod_{k \in \Lambda} v_{jk}^{|\Lambda|_k}\right| \\
&\le \sum_{j=1}^{r'} \left|v_{jj}^{|\Lambda|_j} V^{l-|\Lambda|_j}\right| \\
&= \sum_{j=1}^r \left|v_{jj}^{|\Lambda|_j} V^{l-|\Lambda|_j}\right| + \sum_{j=r+1}^{r'} \left|v_{jj}^{|\Lambda|_j} V^{l-|\Lambda|_j}\right| \\
&\le r \sigma_1^{\frac{l-1}{l}} V + (r' - r) \left(\max_{j \ge r+1} \left|v_{jj}^l\right| + V^l\right) \\
&\le 2 r \sigma_1^{\frac{l-1}{l}} V,
\end{aligned}
\tag{111}
$$

where in the last inequality, we used the assumption that $V \lesssim \sigma_1^{1/l} d^{-\frac{1}{l-1}}$ and $\max_{j \geq r+1} |v_{jj}^l| \lesssim \sigma_1^{-\frac{l-1}{l}} V$. This completes the proof. $\qquad\square$

### G.2 PROOF OF PROPOSITION 7

In this section, we provide the proof for Proposition 7. For simplicity and whenever there is no ambiguity, we omit the iteration $t$ and show iteration $t+1$ with superscript '+'. For instance, we write $v_{ij} = v_{ij}(t)$ and $v_{ij}^+ = v_{ij}(t+1)$.

Recall that $v_{ij} = \langle \boldsymbol{u}_i, \boldsymbol{z}_j \rangle$. For simplicity, we denote $\mu = \sigma_1^{1/l}$. Hence, by our assumption, we have $v_{ii} \leq \mu, \forall 1 \leq i \leq r$. We first provide the exact dynamic of $v_{ij}$ in the following lemma.

**Lemma 6.** *The one-step dynamic of $v_{ij}$ takes the following form*

$$v_{ij}^+ = v_{ij} + \eta l (\sigma_j - v_{jj}^l) v_{ij}^{l-1} - \eta l \sum_{k \in [r'], k \neq j} v_{kj}^l v_{ij}^{l-1}$$
$$- \eta \sum_{s \in [l-1]} s \sum_{\Lambda : |\Lambda|_j = s} \beta_\Lambda \left( \prod_{k \in \Lambda, k \neq j} v_{ik}^{|\Lambda|_k} \right) v_{ij}^{s-1}. \tag{112}$$

*Proof.* Recall that $\mathcal{L} = \frac{1}{2} \sum_\Lambda (\beta_\Lambda - \beta_\Lambda^\star)^2$ and $\beta_\Lambda = \sum_{i=1}^{r'} \prod_{k \in \Lambda} \langle \boldsymbol{u}_i, \boldsymbol{z}_k \rangle^{|\Lambda|_k}$. Moreover, we have $\beta_{\Lambda_j}^\star = \sigma_j$ for $1 \leq j \leq r$, and $\beta_\Lambda^\star = 0$ otherwise. We first calculate

$$\nabla_{\boldsymbol{u}_i} \beta_\Lambda = \sum_{s \in \Lambda} |\Lambda|_s \left( \prod_{k \in \Lambda, k \neq s} \langle \boldsymbol{u}_i, \boldsymbol{z}_k \rangle^{|\Lambda|_k} \right) \langle \boldsymbol{u}_i, \boldsymbol{z}_s \rangle^{|\Lambda|_s - 1} \boldsymbol{z}_s$$
$$= \sum_{s \in \Lambda} |\Lambda|_s \left( \prod_{k \in \Lambda, k \neq s} v_{ik}^{|\Lambda|_k} \right) v_{is}^{|\Lambda|_s - 1} \boldsymbol{z}_s. \tag{113}$$

Hence, the partial derivative of $\mathcal{L}(t)$ with respect to $\boldsymbol{u}_i$ is

$$\nabla_{\boldsymbol{u}_i} \mathcal{L} = \sum_{\forall \Lambda} (\beta_\Lambda - \beta_\Lambda^\star) \nabla_{\boldsymbol{u}_i} \beta_\Lambda = \sum_\Lambda (\beta_\Lambda - \beta_\Lambda^\star) \sum_{s \in \Lambda} |\Lambda|_s \left( \prod_{k \in \Lambda, k \neq s} v_{ik}^{|\Lambda|_k} \right) v_{is}^{|\Lambda|_s - 1} \boldsymbol{z}_s.$$

Note that $\{\boldsymbol{z}_j\}_{j \in [d]}$ are unit orthogonal vectors. Hence, we have

$$\langle \nabla_{\boldsymbol{u}_i} \beta_\Lambda, \boldsymbol{z}_j \rangle = \begin{cases} |\Lambda|_j \left( \prod_{k \in \Lambda, k \neq j} v_{ik}^{|\Lambda|_k} \right) v_{ij}^{|\Lambda|_j - 1} & \text{if } j \in \Lambda \\ 0 & \text{if } j \notin \Lambda. \end{cases} \tag{114}$$

By the definition of $v_{ij}$, its update rule can be written in the following way

$$\begin{aligned} v_{ij}^+ &= \langle \boldsymbol{u}_i^+, \boldsymbol{z}_j \rangle \\ &\overset{(a)}{=} v_{ij} - \eta \langle \nabla_{\boldsymbol{u}_i} \mathcal{L}, \boldsymbol{z}_j \rangle \\ &= v_{ij} + \eta \sum_\Lambda (\beta_\Lambda^\star - \beta_\Lambda) \langle \nabla_{\boldsymbol{u}_i} \beta_\Lambda, \boldsymbol{z}_j \rangle \\ &\overset{(b)}{=} v_{ij} + \eta \sum_{\Lambda : |\Lambda|_j \geq 1} (\beta_\Lambda^\star - \beta_\Lambda) |\Lambda|_j \left( \prod_{k \in \Lambda, k \neq j} v_{ik}^{|\Lambda|_k} \right) v_{ij}^{|\Lambda|_j - 1} \\ &\overset{(c)}{=} v_{ij} + \eta \sum_{s \in [l]} \sum_{\Lambda : |\Lambda|_j = s} |\Lambda|_j (\beta_\Lambda^\star - \beta_\Lambda) \left( \prod_{k \in \Lambda, k \neq j} v_{ik}^{|\Lambda|_k} \right) v_{ij}^{s-1}. \end{aligned} \tag{115}$$

Here in $(a)$, we used the update rule for $\boldsymbol{u}_i$. In $(b)$, we applied equation 114 to exclude those $\Lambda$ without $j$. In $(c)$, we simply rearranged the above equation according to the cardinality $|\Lambda|_j$. We further isolate the term that only has $v_{ij}$:

$$
\begin{aligned}
v_{ij}^+ &\overset{(a)}{=} v_{ij} + \eta l(\sigma_j - \beta_{\Lambda_j})v_{ij}^{l-1} - \eta \sum_{s \in [l-1]} s \sum_{\Lambda:|\Lambda|_j=s} \beta_\Lambda \left( \prod_{k \in \Lambda, k \neq j} v_{ik}^{|\Lambda|_k} \right) v_{ij}^{s-1} \\
&\overset{(b)}{=} v_{ij} + \eta l(\sigma_j - v_{jj}^l)v_{ij}^{l-1} - \eta l \sum_{k \in \Lambda, k \neq j} v_{kj}^l v_{ij}^{l-1} \\
&\quad - \eta \sum_{s \in [l-1]} s \sum_{\Lambda:|\Lambda|_j=s} \beta_\Lambda \left( \prod_{k \in \Lambda, k \neq j} v_{ik}^{|\Lambda|_k} \right) v_{ij}^{s-1}.
\end{aligned}
\tag{116}
$$

Here in $(a)$, we rearranged terms and isolated the term with $|\Lambda|_j = l$. Note that the remaining terms must satisfy $1 \leq |\Lambda|_j \leq l-1$, which indicates that there must be at least 2 different indexes in $\Lambda$ which in turn implies $\beta_\Lambda^\star = 0$. In $(b)$, we used the definition of $\beta_{\Lambda_j}$. This completes the proof of Lemma 6. $\qquad\square$

Equipped with Lemma 6, we are ready to prove Proposition 7.

*Proof of Proposition 7.* The proof is divided into three parts:

**Signal Term:** $v_{ii}(t), 1 \leq i \leq r$

We first consider the signal terms $v_{ii}(t), 1 \leq j \leq r$. First, upon setting $i = j$ in Lemma 6, we have

$$
\begin{aligned}
v_{ii}^+ &= v_{ii} + \eta l \left( \sigma_i - v_{ii}^l - \sum_{k \neq i} v_{ki}^l \right) v_{ii}^{l-1} - \eta \sum_{s=1}^{l-1} s \sum_{\Lambda:|\Lambda|_i=s} \beta_\Lambda \left( \prod_{k \in \Lambda, k \neq i} v_{ik}^{|\Lambda|_k} \right) v_{ii}^{s-1} \\
&\geq v_{ii} + \eta l \left( \sigma_i - v_{ii}^l - mV^l \right) v_{ii}^{l-1} - \eta \sum_{s=1}^{l-1} s \sum_{\Lambda:|\Lambda|_i=s} \beta_\Lambda \left( \prod_{k \in \Lambda, k \neq i} v_{ik}^{|\Lambda|_k} \right) v_{ii}^{s-1}.
\end{aligned}
\tag{117}
$$

Now we aim to control $(A) = \beta_\Lambda \left( \prod_{k \in \Lambda, k \neq i} v_{ik}^{|\Lambda|_k} \right) v_{ii}^{s-1}$ for $|\Lambda|_i = s \in \{1, \ldots, l-1\}$. We have

$$
\begin{aligned}
(A) &= \sum_{j=1}^{r'} \left( \prod_{h \in \Lambda} v_{jh}^{|\Lambda|_h} \right) \left( \prod_{k \in \Lambda, k \neq i} v_{ik}^{|\Lambda|_k} \right) v_{ii}^{|\Lambda|_i - 1} \\
&\overset{(a)}{\leq} \sum_{j=1}^{r'} v_{jj}^{|\Lambda|_j} V^{l-|\Lambda|_j} V^{l-s} v_{ii}^{s-1} \\
&\overset{(b)}{\leq} v_{ii}^{2s-1} V^{2l-2s} + r' \max_{j \neq i} \left| v_{jj}^{|\Lambda|_j} V^{2l-s-|\Lambda|_j} v_{ii}^{s-1} \right| \\
&\overset{(c)}{\leq} v_{ii}^{2l-3} V^2 + r' \max_{j \neq i} \mu^{|\Lambda|_j + s - 1} V^{2l-s-|\Lambda|_j} \\
&\leq v_{ii}^{2l-3} V^2 + r' \mu^{l-1} V^l.
\end{aligned}
\tag{118}
$$

In $(a)$, we used the fact that $|v_{ij}| \leq V, \forall i \neq j$. In $(b)$, we isolated the term with $j = i$ and bounded the remaining terms with their maximum value. In $(c)$, we used the fact that $V \leq v_{ii} \leq \mu$.

After substituting equation 118 into equation 117, we have

$$
\begin{aligned}
v_{ii}^+ &\geq v_{ii} + \eta l \left( \sigma_i - v_{ii}^l - r'V^l \right) v_{ii}^{l-1} - \eta \sum_{s=1}^{l-1} s \sum_{\Lambda:|\Lambda|_i=s} \left( v_{ii}^{2l-3} V^2 + r' \mu^{l-1} V^l \right) \\
&\geq v_{ii} + \eta l \left( \sigma_i - v_{ii}^l - r'V^l \right) v_{ii}^{l-1} - \eta \left( v_{ii}^{2l-3} V^2 + r' \mu^{l-1} V^l \right) \sum_{s=1}^{l-1} s C_l^s (d-1)^{l-s},
\end{aligned}
\tag{119}
$$

where $C_l^s = \binom{l}{s}$. Note that $\sum_{s=1}^{l-1} s C_l^s (d-1)^{l-s} = ld^{l-1} - l \leq ld^{l-1}$. Therefore, we have

$$
\begin{aligned}
v_{ii}(t+1) &\geq v_{ii} + \eta l \left( \sigma_i - v_{ii}^l - r'V^l \right) v_{ii}^{l-1} - ld^{l-1}\eta \left( v_{ii}^{2l-3}V^2 + r'\mu^{l-1}V^l \right) \\
&\geq v_{ii} + \eta l \left( \sigma_i - v_{ii}^l - r'V^l - d^{l-1}v_{ii}^{l-2}V^2 \right) v_{ii}^{l-1} - ld^{l-1}\eta r'\mu^{l-1}V^l \\
&\geq v_{ii} + \eta l \left( \sigma_i - v_{ii}^l - 2d^{l-1}v_{ii}^{l-2}V^2 \right) v_{ii}^{l-1} - ld^{l-1}\eta r'\mu^{l-1}V^l \\
&\geq v_{ii} + \eta l \left( \sigma_i - v_{ii}^l - 2d^{l-1}v_{ii}^{l-2}V^2 \right) v_{ii}^{l-1} - ld^l\eta \sigma_1^{\frac{l-1}{l}} V^l,
\end{aligned}
\tag{120}
$$

where in the last inequality, we used the fact that $r' \leq d$.

**Diagonal Residual Term:** $v_{ii}, r + 1 \leq i \leq d$

In this case we consider the terms $v_{ii}$ with $r + 1 \leq i \leq d$, which is similar to the case $1 \leq i \leq r$. Without loss of generality, we assume that $v_{ii} \geq 0$. The case $v_{ii} \leq 0$ can be argued in an identical fashion. By equation 117, we have

$$
\begin{aligned}
v_{ii}^+ &= v_{ii} - \eta l \left( v_{ii}^l - \sum_{k \neq i} v_{ki}^l \right) v_{ii}^{l-1} - \eta \sum_{s=1}^{l-1} s \sum_{\Lambda:|\Lambda|_i=s} \beta_\Lambda \left( \prod_{k \in \Lambda, k \neq i} v_{ik}^{|\Lambda|_k} \right) v_{ii}^{s-1} \\
&\leq v_{ii} - \eta l \left( v_{ii}^l - r'V^l \right) v_{ii}^{l-1} - \eta \sum_{s=1}^{l-1} s \sum_{\Lambda:|\Lambda|_i=s} \beta_\Lambda \left( \prod_{k \in \Lambda, k \neq i} v_{ik}^{|\Lambda|_k} \right) v_{ii}^{s-1}.
\end{aligned}
\tag{121}
$$

For $(A) = \beta_\Lambda \left( \prod_{k \in \Lambda, k \neq i} v_{ik}^{|\Lambda|_k} \right) v_{ii}^{s-1}$, we further have

$$
\begin{aligned}
(A) &= \sum_{j=1}^{r'} \left( \prod_{h \in \Lambda} v_{jh}^{|\Lambda|_h} \right) \left( \prod_{k \in \Lambda, k \neq i} v_{ik}^{|\Lambda|_k} \right) v_{ii}^{s-1} \\
&\geq \left( \prod_{h \in \Lambda} v_{ih}^{|\Lambda|_h} \right) \left( \prod_{k \in \Lambda, k \neq i} v_{ik}^{|\Lambda|_k} \right) v_{ii}^{s-1} - \sum_{j \neq i} \left( \prod_{h \in \Lambda} v_{jh}^{|\Lambda|_h} \right) \left( \prod_{k \in \Lambda, k \neq i} v_{ik}^{|\Lambda|_k} \right) v_{ii}^{s-1} \\
&\geq \underbrace{\left( \prod_{k \in \Lambda, k \neq i} v_{ik}^{2|\Lambda|_k} \right) v_{ii}^{2s-1}}_{\geq 0} - r' \max_{j \neq i} \left| v_{jj}^{|\Lambda|_j} V^{2l-s-|\Lambda|_j} v_{ii}^{s-1} \right| \\
&\geq -r'\mu^{l-1}V^l.
\end{aligned}
\tag{122}
$$

Therefore, we obtain

$$
\begin{aligned}
v_{ii}^+ &\leq v_{ii} - \eta l \left( v_{ii}^l - r'V^l \right) v_{ii}^{l-1} + \eta ld^{l-1}r'\mu^{l-1}V^l \\
&\leq v_{ii} - \eta l v_{ii}^{2l-1} + 2\eta ld^{l-1}r'\mu^{l-1}V^l.
\end{aligned}
\tag{123}
$$

**Off-diagonal Residual Term:** $V(t)$

Finally, we characterize the dynamic of $V(t) = \max_{i \neq j} |v_{ij}(t)|$. To this goal, we first consider the dynamic of each $v_{ij}$ such that $i \neq j$. Without loss of generality, we assume that $v_{ij} \geq 0$. One can write

$$
\begin{aligned}
v_{ij}^+ &= v_{ij} + \eta l \left( \sigma_j - \sum_{i \in [r']} v_{ij}^l \right) v_{ij}^{l-1} - \eta \sum_{s \in [l-1]} s \sum_{\Lambda:|\Lambda|_j=s} \beta_\Lambda \left( \prod_{k \in \Lambda, k \neq j} v_{ik}^{|\Lambda|_k} \right) v_{ij}^{s-1} \\
&\leq V + \eta l \sigma_1 V^{l-1} + \eta l r'V^{2l-1} - \eta \sum_{s \in [l-1]} s \sum_{\Lambda:|\Lambda|_j=s} \beta_\Lambda \left( \prod_{k \in \Lambda, k \neq j} v_{ik}^{|\Lambda|_k} \right) v_{ij}^{s-1} \\
&\leq V + 2\eta l \sigma_1 V^{l-1} - \eta \sum_{s \in [l-1]} s \sum_{\Lambda:|\Lambda|_j=s} \beta_\Lambda \left( \prod_{k \in \Lambda, k \neq j} v_{ik}^{|\Lambda|_k} \right) v_{ij}^{s-1},
\end{aligned}
\tag{124}
$$

where in the last inequality we use the assumption $V \lesssim \sigma_1^{1/l} d^{-\frac{1}{l-1}}$. Similar to the previous case, we next bound $(A) = \beta_\Lambda \left( \prod_{k \in \Lambda, k \neq j} v_{ik}^{|\Lambda|_k} \right) v_{ij}^{s-1}$. First note that $|\Lambda|_j = s \in [l-1]$. Hence, we have that

$$
\begin{aligned}
(A) &= \sum_{s=1}^{r'} \prod_{k \in \Lambda} v_{sk}^{|\Lambda|_k} \left( \prod_{k \in \Lambda, k \neq j} v_{ik}^{|\Lambda|_k} \right) v_{ij}^{s-1} \\
&= \underbrace{\left( \prod_{k \in \Lambda, k \neq j} v_{ik}^{2|\Lambda|_k} \right) v_{ij}^{2s-1}}_{\geq 0} - r' \max_{h \neq i} \left| \prod_{k \in \Lambda} v_{hk}^{|\Lambda|_k} \left( \prod_{k \in \Lambda, k \neq j} v_{ik}^{|\Lambda|_k} \right) v_{ij}^{h-1} \right| \\
&\geq -r' \max_{h \neq i} \left| \prod_{k \in \Lambda} v_{hk}^{|\Lambda|_k} \left( \prod_{k \in \Lambda, k \neq j} v_{ik}^{|\Lambda|_k} \right) v_{ij}^{h-1} \right| \\
&\geq -r' \mu^{|\Lambda|_s + |\Lambda|_i} V^{2l - |\Lambda|_s - |\Lambda|_i - 1}.
\end{aligned}
\tag{125}
$$

We further note that $|\Lambda|_h + |\Lambda|_i \leq l - s$. Hence, it can be lower bounded by

$$
(A) \geq -r' \mu^{l-s} V^{l+s-1} \geq -r' \mu^{l-1} V^l.
\tag{126}
$$

Pluggin our estimation for $(A)$ into equation 124, we finally have

$$
v_{ij}^+ \leq V + 2\eta l \sigma_1 V^{l-1} + \eta l d^{l-1} r' \mu^{l-1} V^l \leq V + 3\eta l \sigma_1 V^{l-1}.
\tag{127}
$$

The last inequality comes from the assumption that $V \leq \frac{1}{d^l}$. This completes the proof of Proposition 7.

### G.3 PROOF OF PROPOSITION 8

In this section, we provide the proof of Proposition 8. We will show that the signal terms $v_{ii}(t), 1 \leq i \leq r$ quickly converge to $\sigma_i^{1/l}$, and the residual terms remain small. To this goal, we first study the dynamic of $V(t)$.

**Iteration complexity of the off-diagonal residual term $V(t)$.**

To start with the proof, we first study the time required for the off-diagonal term $V(t)$ to go from $V(0)$ to $2^{1/l} V(0)$, i.e., $T_1 = \min_{t \geq 0} \{ V(t) \geq 2^{1/l} V(0) \}$. By Proposition 7, we know

$$
V(t+1) \leq V(t) + 3\eta \sigma_1 l V^{l-1}(t) = \left( 1 + 3\eta l \sigma_1 V^{l-2}(t) \right) V(t).
\tag{128}
$$

Hence, for $0 \leq t \leq T_1$, we have

$$
V(t) \leq \prod_{s=0}^{t-1} \left( 1 + 3\eta l \sigma_1 V^{l-2}(s) \right) V(0) \leq \left( 1 + 6\eta l \sigma_1 V^{l-2}(0) \right)^t V(0).
$$

Note that $V(T_1) \geq 2^{1/l} V(0)$. Therefore,

$$
\left( 1 + 6\eta l \sigma_1 V^{l-2}(0) \right)^{T_1} V(0) \geq 2^{1/l} V(0).
\tag{129}
$$

Solving the above inequality for $T_1$, we obtain

$$
T_1 \geq \frac{\log(2^{1/l})}{\log(1 + 6\eta l \sigma_1 V^{l-2}(0))} \geq \frac{\log(2)}{6\eta l^2 \sigma_1 V^{l-2}(0)}.
\tag{130}
$$

On the other hand, our initial point satisfies $\sin(\boldsymbol{u}_i(0), \boldsymbol{z}_i) \leq \gamma$ and $\|\boldsymbol{u}_i(0)\| = \alpha^{1/l}$. Hence, we have $V(0) \leq \alpha^{1/l} \gamma$. Substituting this into the above equation, we conclude that

$$
T_1 \geq \frac{\log(2)}{6\eta l^2 \sigma_1 \gamma^{l-2}} \alpha^{-\frac{l-2}{l}}.
\tag{131}
$$

Note that $T_1 \geq \frac{\log(2)}{6\eta l^2 \sigma_1 \gamma^{l-2}} \alpha^{-\frac{l-2}{l}} \gg t^\star$. Hence, we have $V(t^\star) \leq 2^{1/l} V(0) \leq (2\alpha)^{1/l} \gamma$.

**Iteration complexity of the diagonal residual term** $v_{ii}(t), r + 1 \leq i \leq d$**.**

Next, we show that $v_{ii}(t), r + 1 \leq i \leq d$ will remain small during $0 \leq t \leq t^\star = \frac{8}{\eta l \sigma_r} \alpha^{-\frac{l-2}{l}}$. First by equation 123 in the proof of Lemma 6, we have for every $0 \leq t \leq t^\star$

$$v_{ii}(t + 1) \leq v_{ii}(t) + 2\eta l d^l \sigma_1^{l-1} V^l(t). \tag{132}$$

Note that $V(t) \leq 2^{1/l} V(0), \forall t \leq t^\star$, which leads to

$$\begin{aligned} v_{ii}(t) &\leq v_{ii}(0) + 4\eta l d^l \sigma_1^{l-1} \gamma^l \alpha t \\ &\leq v_{ii}(0) + 4\eta l d^l \sigma_1^{l-1} \gamma^l \alpha t^\star \\ &\leq \alpha^{1/l} + \mathcal{O}\left(d^l \kappa \alpha^{2/l} \gamma^l\right) \\ &\leq 2\alpha \end{aligned} \tag{133}$$

for $0 \leq t \leq t^\star$. Here we used the assumption that $\alpha \lesssim \frac{1}{d^{l3}}$ and $\gamma \lesssim \frac{1}{\kappa l}^{\frac{1}{l-2}}$.

**Iteration complexity of the signal term** $v_{ii}(t), 1 \leq i \leq r$**.**

As the last piece of the proof, we show that by iteration $t^\star = \frac{8}{\eta l \sigma_r} \alpha^{-\frac{l-2}{l}}$, the signal $v_{ii}, 1 \leq i \leq r$ will converge to the eigenvalue $\sigma_i^{1/l}$. First, recall that

$$\begin{aligned} v_{ii}(t + 1) &\geq v_{ii}(t) + \eta l \left(\sigma_i - v_{ii}^l(t) - 2d^{l-1} v_{ii}^{l-2}(t) V^2(t)\right) v_{ii}^{l-1}(t) - 2l d^l \eta \sigma_1^{l-1} V^l(t) \\ &\geq v_{ii}(t) + \eta l \left(\sigma_i - v_{ii}^l(t) - 4d^{l-1} v_{ii}^{l-2}(t) \alpha^{2/l} \gamma^2\right) v_{ii}^{l-1}(t) - 4l d^l \eta \sigma_1^{l-1} \gamma^l \alpha, \end{aligned} \tag{134}$$

where in the last inequality we used the fact that $V(t) \leq 2^{1/l} V(0)$ for $0 \leq t \leq t^\star$. In light of the above inequality, we characterize the convergence of $v_{ii}$ using a similar method as in (Ma & Fattahi, 2022a). In particular, we divide our analysis into two phases.

**Phase 1.** In the first phase, we have $v_{ii} \leq (0.5\sigma_i)^{1/l}$. First, since $v_{ii}(0) \geq \alpha^{1/l}\sqrt{1 - \gamma^2}$, we can easily conclude that $v_{ii}(t + 1) \geq v_{ii}(t)$ by induction. Hence, we can simplify the dynamic as

$$v_{ii}(t + 1) \geq \left(1 + \eta l (0.99\sigma_i - v_{ii}^l) v_{ii}^{l-2}(t)\right) v_{ii}(t) \geq \left(1 + 0.49\eta l \sigma_i v_{ii}^{l-2}(t)\right) v_{ii}(t). \tag{135}$$

Next, we further split the interval $\mathcal{I} = \left[0, 0.5\sigma_i^{1/l}\right]$ into $N = \mathcal{O}\left(\log\left(0.5\sigma_i^{1/l}/\alpha\right)\right)$ sub-intervals $\{\mathcal{I}_0, \cdots, \mathcal{I}_{N-1}\}$, where $\mathcal{I}_k = [2^k v_{ii}(0), 2^{k+1} v_{ii}(0))$. Let $\mathcal{T}_k$ collect the iterations that $v_{ii}$ spends in $\mathcal{I}_k$. Accordingly, let $|\mathcal{T}_k| = t_k$ be the number of iterations that $v_{ii}$ spends within $\mathcal{I}_k$. First note that $v_{ii}(t) \geq 2^k v_{ii}(0)$ for every $t \in \mathcal{T}_k$. Hence, we have

$$\left(1 + 0.49\eta l \sigma_i 2^{(l-2)k} v_{ii}^{l-2}(0)\right)^{t_k} \geq 2. \tag{136}$$

which implies

$$t_k \leq \frac{\log(2)}{0.49\eta l \sigma_i v_{ii}^{l-2}(0)} 2^{-(l-2)k}. \tag{137}$$

By summing over $k = 0, \cdots, N - 1$, we can upper bound the required number of iterations $T_3$

$$T_3 \leq \sum_{k=0}^{\infty} t_k \leq \sum_{k=0}^{\infty} \frac{\log(2)}{0.49\eta l \sigma_i v_{ii}^{l-2}(0)} 2^{-(l-2)k} \leq \frac{4}{\eta l \sigma_i \alpha^{l-2}} \leq \frac{4}{\eta l \sigma_r \alpha^{l-2}} \ll T_1, \tag{138}$$

where the last inequality is due to our assumption $\gamma \lesssim \frac{1}{\kappa l}^{\frac{1}{l-2}}$.

**Phase 2.** In the second phase, we have $v_{ii} \geq 0.5\sigma_i^{1/l}$. We further simplify equation 134 as

$$\begin{aligned} v_{ii}(t + 1) &\geq v_{ii}(t) + \eta l \left(\sigma_i - 8d^{l-1} \sigma_i^{\frac{l-2}{l}} \alpha^2 \gamma^2 - v_{ii}^l(t)\right) v_{ii}^{l-1}(t) \\ &\geq v_{ii}(t) + \eta l \left(\tilde{\sigma}_i - v_{ii}^l(t)\right) v_{ii}^{l-1}(t), \end{aligned} \tag{139}$$

where we denote $\tilde{\sigma}_i = \sigma_i - 8d^{l-1} \sigma_i^{\frac{l-2}{l}} \alpha^{2/l} \gamma^2$. Then, via a similar trick, within additional $T_4 \leq \frac{4}{\eta l \sigma_r} \alpha^{-\frac{l-2}{l}}$ iterations, we have $v_{ii}^l(t) \geq \sigma_i - 8d^{l-1} \sigma_i^{\frac{l-2}{l}} \alpha^{2/l} \gamma^2$. A similar argument on the upper bound shows $v_{ii}^l(t) \leq \sigma_i + 8d^{l-1} \sigma_i^{\frac{l-2}{l}} \alpha^{2/l} \gamma^2$, which completes the proof. $\square$

## H   AUXILIARY LEMMAS

**Lemma 7** (Bernoulli inequality). *For $0 \leq x < \frac{1}{r-1}$, and $r > 1$, we have*

$$(1 + x)^r \leq 1 + \frac{rx}{1 - (r - 1)x}. \tag{140}$$

