# OpenReview forum: "Behind the Scenes of Gradient Descent: A Trajectory Analysis via Basis Function Decomposition"
_ICLR.cc/2023/Conference — ICLR 2023 poster_

### Official Review · Reviewer_CW4U · 2022-10-17

**Confidence:** 3
**Clarity, Quality, Novelty And Reproducibility:** The paper is easy to follow. The over…
**Correctness:** 4
**Technical Novelty And Significance:** 2
**Empirical Novelty And Significance:** 2
**Recommendation:** 6

**Strength And Weaknesses:**

#Strength
- The paper proposes a natural framework to study gradient descent dynamics in Hilbert space.
- The paper recovers existing convergence results for symmetry matrix/ tensor decomposition.

# Weaknesses

- The GIA is too strong and almost trivializes the dynamics. My understanding is that the rich, structural, and coupled dynamics between eigenmodes are the most interesting part of GD + representation learning. Note that this framework does not even hold for symmetry matrix/tensor decomposition, which is much simpler than neural networks. Note that the authors argue that, empirically, the assumptions (GIA, Gradient dominant) approximately hold. However, I don't think such empirical results suffice to justify the assumptions in Theorem 1. For example $\sum_{j\neq i} \langle \nabla \beta(\theta_j) , \nabla\beta(\theta_i) \rangle$ can be large even each $\langle \nabla \beta(\theta_j) , \nabla\beta(\theta_i) \rangle$ is small.

- The framework does not yield new results or solve a new problem. Further work needs to be done to justify the usefulness of the framework.



**Summary Of The Paper:**

This paper studies the training dynamics of gradient descent (GD) on Hilbert Space ($L^2$). A key idea from this paper is to utilize orthogonalities from the Hilbert space to decompose the GD dynamics into a union of (possibly countably infinite) per eigenmode dynamics. In the general and realistic setting, the per eigenmode dynamics is in-trackable as each mode interacts with other modes in a complex and structural manner. To simplify the analysis, the authors make a key assumption, among many others: (`GIA`=Gradient Independent Assumption) gradients of the loss with respect to modes are `mutual` independent (orthogonal) throughout the whole training trajectory. This assumption reduces the original complex system into a 1-dimensional problem in the sense that each eigenmode evolves independently. With an additional Polyak-Lojasiewicz type assumption (called Gradient dominant in the paper), convergence to the global minima can be proved.

The GIA does not strictly hold in general (the only known example in the paper is the kernel regression). Nevertheless, the authors can apply the idea of eigendecomposition to symmetry matrix/tensor decomposition (with some additional assumptions, and finer analysis to control the coefficient gradients) and recover/improve convergence results from prior work.

Overall, I think the decomposition of the dynamics of GD into eigenmodes is natural and interesting. However, the GIA is too strong and unnatural, which trivializes the rich and complex dynamics of GD into a union of independent 1D dynamics.

**Summary Of The Review:**

The paper has several interesting insights. However, the assumptions on Theorem 1, in particular, the GIA, are too strong and unnatural, which basically remove all interesting structures in GD dynamics.

---

> ### Author Response · Authors · 2022-11-12
> **Response to Reviewer CW4U (1/2)**
>
> We are grateful that the reviewer thinks our idea is interesting. We kindly invite the reviewer to read our general response. In what follows, we try to address the raised comments.
>
> > - The GIA is too strong and almost trivializes the dynamics. My understanding is that the rich, structural, and coupled dynamics between eigenmodes are the most interesting part of GD + representation learning. Note that this framework does not even hold for symmetry matrix/tensor decomposition, which is much simpler than neural networks.
>
> We thank the reviewer for this comment. We believe that there is a misunderstanding regarding our assumptions. As the reviewer pointed out, the dynamic of GD in the parameter space of DNNs indeed resembles a rich, intertwined, and complex behavior. Indeed, our assumptions **do not** impose any independence condition on this dynamic; instead, it shows the possibility of reparameterizing the dynamic via a (potentially hidden) function basis that would decompose the trajectory of GD into a series of trajectories that are almost independent and monotone, while maintaining their rich and coupled dynamics in the original parameter space. We would like to argue that assuming the existence of such function basis is much milder than assuming independence in the original space. In fact, we provide both empirical and theoretical evidence that in a wide range of learning tasks, from symmetric matrix factorization to modern DNNs, these function basis exist and can be used to establish the convergence of GD. Indeed, the proper choice of the function basis may differ for different learning tasks, and it may even depend on the unknown optimal solution. Such flexibility in the choice of the basis function is precisely the reason why our framework has the potential to explain rather complex trajectories of GD.
>
> > - Note that the authors argue that, empirically, the assumptions (GIA, Gradient dominant) approximately hold. However, I don't think such empirical results suffice to justify the assumptions in Theorem 1. For example $\sum_{j\neq i}\langle\nabla \beta(\theta_j),\nabla \beta(\theta_i)\rangle$ can be large even each $\langle\nabla \beta(\theta_j),\nabla \beta(\theta_i)\rangle$ is small.
>
> We thank the reviewer for bringing up this great point. The reviewer is spot on in their observation that $\sum_{j\neq i}\langle\nabla \beta(\theta_j),\nabla \beta(\theta_i)\rangle$ may be large, even if the individual terms $\langle\nabla \beta(\theta_j),\nabla \beta(\theta_i)\rangle$ are small. To address the reviewer's comment, we provide a relaxed variant of GI condition that guarantees the main result of Theorem 1. Note that, according to Proposition 2, one can write
>
> $$\beta_i(\theta_{t+1})\approx\beta_i(\theta_{t})-\eta (\beta_i(\theta_t)-\beta_i^{\star})\Vert\nabla \beta(\theta_i)\Vert^2-\eta \sum_{j\neq i}(\beta_j(\theta_t)-\beta_j^{\star})\langle \nabla \beta(\theta_j),\nabla \beta(\theta_i)\rangle$$
>
> Therefore, if we can show the existence of some $0\leq \rho<1$ such that
>
> $$0\leq \frac{\sum_{j\neq i}(\beta_j(\theta_t)-\beta_j^{\star})\langle \nabla \beta(\theta_j),\nabla \beta(\theta_i)\rangle}{(\beta_i(\theta_t)-\beta_i^{\star})\Vert\nabla \beta(\theta_i)\Vert^2}\leq \rho$$
>
> then we can write
>
> $$\beta_i(\theta_{t+1})\approx\beta_i(\theta_{t})-\eta(1-\rho)(\beta_i(\theta_t)-\beta_i^{\star})\Vert\nabla \beta(\theta_i)\Vert^2$$
>
> and our analysis would go through line by line with the only difference that the stepsize $\eta$ is replaced by the "effective stepsize" $\eta(1-\rho)$. Therefore, it is left to see whether the existence of such $0\leq \rho<1$ is reasonable for DNNs. Using the same dataset as Figure 2, we plot the value of $\frac{\sum_{j\neq i}(\beta_j(\theta_t)-\beta_j^{\star})\langle \nabla \beta(\theta_j),\nabla \beta(\theta_i)\rangle}{(\beta_i(\theta_t)-\beta_i^{\star})\Vert\nabla \beta(\theta_i)\Vert^2}$ for $1\leq i\leq 10$ in the following link:
> https://anonymous.4open.science/r/ICLR-rebuttal-F0EE/GIA.pdf
>  It can be seen that $\rho$ can be picked as $0.7$, which implies that the assumption $\rho<1$ is reasonable. We will certainly revise Theorem 1 with this assumption and acknowledge the reviewer for bringing up this important point.

---

> > ### Author Response · Authors · 2022-11-12
> > **Response to Reviewer CW4U (2/2)**
> >
> > > - The framework does not yield new results or solve a new problem. Further work needs to be done to justify the usefulness of the framework.
> >
> > With all due respect, we disagree with the reviewer. Aside from the implications of Theorem 1 which were discussed in our general response, we would like to briefly summarize our theoretical contributions for symmetric matrix factorization (SMF) and orthogonal symmetric tensor decomposition (OSTD):
> >
> > 1. For SMF, we improve the convergence of GD with a larger stepsize and with a smaller error. In particular, we increase the allowable stepsize for GD from $\kappa^{-3}$ [1, 2] to $\sigma_1^{-1}$. Moreover, we improve the dependency of the final error on the initialization scale from $\alpha^{21/16}$ \[1\] (which is the best known error) to $\alpha^2$.
> >
> > 2. For (overparameterized) OSTD, we provide the *first* convergence proof of the vanilla GD; previous results on OSTD only apply to modified variants of GD [3, 4]. The main reason that has precluded the previous work from studying the vanilla GD for OSTD is the existence of higher-order saddle points in the landscape which are known to be problematic for vanilla GD. Using our proposed basis function decomposition, we have shown that vanilla GD can escape these higher-order saddle points in polynomial time.
> >
> > 3. We prove that GD with **large stepsize** enjoys an incremental learning phenomenon when applied to both SMF and OSTD. This is an important phenomenon that has been observed in the past [3], but not rigorously proven.
> >
> > We believe that all of the above results are new and important, and they crucially rely on our proposed basis function decomposition. We would be happy to further clarify our contributions, if needed.
> >
> >
> >
> > [1] Li, Yuanzhi, Tengyu Ma, and Hongyang Zhang. "Algorithmic regularization in over-parameterized matrix sensing and neural networks with quadratic activations." *Conference On Learning Theory*. PMLR, 2018.
> >
> > [2] Stöger, Dominik, and Mahdi Soltanolkotabi. "Small random initialization is akin to spectral learning: Optimization and generalization guarantees for overparameterized low-rank matrix reconstruction." *Advances in Neural Information Processing Systems* 34 (2021): 23831-23843.
> >
> > [3] Gissin, Daniel, Shai Shalev-Shwartz, and Amit Daniely. "The implicit bias of depth: How incremental learning drives generalization." *arXiv preprint arXiv:1909.12051* (2019).

---

> > > ### Comment · Reviewer_CW4U · 2022-11-21
> > > **update**
> > >
> > > I thank the authors for the clarification and updates, in particular, the new plot.
> > > However, proving the finiteness of $\rho$ in the above equation is highly non-trivial and is the core of the problem for neural networks. I agree doing eigenprojection is the first natural step. However, to make further progress, controlling the error terms $\sum_{j\neq i}$ is the key step and I would like to see more progress in this direction.
> > >
> > > I also increase the score to encourage the authors to continue working on this direction as it is promising.

---

> > > > ### Author Response · Authors · 2022-11-21
> > > > **Thanks for your support!**
> > > >
> > > > We thank the reviewer for their support. We agree with the reviewer that showing a small value for $\rho$ (the interaction among different basis) might be the key step toward understanding the optimization of practical neural networks. Backed by our simulations, we believe that such an assumption is realistic. As a future direction, we will certainly investigate different approaches to prove the validity of this assumption.

---

### Official Review · Reviewer_iyzY · 2022-10-23

**Confidence:** 2
**Correctness:** 3
**Technical Novelty And Significance:** 2
**Empirical Novelty And Significance:** 2
**Recommendation:** 6

**Clarity, Quality, Novelty And Reproducibility:**

Clarity: the paper is clear and easy to read.

Quality: the quality of this paper is fine, mainly on analyzing gradient descent for a few models using basis function decomposition.

Reproducibility: Not available.

**Strength And Weaknesses:**

Strength:

1. The paper is clearly written and easy to read.

2. The analysis seems new to me.

Weakness and questions to the authors (I am not an expert in this area):

1. For the motivating example, what is the behavior of SGD with momentum for training ResNet-18? I am asking as SGD with momentum is the baseline method for training ResNet-18.


2. What are the major advantages of the proposed analysis of GD via basis function decomposition over the existing analysis of GD? Indeed, we also analyze the eigendecomposition of the GD dynamics for the classical analysis.


3. In practice, we may not be able to get the exact gradient. Can authors comment on how to extend the analysis to stochastic gradient descent and related algorithms?


4. One weak point is the analysis relies on Assumption 1, which is much stronger than the assumption used in the classical analysis of gradient descent.


5. For neural networks, how to choose the orthogonal basis? Perhaps I missed something, but after reading the paper, I still do not know how to do this.


6. In Theorem 1, can authors comment on the choice of the step size \eta?


7. In section 3.1, how can we assume the kernel function are orthonormal for the neural tangent kernel?


**Summary Of The Paper:**

In this paper, the authors show how an appropriate basis function decomposition can be used to provide a much simpler convergence analysis for gradient-based algorithms on several representative learning problems, from simple kernel regression to complex DNNs. 1) The authors prove that GD learns the coefficients of an appropriate function basis that forms the true model. 2) The authors improve the convergence of GD on the symmetric matrix factorization and provide an entirely new convergence result for GD on the orthogonal symmetric tensor decomposition. 3) The authors show that different gradient-based algorithms monotonically learn the coefficients of a particular function basis defined as the eigenvectors of the conjugate kernel after training.


**Summary Of The Review:**

The paper proposes analyzing the convergence of gradient descent via basis function decomposition, which seems interesting. However, I have a few questions on this paper, see details in "Strength and Weaknesses"

---

> ### Author Response · Authors · 2022-11-10
> **Response to Reviewer iyzY (1/2)**
>
> We are grateful that the reviewer found our paper interesting. We kindly invite the reviewer to read our general response. In what follows, we try to address all of the raised comments.
>
> > 1. For the motivating example, what is the behavior of SGD with momentum for training ResNet-18? I am asking as SGD with momentum is the baseline method for training ResNet-18.
>
> Thanks for your comments! We run this simulation and the result is ([here](https://anonymous.4open.science/r/ICLR-rebuttal-F0EE/resnet18-sgd.pdf)). It can be seen that in this simulation the dynamics of each coefficient still have a monotonic behavior, which aligns with our theoretical results. Moreover, the final test accuracy is $94.45\%$, which implies that our result is beyond NTK. For the experimental setting, we set the learning rate to be $0.3$ (with $5$ warm-up epochs starting at $1\times 10^{-5}$); batch size $512$. We decay the learning rate with $0.33$ every $50$ epochs.
>
> > 2. What are the major advantages of the proposed analysis of GD via basis function decomposition over the existing analysis of GD? Indeed, we also analyze the eigendecomposition of the GD dynamics for the classical analysis.
>
> We thank the reviewer for this comment. To the best of our knowledge, the existing decomposition techniques for GD are developed in an ad hoc manner and are tailored to specific learning tasks. In contrast, our general Theorem 1 does not rely on any specific choice of the function basis; instead, it sets forth a series of conditions on the decomposition that would guarantee the convergence of GD. Indeed, any basis function decomposition that satisfies these conditions is a valid choice. In later parts, we provide concrete examples of function basis that satisfy these conditions. Indeed, some of our proposed function basis rely on the eigendecomposition of a certain matrix/tensor (the choice of this matrix/tensor is task-dependent), but there might exist other "good" choices of function basis. For instance, in Example 1 we propose a valid function basis for polynomial functions that does not rely on eigendecomposition.
>
> We hope to have convinced the reviewer that our approach is novel and merits further investigation. We would appreciate it very much if the reviewer could point out to any work that takes a similar approach, and we would be happy to compare our technique with those works.
>
> > 3. In practice, we may not be able to get the exact gradient. Can authors comment on how to extend the analysis to stochastic gradient descent and related algorithms?
>
> We truly appreciate the reviewer's insightful comment. It is indeed possible to extend our technique to inexact gradient descent. Here we take SGD as an example to illustrate one such possible way. For simplicity, we assume that at iteration $t$, we sample $x_t,y_t$ i.i.d. from the distribution $\mathcal{D}$. Hence, the update can be written as
>
> $\theta_{t+1}=\theta_t-\eta \nabla L_t(\theta_t)$
>
> where $L_t(\theta_t)=\frac{1}{2}(f_{\theta_t}(x_t)-y_t)^2$ is an unbiased estimator of $L(\theta_t)$. Hence, for each coefficient $\beta_i$, its dynamic can be characterized by
>
> $$\beta_i(\theta_{t+1})=\beta_i(\bar\theta_{t+1}+\eta (\nabla L(\theta_t)-\nabla L_t(\theta_t)))\approx \beta_i(\bar\theta_{t+1})+\eta\langle \nabla\beta_i(\bar\theta_{t+1}),\nabla L(\theta_t)-\nabla L_t(\theta_t)\rangle.$$
>
> Here $\bar\theta_{t+1}=\theta_t-\eta \nabla L(\theta_t)$ is the GD update. Under some standard assumptions on the variance of the gradient as well as the stepsize, the second term, which characterizes the difference between the stochastic gradient and the true gradient, can be controlled properly and the convergence of SGD can be established. We will make this argument precise in the revised manuscript.

---

> > ### Author Response · Authors · 2022-11-10
> > **Response to Reviewer iyzY (2/2)**
> >
> >
> >
> > > 4. One weak point is the analysis relies on Assumption 1, which is much stronger than the assumption used in the classical analysis of gradient descent.
> >
> > We start by agreeing with the reviewer that Assumption 1 is indeed strong and may not be satisfied in realistic settings. However, we would like to argue that relaxing this assumption for general learning problems is not an easy task and may not be even possible. In particular, the recent paper [1] has shown that in the absence of gradient Lipschitzness, the iterations of GD either approach a region of zero gradient infinitely often or diverge. As such, general convergence results for GD will inevitably rely on Assumption 1 (or its variants). On the positive side, however, we would like to mention that under a boundedness assumption on the solution trajectory (which is considerably milder) and twice differentiability of the model, Assumption 1 is automatically satisfied along the solution trajectory. This is important since, for a wide range of learning tasks, the boundedness of the iterations can be easily established. For instance, we show that the iterations of GD remain bounded for KR, SMT, and OSTD, thereby obviating the need for Assumption 1.
> >
> > > 5. For neural networks, how to choose the orthogonal basis? Perhaps I missed something, but after reading the paper, I still do not know how to do this.
> >
> > We kindly invite the reviewer to read **Appendix A.2**, where we explain how to choose the orthogonal basis and calculate the corresponding coefficients in detail. Due to space restrictions, we could not provide these details in the main body of the paper.
> >
> > > 6. In Theorem 1, can authors comment on the choice of the step size \eta?
> >
> > Thank you for raising this great question. Due to its generality, the convergence result for GD in Theorem 1 relies on a conservative step size that scales with $O(1/(d\log d)$. Indeed, GD with such a small stepsize is rarely used for DNN training. However, we would like to argue that Theorem 1 is useful for two reasons: first, by considering a conservative scenario, it takes the first step towards justifying the empirical promise of the proposed basis function decomposition technique shown in different realistic DNN training. Second, it lays the groundwork for proving/improving the convergence of GD with large step sizes on two important learning tasks, namely SMF and OSTD. Indeed, our new convergence results for SMF and OSTD are built upon Theorem 1 and would not have been possible without this theorem. Moreover, our experiments on CNN indicate that our framework empirically works in the large stepsize regime as well. We hope to extend our theoretical results to large stepsize in the future.
> >
> > > 7. In section 3.1, how can we assume the kernel function are orthonormal for the neural tangent kernel?
> >
> > Since the kernel functions are obtained based on the SVD of the neural tangent kernel, they are naturally orthogonal. We would also like to point out that for any set of kernel functions (not necessarily orthogonal), there exists a transformation that would make the kernel functions orthogonal. We kindly invite the reviewer to read Footnote 5 on Page 6, where we explain how to obtain such transformation.
> >
> >
> >
> > [1] Patel, Vivak, and Albert S. Berahas. "Gradient descent in the absence of global Lipschitz continuity of the gradients: Convergence, divergence and limitations of its continuous approximation." *arXiv preprint arXiv:2210.02418* (2022).

---

### Official Review · Reviewer_5bKV · 2022-10-27

**Confidence:** 3
**Correctness:** 3
**Technical Novelty And Significance:** 3
**Empirical Novelty And Significance:** 4
**Recommendation:** 6

**Clarity, Quality, Novelty And Reproducibility:**

the presentation is mostly clear. The paper is a bit analysis heavy so was a bit difficult to parse (maybe quite inevitable for this kind of result).

**Strength And Weaknesses:**

Strength: the authors find a quite interesting way to construct the orthogonal basis, and under some special circumstances (the applications), the basis can be explicitly written down. The analysis seems substantial.

Weakness: it would be helpful if the authors can clarify my two questions above.

**Summary Of The Paper:**

This paper studies the convergence behavior of SGD. Specifically, it says that there is a way to project the learning process onto a set of orthogonal basis functions so that the leading coefficients in the basis functions exhibit 'smoothly' convergence behavior. While not explicitly stated in the paper, I think that implies certain generalization properties of the SGD (still a bit confused see below). The authors then proceed to analyze three specific applications/models and demonstrate the power of their generic framework.

I have some clarification questions:

Q1. If I understand the result correctly, it is a convergence result of the optimization problem and it does not explicitly say anything about generalization errors. But because the coefficients of leading basis functions get stabilized over time, does that also imply some generalization result, and if so, is there an explicit way to characterize that.
Q2. For application 2 (matrix completion), the result looks too strong (also related to my Q1). Specifically, this is only a convergence result but at the end it also seems to imply all local optimal are the same. Ge, Jin, and Zheng's result made a quite heavy effort in matrix manipulation to prove the no spurious local optimal result. I wonder how this result manages to circumvent the technical barriers there. Also, it seems these results usually require some incoherence assumptions. I cannot find the counterpart in this paper too.

**Summary Of The Review:**

1. it is a strong result and presents a new and interesting way to interpret the SGD.
2. In fact, it actually appears to be too strong so I need some clarification.

---

> ### Author Response · Authors · 2022-11-10
> **Response to Reviewer 5bKV**
>
> We are glad to hear that the reviewer found our idea interesting. We kindly invite the reviewer to read our general response. In what follows, we provide point-by-point responses to the raised comments.
>
> > Q1. If I understand the result correctly, it is a convergence result of the optimization problem and it does not explicitly say anything about generalization errors. But because the coefficients of leading basis functions get stabilized over time, does that also imply some generalization result, and if so, is there an explicit way to characterize that.
>
> We thank the reviewer for this insightful comment. We point out that any meaningful result on the generalization must be in the *finite sample regime*, i.e., on the empirical loss. As mentioned in [1,2], the first step towards this goal is to analyze the behavior of the GD on the expected loss, i.e., when the number of samples approaches infinity. In this work, we take this first step by establishing the convergence of GD on the expected/population loss. It is worth noting that in this case, the solution recovered by GD coincides with the one that is closest to the ground truth (i.e., the one with the best generalization).  In our follow-up work, we plan to extend our result to the empirical loss by utilizing an orthogonal basis over the empirical distribution, i.e., an orthogonal basis in the metric $L^2(\mathcal{D}_n)$ instead of $L^2(\mathcal{D})$.
>
> > Q2. For application 2 (matrix completion), the result looks too strong (also related to my Q1). Specifically, this is only a convergence result but at the end it also seems to imply all local optimal are the same. Ge, Jin, and Zheng's result made a quite heavy effort in matrix manipulation to prove the no spurious local optimal result. I wonder how this result manages to circumvent the technical barriers there. Also, it seems these results usually require some incoherence assumptions. I cannot find the counterpart in this paper too.
>
> We thank the reviewer for this great question. We would like to point out that our studied problem, i.e., the symmetric matrix factorization (SMF), can be regarded as the population version of the matrix completion problem (equivalently, matrix completion with sampling probability $p=1$). The incoherence assumption is only needed in the subsampled regime with $p<1$, and not in the full sampling regime; e.g. see Section 4 in [5]. We will highlight this difference in the revised version.
>
> We would also like to point out that the absence of spurious local minima is neither necessary nor sufficient for the efficient convergence of GD. In particular, GD may converge efficiently despite the existence of spurious local (or even global) minima; see e.g. Figure 2 in [3]. Conversely, GD may take exponential time to converge even if the problem is devoid of spurious local minima; see e.g. Theorem 4.1 in [4]. Our implementation of GD starts at a small random initial point and converges to the ground truth along a specific path. Our result shows that along this path, there are no bad local minima, but does not say anything about points outside this path.
>
>
>
> [1] Ye, Tian, and Simon S. Du. "Global convergence of gradient descent for asymmetric low-rank matrix factorization." *Advances in Neural Information Processing Systems* 34 (2021): 1429-1439.
>
> [2] Nakkiran, Preetum, Behnam Neyshabur, and Hanie Sedghi. "The deep bootstrap framework: Good online learners are good offline generalizers." *arXiv preprint arXiv:2010.08127*(2020).
>
> [3] Ma, Jianhao, and Salar Fattahi. "Global Convergence of Sub-gradient Method for Robust Matrix Recovery: Small Initialization, Noisy Measurements, and Over-parameterization." *arXiv preprint arXiv:2202.08788* (2022).
>
> [4] Du, Simon S., Chi Jin, Jason D. Lee, Michael I. Jordan, Aarti Singh, and Barnabas Poczos. "Gradient descent can take exponential time to escape saddle points." *Advances in neural information processing systems 30* (2017).
>
> [5] Jin, Chi, Rong Ge, Praneeth Netrapalli, Sham M. Kakade, and Michael I. Jordan. "How to escape saddle points efficiently." *In International Conference on Machine Learning, pp. 1724-1732. PMLR,* 2017.

---

### Official Review · Reviewer_yaF6 · 2022-10-27

**Confidence:** 2
**Correctness:** 4
**Technical Novelty And Significance:** 3
**Empirical Novelty And Significance:** 3
**Recommendation:** 6

**Clarity, Quality, Novelty And Reproducibility:**

The content is clearly explained, with intuition, formal theorems, then discussion of the assumptions. The experiments support the claim.

I am unable to appreciate the novelty and correctness of the convergence rate proofs.

**Strength And Weaknesses:**

I am unable to fully evaluate this current work due to time mismanagement from my side, sorry.

Here are however a few remarks:
 - I would have appreciated (I am not an expert) a more comprehensive comparison to previous convergence rates (theorem 1 and proposition 2 e.g.)
 - in theorem 1, the required learning rate depends in $\frac{1}{\sqrt{d}\log{d}}$. For problems in high dimension, it implies a very small learning rate. Doesn't it force the GD dynamics to the gradient flow regime? In the CNN examples, it is common practice to use a learning rate as large as possible: would you say that your analysis still applies?
 - in section 3.4 "Surprisingly, A-CK [...] captures the underlining solution of different gradient-based algorithms" -> can you elaborate on what makes A-CK so special that its SVD can be used in your framework, instead of any other choice of basis. How is it "surprising" ?

**Summary Of The Paper:**

This paper analyzes the training dynamics of gradient descent based algorithms using a particular orthonormal basis for the function space spanned by the (function) iterates during training. Convergence to the function in the model space that minimizes the expected l2 loss is shown to occur at a novel convergence rate under some assumptions.

**Summary Of The Review:**

Sorry for a very lightweight review.

---

> ### Author Response · Authors · 2022-11-10
> **Response to Reviewer yaF6**
>
> We are grateful that the reviewer found our idea novel. We kindly invite the reviewer to read our general response. In what follows, we try to address all of the comments.
>
> > - I would have appreciated (I am not an expert) a more comprehensive comparison to previous convergence rates (theorem 1 and proposition 2 e.g.)
>
> We thank the reviewer for this great suggestion. To the best of our knowledge, Theorem 1 does not have any counterpart in the literature. On the contrary, the literature on SMF and OSTD is vast. For SMF, we show that GD converges with a larger stepsize and with a smaller error. In particular, we increase the allowable stepsize for GD from $\kappa^{-3}$ [1, 2] to $\sigma_1^{-1}$. Moreover, we improve the dependency of the final error on the initialization scale $\alpha$ from $\alpha^{21/16}$ \[1\] (which is the best-known error) to $\alpha^2$. For (overparameterized) OSTD, we provide the *first* convergence proof of the vanilla GD; previous results on OSTD only apply to modified variants of GD [3, 4]. The main reason that has precluded the previous work from studying the vanilla GD for OSTD is the existence of higher-order saddle points in the landscape which are known to be problematic for vanilla GD. Using our proposed basis function decomposition, we have shown that vanilla GD can escape these higher-order saddle points in polynomial time. We will further clarify these comparisons in the revised paper.
>
> > - in theorem 1, the required learning rate depends in $\frac{1}{\sqrt{d}\log⁡d}$. For problems in high dimension, it implies a very small learning rate. Doesn't it force the GD dynamics to the gradient flow regime? In the CNN examples, it is common practice to use a learning rate as large as possible: would you say that your analysis still applies?
>
> We truly appreciate the reviewer's insightful comment. The reviewer is indeed correct. We do not claim that Theorem 1 can readily explain the behavior of the currently used solvers in the realistic NNs. Indeed, GD with small stepsize is rarely used for DNN training. The main purpose of Theorem 1 is twofold: first, it takes the first step towards justifying the empirical promise of the proposed basis function decomposition technique shown in different realistic DNN training. Second, it lays the groundwork for proving/improving the convergence of GD with large step sizes on two important learning tasks, namely SMF and OSTD. Moreover, our experiments on CNN indicate that our framework empirically works in the large stepsize regime. We hope to extend our theoretical results to large stepsize in the future.
>
> > - in section 3.4 "Surprisingly, A-CK [...] captures the underlining solution of different gradient-based algorithms" -> can you elaborate on what makes A-CK so special that its SVD can be used in your framework, instead of any other choice of basis. How is it "surprising" ?
>
> We thank the reviewer for this constructive comment. We start by recalling that the A-CK regime relies on the SVD of the conjugate kernel after training. A-CK was introduced to explain the desirable generalization, robustness, and feature learning properties of DNNs at the **last epoch** [5], but it has never been used to explain the optimization of deep learning. We believe to be the first to show that A-CK can also be used to explain the optimization trajectory in DNNs. Our surprising observation is the fact that the SVD of the A-CK can be used as an effective basis, not only at the last epoch (as suggested in [5]) but also for the **entire solution trajectory**. We will further clarify this observation in the revised paper.
>
> In terms of other choices of basis, we do not rule out the existence of other function bases that can be used to study the optimization of DNNs. In fact, our Theorem 1 applies to *any* good choice of function basis that satisfies its assumption. Our proposed function basis based on A-CK is one such good choice, but there might exist other choices of function basis that can be used to "monotonize" the solution trajectory.
>
>
>
> [1] Li, Yuanzhi, Tengyu Ma, and Hongyang Zhang. "Algorithmic regularization in over-parameterized matrix sensing and neural networks with quadratic activations." *Conference On Learning Theory*. PMLR, 2018.
>
> [2] Stöger, Dominik, and Mahdi Soltanolkotabi. "Small random initialization is akin to spectral learning: Optimization and generalization guarantees for overparameterized low-rank matrix reconstruction." *Advances in Neural Information Processing Systems* 34 (2021): 23831-23843.
>
> [3] Wang, Xiang, et al. "Beyond lazy training for over-parameterized tensor decomposition." *Advances in Neural Information Processing Systems* 33 (2020): 21934-21944.
>
> [4] Ge, Rong, et al. "Understanding Deflation Process in Over-parametrized Tensor Decomposition." *Advances in Neural Information Processing Systems* 34 (2021): 1299-1311.
>
> [5] Long, Philip M. "Properties of the after kernel." *arXiv preprint arXiv:2105.10585* (2021).

---

> > ### Comment · Reviewer_yaF6 · 2022-11-18
> > **Thanks for your response**
> >
> > Out of curiosity, I was wondering if you would obtain similar results if you instead used the SVD of the final NTK rather than the A-CK, since it also somehow adapts to the task [1, 2]. It is of course not a last minute edit request :-)
> >
> > [1] Stanislav Fort, Gintare Karolina Dziugaite, Mansheej Paul, Sepideh Kharaghani, Daniel M. Roy, Surya Ganguli "Deep learning versus kernel learning: an empirical study of loss landscape geometry and the time evolution of the Neural Tangent Kernel"
> >
> > [2] Aristide Baratin, Thomas George, César Laurent, R Devon Hjelm, Guillaume Lajoie, Pascal Vincent, Simon Lacoste-Julien "Implicit Regularization via Neural Feature Alignment"

---

> > > ### Author Response · Authors · 2022-11-21
> > > **Technical Difficulty to apply after NTK**
> > >
> > > We thank the reviewer for this great question! NTK has achieved great success in the past few years. However, we conjecture that NTK might not be suitable in our setting. We briefly mention the technical difficulty in using the "after NTK" (A-NTK) framework, i.e., the first-order Taylor expansion of the trained model around the last epoch, to derive the basis function below. For simplicity, we consider the scalar case, that is, $f_{\theta}(x)\in\mathbb{R}$:
> > >
> > > $$f_{\theta_t}(x) - f_{\theta_{\infty}}(x)\approx \langle \nabla f_{\theta_{\infty}}(x),\theta_t-\theta_{\infty}\rangle.$$
> > >
> > > Here $\theta_{\infty}$ is the parameter in the last epoch. Given the above approximation based on A-NTK, we consider a general class of the basis functions in the form of $\phi_i(x)=\langle \nabla f\_{\theta_{\infty}}(x), w_i\rangle$, where the sequence of vectors $\\{w_i\\}\_{i\in I}$ are chosen such that the basis functions remain orthonormal. A crucial observation here is that, no matter what choices of $\\{w_i\\}\_{i\in I}$ we use, the corresponding coefficient in the last epoch will be zero, i.e.,  $\beta_i(\theta_{\infty})=0,\forall i$. Therefore, we cannot expect the dynamic of each coefficient $\beta_i(\theta_t)$ to be monotonic and characterize the evolution of the corresponding features.
> > >
> > > Here, we explain the main reason behind this shortcoming of A-NTK. As the reviewer pointed out, the NTK has been used in [1,2] to analyze the evolution of the feature space. Roughly speaking, when we study the evolution of the kernel in the training process, we calculate the difference between two kernels, where we normalize the eigenfunctions and only care about their directions. However, in our setting, we require information for both the direction and **magnitude**. As we demonstrated above, the magnitude will be zero in the last epoch, which prevents us from directly applying NTK in our setting.

---

### Author Response · Authors · 2022-11-10
**General Response**

We are glad that all of the reviewers think our proposed framework is novel and interesting. We also thank them for their constructive feedback. In this general response, our goal is to highlight our main contributions and answer some common concerns raised by some reviewers.


The high-level idea behind our result is the following:

> **Intuition:** Although the optimization trajectory in the **high-dimensional** parameter space can be overwhelmingly complex, it may be significantly simplified after projecting onto an appropriate choice of orthogonal function basis. In particular, the projected trajectory along each basis function can be analyzed in isolation, which in turn decomposes the highly complex dynamic of the solution trajectory into independent, **one-dimensional** trajectories in the projected space, which are potentially much more tractable to analyze.

Theorem 1 takes the first step towards formalizing the above intuition in the "ideal" scenario, where it is assumed that there exists a (potentially unknown) function basis with orthogonal gradients. Under this orthogonality assumption, we show that gradient descent with a special initialization and conservative stepsize converges to the true model. Admittedly, the existence of such an ideal function basis is not guaranteed; hence, the assumptions of Theorem 1 may not hold in realistic settings. However, through extensive simulations on different realistic DNNs, we hope to have convinced the reviewers that it is reasonable to assume the existence of candidate basis functions that *approximately* satisfy the assumptions of Theorem 1.

More importantly, Theorem 1 lays the groundwork for proving/improving the convergence of GD on several important learning tasks, namely symmetric matrix factorization (SMF) and orthogonal symmetric tensor decomposition (OSTD). For SMF, we show that GD converges with a larger stepsize and with a smaller error, improving upon the state-of-the-art results [1, 2]. For OSTD, we provide--to the best of our knowledge--the *first* convergence proof of the vanilla GD; previous results on OSTD only consider modified variants of GD [3, 4]. In addition to the aforementioned convergence results, we prove that GD with large stepsize enjoys incremental learning phenomenon; a phenomenon that has been observed in the past [5], but not rigorously proven.

In addition to the above theoretical results, we empirically show that similar behavior is consistently observed in more complex DNNs across different datasets, model architectures, and optimizers. We hope that our findings will inspire future efforts aimed at studying the optimization of deep learning through the lens of basis function decomposition.



[1] Li, Yuanzhi, Tengyu Ma, and Hongyang Zhang. "Algorithmic regularization in over-parameterized matrix sensing and neural networks with quadratic activations." *Conference On Learning Theory*. PMLR, 2018.

[2] Stöger, Dominik, and Mahdi Soltanolkotabi. "Small random initialization is akin to spectral learning: Optimization and generalization guarantees for overparameterized low-rank matrix reconstruction." *Advances in Neural Information Processing Systems* 34 (2021): 23831-23843.

[3] Wang, Xiang, et al. "Beyond lazy training for over-parameterized tensor decomposition." *Advances in Neural Information Processing Systems* 33 (2020): 21934-21944.

[4] Ge, Rong, et al. "Understanding Deflation Process in Over-parametrized Tensor Decomposition." *Advances in Neural Information Processing Systems* 34 (2021): 1299-1311.

[5] Gissin, Daniel, Shai Shalev-Shwartz, and Amit Daniely. "The implicit bias of depth: How incremental learning drives generalization." *arXiv preprint arXiv:1909.12051* (2019).

---

### Decision · Program_Chairs · 2023-01-20

**Decision:**

Accept: poster

**Justification For Why Not Higher Score:**

The assumptions required for this work are very strong.

**Justification For Why Not Lower Score:**

All reviewers appreciated that novel perspective on trajectory analysis. That being said, the average confidence is low and the paper could definitely be rejected.

**Metareview: Summary, Strengths And Weaknesses:**

This work analyzes training dynamics by finding a decomposition of the trajectory. The reviewers appreciated the insight provided by this perspective but they were also uncomfortable with the strength of the required assumptions.

That being said, they all leant toward acceptance and I follow their position.

**Note From Pc:**

if the above contains the word "oral" or "spotlight" please see: "oral" presentation means -> notable-top-5% and "spotlight" means -> notable-top-25%. As stated in our emails, we are disassociating presentation type from AC recommendations

**Summary Of Ac-Reviewer Meeting:**

There was no meeting because, despite my best attempts, I did not manage to find the time. This was compounded by the demotivating lack of replies from the reviewers.